# Beyond Grids: Multi-objective Bayesian Optimization With Adaptive Discretization

**Andi Nika**[†]
*Max Planck Institute for Software Systems*
*Germany*

*andinika@mpi-sws.org*

**Sepehr Elahi**[†]
*Department of Computer and Communication Sciences*
*EPFL*
*Switzerland*

*sepehr.elahi@epfl.ch*

**Çağın Ararat**
*Department of Industrial Engineering*
*Bilkent University*
*Turkey*

*cararat@bilkent.edu.tr*

**Cem Tekin**
*Department of Electrical and Electronics Engineering*
*Bilkent University*
*Turkey*

*cemtekin@ee.bilkent.edu.tr*

**Reviewed on OpenReview:** *https://openreview.net/forum?id=Wq15OHaRVE*

## Abstract

We consider the problem of optimizing a vector-valued objective function $\boldsymbol{f}$ sampled from a Gaussian Process (GP) whose index set is a well-behaved, compact metric space $(\mathcal{X}, d)$ of designs. We assume that $\boldsymbol{f}$ is not known beforehand and that evaluating $\boldsymbol{f}$ at design $x$ results in a noisy observation of $\boldsymbol{f}(x)$. Since identifying the Pareto optimal designs via exhaustive search is infeasible when the cardinality of $\mathcal{X}$ is large, we propose an algorithm, called Adaptive $\epsilon$-PAL, that exploits the smoothness of the GP-sampled function and the structure of $(\mathcal{X}, d)$ to learn fast. In essence, Adaptive $\epsilon$-PAL employs a tree-based adaptive discretization technique to identify an $\epsilon$-accurate Pareto set of designs in as few evaluations as possible. We provide both information-type and metric dimension-type bounds on the sample complexity of $\epsilon$-accurate Pareto set identification. We also experimentally show that our algorithm outperforms other Pareto set identification methods.

## 1 Introduction

Many complex scientific problems require the optimization of multi-dimensional ($m$-variate) performance metrics (objectives) under uncertainty. When developing a new drug, scientists need to identify the optimal therapeutic doses that maximize benefit and tolerability (Scmidt, 1988). When designing new hardware, engineers need to identify the optimal designs that minimize energy consumption and runtime (Almer et al., 2011). In general, there is no design that can simultaneously optimize all objectives, and hence, one seeks to identify the set of Pareto optimal designs. Moreover, design evaluations are costly, and thus, the optimal designs should be identified with as few evaluations as possible. In practice, this is a formidable task for at least two reasons: design evaluations only provide noisy feedback about ground truth objective values, and the set of designs to explore is usually very large (even infinite). Luckily, in practice, one only needs to identify the

---

[†] Work done while authors were at Bilkent University.

set of Pareto optimal designs up to a desired level of accuracy. Within this context, a practically achievable goal is to identify an $\epsilon$-accurate Pareto set of designs whose objective values form an $\epsilon$-approximation of the true Pareto front for a given $\boldsymbol{\epsilon} = [\epsilon^1, \ldots, \epsilon^m]^\mathsf{T} \in \mathbb{R}_+^m$ (Zuluaga et al., 2016).

There has been a considerable amount of interest in the Pareto set identification problem with a finite design space (Kone et al., 2023; Zuluaga et al., 2013; 2016), where individual designs are compared based on their vector values, according to multi-objective domination criteria. However, in many multi-objective problems of interest, such as the accurate tuning of particle accelerators (Roussel et al., 2021), or robotic systems control (Ariizumi et al., 2014), the design space is not necessarily finite. For these examples, a naive application of methods that are suitable for finite spaces might not yield the desired level of efficiency, due to the lack of rigorous theoretical foundations. Motivated by these observations, in this paper, we consider the Pareto set identification problem assuming a compact design space.

Specifically, we model the identification of an $\epsilon$-accurate Pareto set of designs as an active learning problem. We assume that the designs lie in a well-behaved, compact metric space $(\mathcal{X}, d)$, where the set of designs $\mathcal{X}$ might be very large. The vector-valued objective function $\boldsymbol{f} = [f^1, \ldots, f^m]^\mathsf{T}$ defined over $(\mathcal{X}, d)$ is unknown at the beginning of the experiment. The learner is given prior information about $\boldsymbol{f}$, which states that it is a sample from a multi-output GP with known mean and covariance functions. Then, the learner sequentially chooses designs to evaluate, where evaluating $\boldsymbol{f}$ at design $x$ immediately yields a noisy observation of $\boldsymbol{f}(x)$. The learner uses data from its past evaluations in order to decide which design to evaluate next until it can confidently identify an $\epsilon$-accurate Pareto set of designs.

**Main contribution.** We propose a new learning algorithm, called Adaptive $\epsilon$-PAL, that solves the Pareto active learning (PAL) problem described above, by performing as few design evaluations as possible. Our algorithm employs a tree-based adaptive discretization strategy to dynamically partition $\mathcal{X}$. It uses the GP posterior on $\boldsymbol{f}$ to decide which regions of designs in the partition of $\mathcal{X}$ to discard or to declare as a member of the $\epsilon$-accurate Pareto set of designs. On termination, Adaptive $\epsilon$-PAL guarantees that the returned set of designs forms an $\epsilon$-accurate Pareto set with a high probability. To the best of our knowledge, Adaptive $\epsilon$-PAL is the first algorithm that employs an adaptive discretization strategy in the context of PAL, which turns out to be very effective when dealing with a large $\mathcal{X}$.

We prove information-type and metric dimension-type upper bounds on the sample complexity of Adaptive $\epsilon$-PAL (Theorem 1). Our information-type bound yields a sample complexity upper bound of $\tilde{O}(g(\epsilon))$ where $\epsilon = \min_j \epsilon^j$, $g(\epsilon) = \min\{\tau \geq 1 : \sqrt{\gamma_\tau / \tau} < \epsilon\}$, and $\gamma_\tau$ is the maximum information gain after $\tau$ evaluations. To the best of our knowledge, this is the first information type bound for dependent objectives in the context of PAL. In addition, our metric dimension-type bound yields a sample complexity of $\tilde{O}(\epsilon^{-(\frac{\bar{D}}{\alpha} + 2)})$ for all $\bar{D} > D_1$, where $D_1$ represents the metric dimension of $(\mathcal{X}, d)$ and $\alpha \in (0, 1]$ represents the Hölder exponent of the metric induced by the GP on $\mathcal{X}$. As far as we know, this is the first metric dimension-type bound in the context of PAL. Our bounds complement each other: as we show in the appendix, neither of them dominates the other for all possible GP kernels. Specifically, we provide an example (Proposition 1) under which the information-type bound can be very loose compared to the metric dimension-type bound. Besides theory, we also show via simulations on multi-objective functions that Adaptive $\epsilon$-PAL significantly improves over its competitors in terms of accuracy. Furthermore, we point out the key challenges that necessitate novel algorithmic design and techniques as follows:

- First, due to the nature of the problem, we need to properly define novel confidence hyper-rectangle objects which are designed to capture uncertainty, not only over individual designs, but over whole regions. This requires integrating information and metric-type components into our hyper-rectangle definitions.
- Second, the refining, evaluating and discarding steps of our procedure are substantially different from their analogues in $\epsilon$-PAL (Zuluaga et al., 2016). Again, this is due to the large space of designs to consider. Here, we need to be confident about regions of space, rather than individual points, and choose whether to discard them or maintain them.
- Third, in addition to the Pareto front identification problem, inherent in $\epsilon$-PAL, our formulation also introduces the problem of efficiently expanding the search tree in promising directions (note that the

tree is not uniformly explored). Thus, we need additional analysis which involve information-theoretic and geometric components integrated into a threshold condition.

- Finally, all of the above necessitate new additional results which involve algebraic and analytic arguments to carry through the analysis. Specifically, results such as Lemma 8 (which requires a careful step-by-step analysis applicable only to our setting), Lemma 10, Lemma 12 and Lemma 16, are all results which require non-trivial and novel reasoning steps necessary for our setting.

**Organization.** We provide a detailed comparison with related work in Section 2. We explain the properties of the function to be optimized and the structure of the design space in Section 3. This is followed by the description of Adaptive $\epsilon$-PAL in Section 4. We give sample complexity bounds for Adaptive $\epsilon$-PAL in Section 5, discuss the main aspects of computational complexity analysis in Section 6. We devote Section 7 to the computational experiments, followed by conclusions in Section 8. We present the proof of the main theorem separately in Section B, the appendix. At the end of the paper, we include a table for the frequently used notation.

## 2 Related Work

This section provides a detailed discussion of related work on several lines of research.

### 2.1 Multi-objective optimization

Learning the Pareto optimal set of designs and the Pareto front has received considerable attention in recent years (Belakaria et al., 2024; Auer et al., 2016; Zuluaga et al., 2013; 2016; Hernández-Lobato et al., 2016; Shah & Ghahramani, 2016; Paria et al., 2020; Belakaria & Deshwal, 2019; Belakaria et al., 2020; Daulton et al., 2020; Alizadeh et al., 2024; Mukherjee et al., 2024).

Auer et al. (2016) consider a finite set of designs and formulate the identification of the Pareto front as a pure exploration multi-armed bandit (MAB) problem in the fixed confidence setting. Under the assumption that the centered outcomes are 1-subGaussian and independent, they provide gap-dependent bounds on its sample complexity, which, in the single objective case, yield the well-known gap-dependent sample-complexity bounds for pure exploration in the MAB setting (Mannor & Tsitsiklis, 2004). Other works on Multi-objective MAB include (Xu & Klabjan, 2023; Busa-Fekete et al., 2017; Öner et al., 2018). As opposed to their frequentist approach, our approach is Bayesian, which imposes a Gaussian process prior to the latent function of interest.

Knowles (2006) and Ponweiser et al. (2008) study the multi-objective optimization problem using the paradigm of Efficient Global Optimization (EGO) algorithm (Jones et al., 1998), a GP-based supervised learning approach used to tackle optimization problems with expensive evaluations. Knowles (2006) proposes ParEGO, which is an adaptation of EGO in the $m$-objective case. The author takes a scalarization approach to the problem, where the used acquisition function is the augmented Tchebycheff function, thus essentially reducing the problem to the single-objective one. Such an acquisition function was also used recently by Lin et al. (2022) for Pareto set learning. They extend previous approaches by identifying an approximate Pareto set of (potentially) infinitely many designs, as opposed to stopping with only a finite number of designs. In contrast, we take a direct Pareto set identification approach, in which we utilize the structure of the partial ordering relation between values of evaluated designs, while simultaneously integrating the GP posterior into the selection conditions of our method.

The second application of EGO, SMS-EGO (Ponweiser et al., 2008), does not reduce the problem to a single-objective one, but instead maximizes the gain in hypervolume from optimistic estimates based on a GP model. Having optimistically estimated the GP-based value vectors of their predicted Pareto set, they compute the hypervolume of the gain of choosing such design points. The hypervolume approach to Pareto learning is further exploited by Shah & Ghahramani (2016) and Daulton et al. (2020). In Shah & Ghahramani (2016), the Pareto hypervolume is defined in terms of an arbitrarily chosen reference point which is known to be suboptimal, and the current Pareto frontier. What makes their method attractive is its efficient implementation and computation, which relies on the approximation of a multi-dimensional integral. The computational complexity of computing the expected hypervolume gain is improved by extending the problem to the parallel constrained evaluation setting in Daulton et al. (2020). However, no one of these

Table 1: *Comparison with related works.* [1]Both the algorithm and the performance analysis take into account dependence between the objectives.

| Work | Design space | Function | Dep.[1] | Sample complex. bounds | Adaptive discret. |
|---|---|---|---|---|---|
| Auer et al. (2016) | Finite | Arbitrary | No | Gap-dependent | Not used |
| Zuluaga et al. (2013) | Finite | GP sample | No | Inf.-type | Not used |
| Zuluaga et al. (2016) | Finite | RKHS element | No | Inf.-type | Not used |
| Hernández-Lobato et al. (2016) | Bounded | GP sample | No | No bound | Not used |
| Shah & Ghahramani (2016) | Bounded | GP sample | Yes | No bound | Not used |
| Paria et al. (2020) | Compact | GP sample | No | Bayes Regret | Not used |
| Belakaria & Deshwal (2019) | General | GP sample | No | Regret norm | Not used |
| Belakaria et al. (2020) | Continuous | GP sample | No | Regret norm | Not used |
| Daulton et al. (2020) | Bounded | GP sample | No | Exp. Hypervol. | Not used |
| **Ours** | Compact | GP sample | Yes | Inf. & dim.-type | Used |

methods come with theoretical convergence guarantees, while we provide a comprehensive best-of-both-world type of convergence guarantees under minimal assumptions.

Furthermore, Hernández-Lobato et al. (2016) consider a (possibly infinite) bounded design space $\mathcal{X}$ and assume that the individual objectives are samples from independent GPs. Their algorithm, Predictive Entropy Search for Multi-objective Optimization (PESMO), sequentially queries designs that maximize the acquisition function defined as the expected reduction in the entropy of the posterior distribution over the predicted Pareto set, given the previously sampled data. They provide comprehensive experimental comparisons between PESMO and other multi-objective optimization methods over various objectives and dimensions, in both noisy and noiseless cases. The comparison is done with respect to the relative difference between the hypervolume of the predicted set and the maximum such hypervolume for the given number of evaluations. Although their setup is similar to ours, we take a different approach to the Pareto set identification and provide theoretical guarantees for our method. Recently, Tu et al. (2022) extend the Predictive Entropy Search paradigm to that of Joint Entropy Search (JES), which takes into account the informativeness coming from both the Pareto set designs and their outcome vectors in their acquisition function.

Apart from Pareto front identification, several works consider identifying designs that satisfy certain performance criteria. For instance, Katz-Samuels & Scott (2018) consider the problem of identifying designs whose objective values lie in a given polyhedron in the fixed confidence setting. On the other hand, Locatelli et al. (2016) consider the problem of identifying designs whose objective values are above a given threshold in the fixed budget setting. In addition, Gotovos (2013) consider level set identification when $f$ is a sample from a GP. There also exists a plethora of works developing algorithms for best arm identification in the context of single-objective pure-exploration MAB problems such as the ones by Mannor & Tsitsiklis (2004); Bubeck et al. (2009); Gabillon et al. (2012).

## 2.2 Adaptive discretization

Adaptive discretization is a technique that is mainly used in regret minimization in MAB problems on metric spaces (Kleinberg et al., 2008; Bubeck et al., 2011), including contextual MAB problems (Shekhar et al., 2018), when dealing with large arm and context sets. It consists of adaptively partitioning the ground set along a tree structure into smaller and smaller regions, until, theoretically, the regions converge to a single point (under uniqueness assumptions in the single-objective case). Different from other upper confidence bound-based methods, here, the usual upper confidence bound of a given point $x$ (in both frequentist and Bayesian approaches) is inflated with a factor times the diameter of the region containing $x$, so that the uncertainty coming from the variation of the function values inside the region is also captured. A key efficacy of adaptive discretization comes from the fact that it does not blindly sample the space without first exhaustively partitioning it as long as it is certain. This certainty is formalized by comparing the sample uncertainty diameter with the region diameter. If the latter exceeds the former, then the algorithm decides

to partition the given region into smaller sub-regions. It is known that adaptive discretization can result in a much smaller regret compared to uniform discretization. However, this is significantly different from employing adaptive discretization in the context of PAL. While the regret can be minimized by quickly identifying one design that yields the highest expected reward, PAL requires identifying all designs that can form an $\epsilon$-Pareto front together, while at the same time discarding all designs that are far from being Pareto optimal. Table 1 compares our approach with the related work.

### 2.3  $\epsilon$-**Pareto active learning**

The line of work on which our method builds is that of Zuluaga et al. (2013; 2016). Zuluaga et al. (2016) consider a finite design space and assume that each objective is a sample from an independent GP. We briefly describe the rationale of their algorithm, since it will also be used later on. Their algorithm, $\epsilon$-Pareto Active Learning ($\epsilon$-PAL), is a confidence bound-based method. It partitions the design space into three subsets: the set of undecided designs, that of predicted Pareto-optimal designs, and that of predicted suboptimal designs. The procedure is composed of four main phases. In the first phase, the algorithm uses the posterior estimates in order to compute the confidence hyper-rectangles of designs that are still up for selection. In the second phase, each design is associated with an uncertainty region, dependent on all previous confidence hyper-rectangles, and then checked whether it is safe to discard it. The third phase consists of deciding whether there are any designs that can be safely predicted to be Pareto optimal. In the final phase, $\epsilon$-PAL decides whether or not to evaluate the design of maximal uncertainty.

The algorithm we propose utilizes a similar rationale to that of $\epsilon$-PAL. However, such an extension, although seemingly natural, also brings with it several challenges and key differences, which necessitate novel ways of solving the problem. We summarize these differences in the following.

First, $\epsilon$-PAL iterates through all designs, maintaining relevant statistics of them which it uses to decide whether they are reasonable candidates for Pareto optimality, or whether they can be safely discarded. In our case, this would be a futile attempt, since the design space is potentially infinite. Therefore, we use adaptive discretization techniques (Bubeck et al., 2011) in order to tackle the problem. Next, the application of adaptive discretization in the context of PAL introduces additional technical intricacies: instead of working over individual designs, our method maintains statistics over nodes (centers) of design regions with similar values, and, as a result, with similar levels of confidence. Thus, we design novel bonus terms for each node, which simultaneously take into account both the hyper-rectangular uncertainty and the structural relationship of "nearby"[1] nodes.

On the algorithmic front, the successful integration of adaptive discretization into the PAL setting implies a new evaluation procedure. We part from the $\epsilon$-PAL evaluation subroutine and introduce a new condition, which captures the uncertainty over a given region and implies that the algorithm will evaluate a design from the region only when it is absolutely necessary, thus optimizing the sample complexity of the overall procedure. On the theoretical front, on top of the information-type sample complexity bounds provided by Zuluaga et al. (2016), we additionally provide dimension-type bounds, thus yielding best-of-both-worlds bounds. We do this motivated by examples in which the information-type bounds might actually be loose (see Proposition 1). To make our theoretical analysis rigorous, we prove several additional results that allow for a comprehensive understanding of our bounds. To the best of our knowledge, we are the first to provide such a full comprehensive analysis for both types of bounds, while simultaneously accounting for adaptive discretization in the context of PAL.

## 3  **Background and formulation**

Throughout the paper, let us fix a positive integer $m \geq 2$ and a compact metric space $(\mathcal{X}, d)$. We denote by $\mathbb{R}^m$ the $m$-dimensional Euclidean space and by $\mathbb{R}^m_+$ the set of all vectors in $\mathbb{R}^m$ with nonnegative components. We write $[m] = \{1, \ldots, m\}$. Given a function $\boldsymbol{f} \colon \mathcal{X} \to \mathbb{R}^m$ and a set $\mathcal{S} \subseteq \mathcal{X}$, we denote by $\boldsymbol{f}(\mathcal{S}) = \{\boldsymbol{f}(x) \colon x \in \mathcal{S}\}$ the image of $\mathcal{S}$ under $\boldsymbol{f}$. Given $x \in \mathcal{X}$ and $r \geq 0$, $B(x, r) = \{y \in \mathcal{X} \colon d(x, y) \leq r\}$

---

[1]We assume a tree structure defined over the design space, where parent-child relationships are properly defined between relevant nodes. See Definition 10.

denotes the closed ball centered at $x$ with radius $r$. For a non-empty set $\mathcal{S} \subseteq \mathbb{R}^m$, let $\partial\mathcal{S}$ denote its boundary. If another non-empty set $\mathcal{S}' \subseteq \mathbb{R}^m$ is given, then we define the Minkowski sum and difference of $\mathcal{S}$ and $\mathcal{S}'$ as $\mathcal{S} + \mathcal{S}' = \{\boldsymbol{\mu} + \boldsymbol{\mu}' : \boldsymbol{\mu} \in \mathcal{S}, \boldsymbol{\mu}' \in \mathcal{S}'\}$, $\mathcal{S} - \mathcal{S}' = \{\boldsymbol{\mu} - \boldsymbol{\mu}' : \boldsymbol{\mu} \in \mathcal{S}, \boldsymbol{\mu}' \in \mathcal{S}'\}$, respectively. For a vector $\boldsymbol{\mu}'' \in \mathbb{R}^m$, we define $\boldsymbol{\mu}'' + \mathcal{S} = \{\boldsymbol{\mu}''\} + \mathcal{S}$.

## 3.1 Multi-objective optimization

A multi-objective optimization problem is an optimization problem that involves multiple objective functions (Hwang & Masud, 2012). Formally, letting $f^j : \mathcal{X} \to \mathbb{R}$ be a function for every $j \in [m]$, we write

$$\text{maximize } [f^1(x), \dots, f^m(x)]^\mathsf{T} \text{ subject to } x \in \mathcal{X},$$

where $m \geq 2$ is the number of objectives and $\mathcal{X}$ is the set of designs. We refer to the vector of all objectives evaluated at design $x \in \mathcal{X}$ as $\boldsymbol{f}(x) = [f^1(x), \dots, f^m(x)]^\mathsf{T}$. The objective space is given as $\boldsymbol{f}(\mathcal{X}) \subseteq \mathbb{R}^m$. In order to define a set of Pareto optimal designs in $\mathcal{X}$, we first describe several order relations on $\mathbb{R}^m$.

**Definition 1.** *For $\boldsymbol{\mu}, \boldsymbol{\mu}' \in \mathbb{R}^m$, we say that: (1) $\boldsymbol{\mu}$ is weakly dominated by $\boldsymbol{\mu}'$, written as $\boldsymbol{\mu} \preceq \boldsymbol{\mu}'$, if $\mu_j \leq \mu_j'$ for every $j \in [m]$. (2) $\boldsymbol{\mu}$ is dominated by $\boldsymbol{\mu}'$, written as $\boldsymbol{\mu} \prec \boldsymbol{\mu}'$, if $\boldsymbol{\mu} \preceq \boldsymbol{\mu}'$ and there exists $j \in [m]$ with $\mu_j < \mu_j'$. (3) For $\boldsymbol{\epsilon} \in \mathbb{R}_+^m$, $\boldsymbol{\mu}$ is $\boldsymbol{\epsilon}$-dominated by $\boldsymbol{\mu}'$, written as $\boldsymbol{\mu} \preceq_{\boldsymbol{\epsilon}} \boldsymbol{\mu}'$, if $\boldsymbol{\mu} \preceq \boldsymbol{\mu}' + \boldsymbol{\epsilon}$. (4) $\boldsymbol{\mu}$ is incomparable with $\boldsymbol{\mu}'$, written as $\boldsymbol{\mu} \parallel \boldsymbol{\mu}'$, if neither $\boldsymbol{\mu} \prec \boldsymbol{\mu}'$ nor $\boldsymbol{\mu}' \prec \boldsymbol{\mu}$ holds.*

Based on Definition 1, we define the following induced relations on $\mathcal{X}$.

**Definition 2.** *For designs $x, y \in \mathcal{X}$, we say that: (1) $x$ is weakly dominated by $y$, written as $x \preceq y$, if $\boldsymbol{f}(x) \preceq \boldsymbol{f}(y)$. (2) $x$ is dominated by $y$, written as $x \prec y$, if $\boldsymbol{f}(x) \prec \boldsymbol{f}(y)$. (3) For $\boldsymbol{\epsilon} \in \mathbb{R}_+^m$, $x$ is $\boldsymbol{\epsilon}$-dominated by $y$, written as $x \preceq_{\boldsymbol{\epsilon}} y$, if $\boldsymbol{f}(x) \preceq_{\boldsymbol{\epsilon}} \boldsymbol{f}(y)$. (4) $x$ is incomparable with $y$, written as $x \parallel y$, if $\boldsymbol{f}(x) \parallel \boldsymbol{f}(y)$.*

Note that, while the relation $\preceq$ on $\mathbb{R}^m$ (Definition 1) is a partial order, the induced relation $\preceq$ on $\mathcal{X}$ (Definition 2) is only a preorder since it does not satisfy antisymmetry in general. If a design $x \in \mathcal{X}$ is not dominated by any other design, then we say that $x$ is *Pareto optimal*. The set of all Pareto optimal designs is called the *Pareto set* and is denoted by $\mathcal{O}(\mathcal{X})$. The *Pareto front* is defined as $\mathcal{Z}(\mathcal{X}) = \partial(\boldsymbol{f}(\mathcal{O}(\mathcal{X})) - \mathbb{R}_+^m)$.

We assume that $\boldsymbol{f}$ is not known beforehand and formalize the goal of identifying $\mathcal{O}(\mathcal{X})$ as a sequential decision-making problem. In particular, we assume that evaluating $\boldsymbol{f}$ at design $x$ results in a noisy observation of $\boldsymbol{f}(x)$. The exact identification of the Pareto set and the Pareto front using a small number of evaluations is, in general, not possible under this setup, especially when the cardinality of $\mathcal{X}$ is infinite or a very large finite number. A realistic goal is to identify the Pareto set and the Pareto front in an approximate sense, given a desired level of accuracy that can be specified as an input $\boldsymbol{\epsilon}$. Therefore, our goal in this paper is to identify an $\boldsymbol{\epsilon}$-accurate Pareto set (see Definition 5) that contains a set of near-Pareto optimal designs by using as few evaluations as possible. Next, we define the $\boldsymbol{\epsilon}$-Pareto front and $\boldsymbol{\epsilon}$-accurate Pareto set associated with $\mathcal{X}$.

**Definition 3.** *Given $\boldsymbol{\epsilon} \in \mathbb{R}_+^m$, the set $\mathcal{Z}_{\boldsymbol{\epsilon}}(\mathcal{X}) = (\boldsymbol{f}(\mathcal{O}(\mathcal{X})) - \mathbb{R}_+^m) \setminus (\boldsymbol{f}(\mathcal{O}(\mathcal{X})) - 2\boldsymbol{\epsilon} - \mathbb{R}_+^m)$ is called the $\boldsymbol{\epsilon}$-Pareto front of $\mathcal{X}$.*

Roughly speaking, the $\boldsymbol{\epsilon}$-Pareto front can be thought of as the slab of points of width $2\boldsymbol{\epsilon}$ in $\mathbb{R}^m$ adjoined to the lower side of the Pareto front.

**Definition 4.** *Given $\boldsymbol{\epsilon} \in \mathbb{R}_+^m$ and $\mathcal{S} \subseteq \mathbb{R}^m$, a non-empty subset $\mathcal{C}$ of $\mathcal{S}$ is called an $\boldsymbol{\epsilon}$-covering of $\mathcal{S}$ if for every $\boldsymbol{\mu} \in \mathcal{S}$, there exists $\boldsymbol{\mu}' \in \mathcal{C}$ such that $\boldsymbol{\mu} \preceq_{\boldsymbol{\epsilon}} \boldsymbol{\mu}'$.*

**Definition 5.** *Given $\boldsymbol{\epsilon} \in \mathbb{R}_+^m$, a subset $\mathcal{O}_{\boldsymbol{\epsilon}}$ of $\mathcal{X}$ is called an $\boldsymbol{\epsilon}$-accurate Pareto set if $\boldsymbol{f}(\mathcal{O}_{\boldsymbol{\epsilon}})$ is an $\boldsymbol{\epsilon}$-covering of $\mathcal{Z}_{\boldsymbol{\epsilon}}(\mathcal{X})$.*

Note that the front associated with an $\boldsymbol{\epsilon}$-accurate Pareto set is a subset of the $\boldsymbol{\epsilon}$-Pareto front. As mentioned in (Zuluaga et al., 2016), an $\boldsymbol{\epsilon}$-accurate Pareto set is a natural substitute of the Pareto set since any $\boldsymbol{\epsilon}$-accurate Pareto design is guaranteed to be no worse than $2\boldsymbol{\epsilon}$ of any Pareto optimal design.

## 3.2 Prior knowledge on $\boldsymbol{f}$

We model the vector $\boldsymbol{f} = [f^1, \dots, f^m]^\mathsf{T}$ of objective functions as a realization of an $m$-output GP with zero mean, i.e., $\boldsymbol{\mu}(x) = 0$ for all $x \in \mathcal{X}$, and some positive definite covariance function $\boldsymbol{k}$.

**Definition 6.** *An $m$-**output GP** with index set $\mathcal{X}$ is a collection $(\boldsymbol{f}(x))_{x\in\mathcal{X}}$ of $m$-dimensional random vectors which satisfies the property that $(\boldsymbol{f}(x_1),\ldots,\boldsymbol{f}(x_n))$ is a Gaussian random vector for all $\{x_1,\ldots,x_n\}\subseteq\mathcal{X}$ and $n\in\mathbb{N}$. The probability law of an $m$-output GP $(\boldsymbol{f}(x))_{x\in\mathcal{X}}$ is uniquely specified by its (vector-valued) mean function $x\mapsto\boldsymbol{\mu}(x)=\mathbb{E}[\boldsymbol{f}(x)]\in\mathbb{R}^m$ and its (matrix-valued) covariance function $(x_1,x_2)\mapsto\boldsymbol{k}(x_1,x_2)=\mathbb{E}[(\boldsymbol{f}(x_1)-\boldsymbol{\mu}(x_1))(\boldsymbol{f}(x_2)-\boldsymbol{\mu}(x_2))^\mathsf{T}]\in\mathbb{R}^{m\times m}$.*

Functions generated from a GP naturally satisfy smoothness conditions which are very useful while working with metric spaces, as indicated by the following remark.

**Remark 1.** *Let $g$ be a zero-mean, single-output GP with index set $\mathcal{X}$ and covariance function $k$. The metric $l$ induced by the GP on $\mathcal{X}$ is defined as $l(x_1,x_2)=\left(\mathbb{E}[(g(x_1)-g(x_2))^2]\right)^{1/2}=(k(x_1,x_1)+k(x_2,x_2)-2k(x_1,x_2))^{1/2}$. This gives us the following tail bound for $x_1,x_2\in\mathcal{X}$, and $a\geq 0$: $\mathbb{P}(|g(x_1)-g(x_2)|\geq a)\leq 2\exp\left(-a^2/(2l^2(x_1,x_2))\right)$.*

Let us fix an integer $T\geq 1$. We consider a finite sequence $\tilde{x}_{[T]}=[\tilde{x}_1,\ldots,\tilde{x}_T]^\mathsf{T}$ of designs with the corresponding vector $\boldsymbol{f}_{[T]}=[\boldsymbol{f}(\tilde{x}_1)^\mathsf{T},\ldots,\boldsymbol{f}(\tilde{x}_T)^\mathsf{T}]^\mathsf{T}$ of unobserved objective values and the ($mT$-dimensional) vector $\boldsymbol{y}_{[T]}=[\boldsymbol{y}_1^\mathsf{T},\ldots,\boldsymbol{y}_T^\mathsf{T}]^\mathsf{T}$ of observations, where

$$\boldsymbol{y}_\tau=\boldsymbol{f}(\tilde{x}_\tau)+\boldsymbol{\kappa}_\tau$$

is the observation that corresponds to $\tilde{x}_\tau$ and $\boldsymbol{\kappa}_\tau=[\kappa_\tau^1,\ldots,\kappa_\tau^m]^\mathsf{T}$ is the noise vector that corresponds to this particular evaluation for each $\tau\in[T]$.

The posterior distribution of $\boldsymbol{f}$ given $\boldsymbol{y}_{[T]}$ is that of an $m$-output GP with mean function $\boldsymbol{\mu}_T$ and covariance function $\boldsymbol{k}_T$ given by

$$\boldsymbol{\mu}_T(x)=\boldsymbol{k}_{[T]}(x)(\boldsymbol{K}_{[T]}+\boldsymbol{\Sigma}_{[T]})^{-1}\boldsymbol{y}_{[T]}^\mathsf{T}$$

and

$$\boldsymbol{k}_T(x,x')=\boldsymbol{k}(x,x')-\boldsymbol{k}_{[T]}(x)(\boldsymbol{K}_{[T]}+\boldsymbol{\Sigma}_{[T]})^{-1}\boldsymbol{k}_{[T]}(x')^\mathsf{T}$$

for all $x,x'\in\mathcal{X}$, where $\boldsymbol{k}_{[T]}(x)=[\boldsymbol{k}(x,\tilde{x}_1),\ldots,\boldsymbol{k}(x,\tilde{x}_T)]\in\mathbb{R}^{m\times mT}$,

$$\boldsymbol{K}_{[T]}=\begin{bmatrix}\boldsymbol{k}(\tilde{x}_1,\tilde{x}_1),&\ldots,&\boldsymbol{k}(\tilde{x}_1,\tilde{x}_T)\\\vdots&&\vdots\\\boldsymbol{k}(\tilde{x}_T,\tilde{x}_1),&\ldots,&\boldsymbol{k}(\tilde{x}_T,\tilde{x}_T)\end{bmatrix},\boldsymbol{\Sigma}_{[T]}=\begin{bmatrix}\sigma^2\boldsymbol{I}_m,&\boldsymbol{0}_m,&\ldots,&\boldsymbol{0}_m\\\vdots&&&\vdots\\\boldsymbol{0}_m,&\boldsymbol{0}_m,&\ldots,&\sigma^2\boldsymbol{I}_m\end{bmatrix}\in\mathbb{R}^{mT\times mT},$$

$\boldsymbol{I}_m$ denotes the $m\times m$-dimensional identity matrix, and $\boldsymbol{0}_m$ is the $m\times m$-dimensional zero matrix. Note that this posterior distribution captures the uncertainty in $\boldsymbol{f}(x)$ for all $x\in\mathcal{X}$. In particular, the posterior distribution of $\boldsymbol{f}(x)$ is $\mathcal{N}(\boldsymbol{\mu}_T(x),\boldsymbol{k}_T(x,x))$; and for each $j\in[m]$, the posterior distribution of $f^j(x)$ is $\mathcal{N}(\mu_T^j(x),(\sigma_T^j(x))^2)$, where $(\sigma_T^j(x))^2=k_T^{jj}(x,x)$. Moreover, the distribution of the corresponding observation $\boldsymbol{y}$ is $\mathcal{N}(\boldsymbol{\mu}_T(x),\boldsymbol{k}_T(x,x)+\sigma^2\boldsymbol{I}_m)$.

### 3.3 Information gain

Since we aim at finding an $\boldsymbol{\epsilon}$-accurate Pareto set in as few evaluations as possible, we need to learn the most informative designs. In order to do that, we will make use of the notion of *information gain.* Our sample complexity result in Theorem 1 depends on the maximum information gain.

In Bayesian experimental design, the informativeness of a finite sequence $\tilde{x}_{[T]}$ of designs is quantified by $I(\boldsymbol{y}_{[T]};\boldsymbol{f}_{[T]})=H(\boldsymbol{y}_{[T]})-H(\boldsymbol{y}_{[T]}|\boldsymbol{f}_{[T]})$, where $H(\cdot)$ denotes the entropy of a random vector and $H(\cdot|\boldsymbol{f}_{[T]})$ denotes the conditional entropy of a random vector with respect to $\boldsymbol{f}_{[T]}$. This measure is called the information gain, which gives us the decrease of entropy of $\boldsymbol{f}_{[T]}$ given the observations $\boldsymbol{y}_{[T]}$. We define the *maximum information gain* as $\gamma_T=\max_{\boldsymbol{y}_{[T]}}I(\boldsymbol{y}_{[T]};\boldsymbol{f}_{[T]})$.

# 4 Adaptive $\epsilon$-PAL algorithm

In this section, we introduce our algorithm, *Adaptive $\boldsymbol{\epsilon}$-Pareto Active Learning* (PAL), which builds on the $\epsilon$-PAL algorithm of Zuluaga et al. (2016) and utilizes adaptive discretization techniques to enable an efficient navigation of the continuous search space. The algorithm is largely technical, thus we divide the description of each of its phases into more comprehensible subsections.

**Intuition** On a high level, our algorithm's rationale can be explained as follows. First, Adaptive $\boldsymbol{\epsilon}$-PAL maintains an adaptively changing resolution over the design space, structured along so-called nodes of a tree. That is, starting from the center of the space (the center node), we only 'zoom in' (thus expanding the tree) on points that are of potential interest. In every iteration of the algorithm, we are given the current set of active nodes in our tree-based partition of the space. These nodes serve as proxies for regions of points in the design space which they inhabit. Now, the goal of the algorithm is to return a set of nodes (and, as a consequence, their associated regions) that form an approximate Pareto front with high probability. In order to do that, the algorithm maintains a confidence region for every active node. Applying worst-case arguments over these confidence regions, the algorithm decides which points to discard in every iteration, and which points to maintain as potentially optimal (in the Pareto sense). Next, we move points about which we are confident enough to a predicted Pareto set. Basically, we keep shrinking the set of points about which we are undecided, and we keep growing the set of points which we believe are approximately optimal. Finally, if enough information is obtained on the most uncertain active node, then we decide to expand it into children nodes, since there is enough reason to believe that more relevant information will be obtained from those new nodes.

The system operates in rounds $t \geq 1$. In each round $t$, the algorithm picks a design $x_t \in \mathcal{X}$, and assuming that it already had $\tau$ evaluations, it subsequently decides whether or not to obtain the $\tau + 1$st noisy observation $\boldsymbol{y}_{\tau+1} = [y_\tau^1, \ldots, y_{\tau+1}^m]^\mathsf{T}$ of the latent function $\boldsymbol{f}$ at $x_t$. At the end, our algorithm returns a subset $\hat{P}$ of $\mathcal{X}$ which is guaranteed to be an $\boldsymbol{\epsilon}$-accurate Pareto set with high probability and the associated set $\hat{\mathcal{P}}$ of nodes which we will define later. The pseudocode is given in Algorithm 1.

## 4.1 Modeling

We maintain two sets of time indices, one counting the total number of iterations, denoted by $t$, and the other counting only the evaluation rounds, denoted by $\tau$. The algorithm evaluates a design only in some rounds. For this reason, we also define the following auxiliary time variables which help us understand the chronological connection between the values of $t$ and $\tau$. We let $\tau_t$ represent the number of evaluations before round $t \geq 1$ and let $t_\tau$ denote the round when evaluation $\tau \geq 0$ is made, with the convention $t_0 = 0$. The sequence $(t_\tau)_{\tau \geq 0}$ is an increasing sequence of stopping times. Note that we have $\tau_{t_\tau+1} = \tau$ for each $\tau \in \mathbb{N}$.

Iteration over individual designs may not be feasible when the cardinality of the design space is very large. Thus, we consider partitioning the space into regions of similar designs, i.e., two designs in $\mathcal{X}$ which are at a close distance have similar outcomes in each objective. This is a natural property of GP-sampled functions as discussed in Remark 1.

Since iteration over individual designs is not possible in large spaces, we will instead iterate over 'regions' of interest. Here, we focus on a compact subset $\mathcal{X}$ of the Euclidean space $\mathbb{R}^{m'}$ for some $m' \in \mathbb{N}$. However, we do this purely for simplicity of presentation. In Appendix, we show that our analysis holds when $\mathcal{X}$ is any general 'well-behaved' metric space.[2] For such a metric space, one can easily partition the design space along a tree structure, each level $h$ of which is associated with a partition of $\mathcal{X}$ into $N^h$ equal-sized regions $X_{h,i}$, centered at a node $x_{h,i}$, for all $0 \leq i \leq N^h$, where $N \in \mathbb{N}$.

At each round $t \in \mathbb{N}$, the algorithm maintains a set $\mathcal{S}_t$ of *undecided nodes* and a set $\mathcal{P}_t$ of *decided nodes*. For an undecided node in $\mathcal{S}_t$, its associated cell consists of designs for which we are undecided about including in the $\boldsymbol{\epsilon}$-accurate Pareto set. Similarly, for a decided node in $\mathcal{P}_t$, its associated cell consists of designs that we decide to include in the $\boldsymbol{\epsilon}$-accurate Pareto set. At the beginning of round $t = 1$ (initialization), we set

---

[2]See Section A for detailed definitions.

$\mathcal{S}_1 = \{x_{0,1}\}$ and $\mathcal{P}_1 = \emptyset$. Within each round $t \in \mathbb{N}$, the sets $\mathcal{P}_t$ and $\mathcal{S}_t$ are updated during the discarding, $\boldsymbol{\epsilon}$-covering and refining/evaluating phases of the round; at the end of round $t$, their finalized contents are set as $\mathcal{P}_{t+1}$ and $\mathcal{S}_{t+1}$, respectively, as a preparation for round $t+1$. For each $t \in \mathbb{N}$, the algorithm performs round $t$ as long as $\mathcal{S}_t \neq \emptyset$ at the beginning of round $t$; otherwise, it terminates and returns $\hat{\mathcal{P}} = \mathcal{P}_t$.

In addition to the sets of undecided and decided nodes, the algorithm maintains a set $\mathcal{A}_t$ of *active nodes*, which is defined as the union $\mathcal{S}_t \cup \mathcal{P}_t$ at the beginning of each round $t \in \mathbb{N}$. While the sets $\mathcal{S}_t$ and $\mathcal{P}_t$ are updated within round $t$ as described above, the set $\mathcal{A}_t$ is kept fixed throughout the round with its initial content. Note that $\mathcal{A}_1 = \{x_{0,1}\}$.

At round $t \in \mathbb{N}$, the algorithm considers each active node $x_{h,i} \in \mathcal{A}_t$. Let $j \in [m]$. We define the *lower index* of $x_{h,i}$ in the $j$th objective as $L_t^j(x_{h,i}) = \underline{B}_t^j(x_{h,i}) - V_h$, where $\underline{B}_t^j(x_{h,i})$ is a high probability lower bound on the $j$th objective value at $x_{h,i}$ and is defined as

$$\underline{B}_t^j(x_{h,i}) = \max\{\mu_{\tau_t}^j(x_{h,i}) - \beta_{\tau_t}^{1/2}\sigma_{\tau_t}^j(x_{h,i}), \mu_{\tau_t}^j(p(x_{h,i})) - \beta_{\tau_t}^{1/2}\sigma_{\tau_t}^j(p(x_{h,i})) - V_{h-1}\} \ .$$

Here, $\beta_\tau \in O(\log(\tau^2/\delta))$ and $V_h \in \tilde{O}(\rho^{\alpha h})$ is a high probability upper bound on the maximum variation of the objective $j$ inside region $X_{h,i}$.[3] Similarly, we define the *upper index* of $x_{h,i}$ in the $j$th objective as $U_t^j(x_{h,i}) = \bar{B}_t^j(x_{h,i}) + V_h$, where $\bar{B}_t^j(x_{h,i})$ is a high probability upper bound on the $j$th objective value at $x_{h,i}$ and is defined as

$$\bar{B}_t^j(x_{h,i}) = \min\{\mu_{\tau_t}^j(x_{h,i}) + \beta_{\tau_t}^{1/2}\sigma_{\tau_t}^j(x_{h,i}), \mu_{\tau_t}^j(p(x_{h,i})) + \beta_{\tau_t}^{1/2}\sigma_{\tau_t}^j(p(x_{h,i})) + V_{h-1}\} \ .$$

We denote by $\boldsymbol{L}_t(x_{h,i}) = [L_t^1(x_{h,i}), \ldots, L_t^m(x_{h,i})]^\mathsf{T}$ the *lower index vector* of the node $x_{h,i}$ at round $t$ and similarly by $\boldsymbol{U}_t(x_{h,i}) = [U_t^1(x_{h,i}), \ldots, U_t^m(x_{h,i})]^\mathsf{T}$ the corresponding *upper index vector*. We also let $\boldsymbol{V}_h$ denote the $m$-dimensional vector with all entries being equal to $V_h$. Next, we define the *confidence hyper-rectangle* of node $x_{h,i}$ at round $t$ as

$$\boldsymbol{Q}_t(x_{h,i}) = \{\boldsymbol{y} \in \mathbb{R}^m \colon \boldsymbol{L}_t(x_{h,i}) \preceq \boldsymbol{y} \preceq \boldsymbol{U}_t(x_{h,i})\} \ ,$$

which captures the uncertainty in the learner's prediction of the objective values. Then, the posterior mean vector $\boldsymbol{\mu}_{\tau_t}(x_{h,i}) = [\mu_{\tau_t}^1(x_{h,i}), \ldots, \mu_{\tau_t}^m(x_{h,i})]^\mathsf{T}$ and the variance vector $\boldsymbol{\sigma}_{\tau_t}(x_{h,i}) = [\sigma_{\tau_t}^1(x_{h,i}), \ldots, \sigma_{\tau_t}^m(x_{h,i})]^\mathsf{T}$ are computed by using the GP inference outlined in Section 3. We define the *cumulative confidence hyper-rectangle* of $x_{h,i}$ at round $t$ as

$$\boldsymbol{R}_t(x_{h,i}) = \boldsymbol{R}_{t-1}(x_{h,i}) \cap \boldsymbol{Q}_t(x_{h,i}) \tag{1}$$

assuming that $\boldsymbol{R}_{t-1}(x_{h,i})$ is well-defined at round $t-1$ (the case $t \geq 2$) or using the convention that $\boldsymbol{R}_0(x_{0,1}) = \mathbb{R}^m$ since $\mathcal{A}_1 = \{x_{0,1}\}$ (the case $t = 1$). The well-definedness assumption will be verified in the refining/evaluating phase below.

## 4.2 Discarding phase

In order to correctly identify designs to be discarded under uncertainty, we need to compare the pessimistic and optimistic outcomes of designs. First, we define dominance under uncertainty.

**Definition 7.** *Let $t \in \mathbb{N}$ and let $x, y \in \mathcal{A}_t$ be two nodes with $x \neq y$. We say that $x$ is $\boldsymbol{\epsilon}$-dominated by $y$* ***under uncertainty*** *at round $t$ if $\max(\boldsymbol{R}_t(x)) \preceq_{\boldsymbol{\epsilon}} \min(\boldsymbol{R}_t(y))$, where we define $\max(\boldsymbol{R}_t(x))$ as the unique vector $\boldsymbol{v} \in \boldsymbol{R}_t(x)$ such that $v^j \geq z^j$ for every $j \in [m]$ and $\boldsymbol{z} = (z^1, \ldots, z^j) \in \boldsymbol{R}_t(x)$, and we define $\min(\boldsymbol{R}_t(y))$ in a similar fashion.*

If a node $x \in \mathcal{A}_t$ is $\boldsymbol{\epsilon}$-dominated by any other node in $\mathcal{A}_t$ under uncertainty, then the algorithm is confident enough to discard it. Basically, the condition of Definition 7 implies that, if the best possible value that $x$ can have is still approximately dominated by the worst possible value that $y$ can have, then we can conclude that $y$ dominates $x$ with overwhelming probability. To check this, the algorithm compares $x$ with all of the *pessimistic* available points as introduced next.

---

[3]See Theorem 1 for exact definitions.

**Definition 8.** *(Pessimistic Pareto set)* *Let $t \geq 1$ and let $D \subseteq \mathcal{A}_t$ be a set of nodes. We define $p_{pess,t}(D)$, called the **pessimistic Pareto set** of $D$ at round $t$, as the set of all nodes $x \in D$ for which there is no other node $y \in D \setminus \{x\}$ such that $\min(\boldsymbol{R}_t(x)) \prec \min(\boldsymbol{R}_t(y))$. We call a design in $p_{pess,t}(D)$ a pessimistic Pareto design of $D$ at round $t$.*

Here we are interested in finding the nodes, say $x$, which are Pareto optimal in the most pessimistic scenario when their objective values turn out to be $\min \boldsymbol{R}_t(x)$. We do this in order to identify which nodes (and their associated cells) to discard with overwhelming probability. More precisely, the algorithm calculates $\mathcal{P}_{pess,t} = p_{pess,t}(\mathcal{A}_t)$ first. For each $x_{h,i} \in \mathcal{S}_t \backslash \mathcal{P}_{pess,t}$, it checks if $\max(\boldsymbol{R}_t(x_{h,i})) \preceq_{\boldsymbol{\epsilon}} \min(\boldsymbol{R}_t(x))$ for some $x \in \mathcal{P}_{pess,t}$. In this case, node $x_{h,i}$ is discarded, that is, it is removed from $\mathcal{S}_t$, and will not be considered in the rest of the algorithm; otherwise no change is made in $\mathcal{S}_t$.

### 4.3 $\epsilon$-Covering phase

The overall aim of the learner is to empty the set $\mathcal{S}_t$ of undecided nodes as fast as possible. A node $x_{h,i} \in \mathcal{S}_t$ is moved to the decided set $\mathcal{P}_t$ if it is determined that the associated cell $X_{h,i}$ belongs to an $\boldsymbol{\epsilon}$-accurate Pareto set $\mathcal{O}_{\boldsymbol{\epsilon}}$ with high probability. To check this, the notion in the next definition is useful. Let us denote by $\mathcal{W}_t$ the union $\mathcal{P}_t \cup \mathcal{S}_t$ at the end of the discarding phase. Note that $\mathcal{W}_t \subseteq \mathcal{A}_t$ but the two sets do not coincide in general due to the discarding phase.

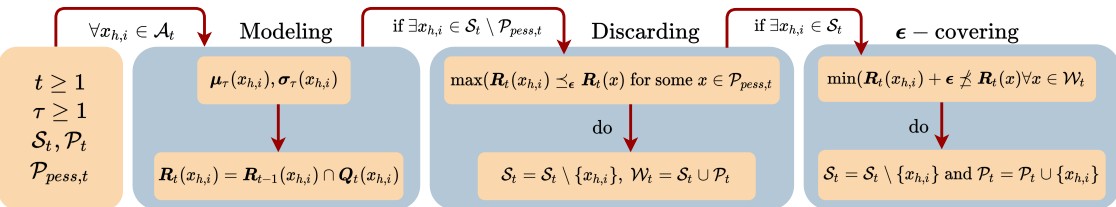

Figure 1: This is an illustration of the modeling, discarding, and $\boldsymbol{\epsilon}$-covering phases.

**Definition 9.** *Let $x_{h,i} \in \mathcal{S}_t$. We say that the cell $X_{h,i}$ associated to node $x_{h,i}$ **belongs to an $\mathcal{O}_{\boldsymbol{\epsilon}}$ with high probability** if there is no $x \in \mathcal{W}_t$ such that $\min(\boldsymbol{R}_t(x_{h,i})) + \boldsymbol{\epsilon} \preceq \max(\boldsymbol{R}_t(x))$.*

For each $x_{h,i} \in \mathcal{S}_t$, the algorithm checks if $X_{h,i}$ belongs to an $\mathcal{O}_{\boldsymbol{\epsilon}}$ with high probability in view of Definition 9. In this case, $x_{h,i}$ is removed from the set $\mathcal{S}_t$ of undecided nodes and is moved to the set $\mathcal{P}_t$ of decided nodes; otherwise, no change is made. The nodes in $\mathcal{P}_t$ are never removed from this set; hence, they will be returned by the algorithm as part of the set $\hat{\mathcal{P}}$ at termination. Intuitively, this step determines which points can be safely predicted to be in the approximately accurate Pareto set if there is no other active node which approximately dominates it in the worst case possible. In the appendix, we show that the union $\hat{P} = \bigcup_{x_{h,i} \in \hat{\mathcal{P}}} X_{h,i}$ of the cells is an $\boldsymbol{\epsilon}$-accurate Pareto cover, according to Definition 5, with high probability. Note that while the sets $\mathcal{S}_t, \mathcal{P}_t$ can be modified during this phase, the set $\mathcal{W}_t$ does not change. The modeling, discarding, and $\boldsymbol{\epsilon}$-covering phases of the algorithm are illustrated in Figure 1.

### 4.4 Refining/evaluating phase

While $\mathcal{S}_t \neq \emptyset$, the algorithm selects a design $x_t = x_{h_t, i_t} \in \mathcal{W}_t$ that corresponds to a node with depth $h_t$ and index $i_t$, according to the following rule. First, for a given node $x_{h,i} \in \mathcal{W}_t$, we define

$$\omega_t(x_{h,i}) = \max_{y, y' \in \boldsymbol{R}_t(x_{h,i})} \|y - y'\|_2 \ , \tag{2}$$

which is the diameter of its cumulative confidence hyper-rectangle in $\mathbb{R}^m$. The algorithm picks the most uncertain node for evaluation in order to decrease uncertainty. Hence, among the available points in $\mathcal{W}_t$, the node $x_{h_t, i_t}$ with the maximum such diameter is chosen by the algorithm. We denote the diameter of the cumulative confidence hyper-rectangle associated with the selected node by $\overline{\omega}_t$ and formally define it as $\overline{\omega}_t = \max_{x_{h,i} \in \mathcal{W}_t} \omega_t(x_{h,i})$. Since the learner is not sure about discarding $x_{h_t, i_t}$ or moving it to $\mathcal{P}_t$, he decides

Sampling/Evaluating

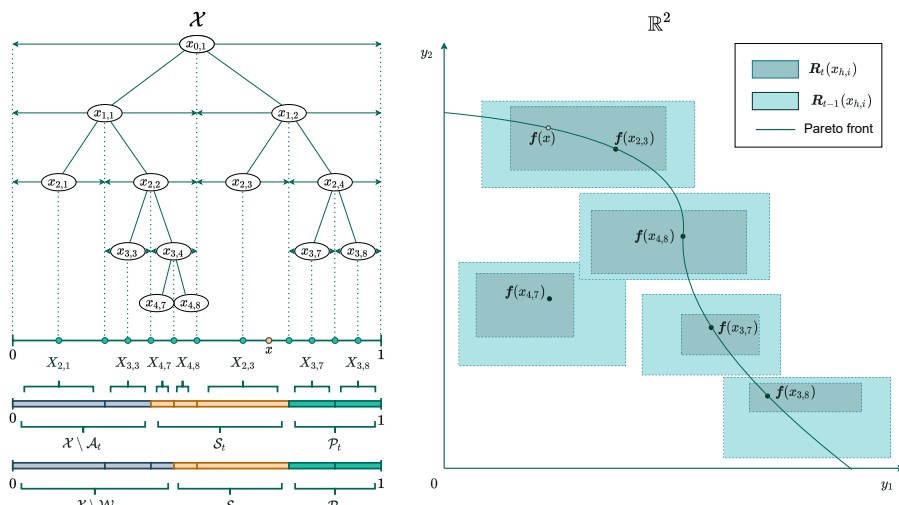

Figure 2: This is an illustration of the sampling and refining phase.

whether to refine the associated region $X_{h_t,i_t}$ or evaluating the objective function at the current node based on the following rule.

- *Refine:* If $\beta_{\tau_t}^{1/2}\|\boldsymbol{\sigma}_{\tau_t}(x_{h_t,i_t})\|_2 \leq \|\boldsymbol{V}_{h_t}\|_2$, then $x_{h_t,i_t}$ is expanded, i.e., the $N$ children nodes $\{x_{h_t+1,j}: N(i_t - 1) + 1 \leq j \leq i_t\}$ of $x_{h_t,i_t}$ are generated. If $x_{h_t,i_t} \in \mathcal{S}_t$, then these newly generated nodes are added to $\mathcal{S}_t$ while $x_{h_t,i_t}$ is removed from $\mathcal{S}_t$. An analogous operation is performed if $x_{h_t,i_t} \in \mathcal{P}_t$. In each case, for each $j$ with $N(i_t - 1) + 1 \leq j \leq i_t$, the newly generated node $x_{h_t+1,j}$ inherits the cumulative confidence hyper-rectangle of its parent node $x_{h_t,i_t} \in \mathcal{A}_t$ as calculated by equation 1, that is, we define $\boldsymbol{R}_t(x_{h_t+1,j}) = \boldsymbol{R}_t(x_{h_t,i_t}) = \boldsymbol{R}_{t-1}(x_{h_t,i_t}) \cap \boldsymbol{Q}_t(x_{h_t,i_t})$. This way, for every node $x \in \mathcal{P}_t \cup \mathcal{S}_t$ at the end of refining, the cumulative confidence hyper-rectangles up to round $t$ are well-defined and we have $\boldsymbol{R}_0(x) \supseteq \boldsymbol{R}_1(x) \supseteq \ldots \supseteq \boldsymbol{R}_t(x)$. In particular, the well-definedness assumption for equation 1 is verified for round $t+1$ since $\mathcal{A}_{t+1}$ is defined as $\mathcal{P}_t \cup \mathcal{S}_t$ at the end of this phase.
- *Evaluate:* If $\beta_{\tau_t}^{1/2}\|\boldsymbol{\sigma}_{\tau_t}(x_{h_t,i_t})\|_2 > \|\boldsymbol{V}_{h_t}\|_2$, then the objective function is evaluated at the point $x_{h_t,i_t}$, i.e., we observe the noisy sample $\boldsymbol{y}_{\tau_t}$ and update the posterior statistics of $x_{h_t,i_t}$. No change is made in $\mathcal{S}_t$ and $\mathcal{P}_t$.

This phase of the algorithm is illustrated in Figure 2. The evolution of the partitioning of the design space is illustrated in Figure 3.

Figure 3: This is an illustration of the structural way of partitioning the design space. In this example we take $\mathcal{X} = [0,1]$, $m = 2$ and $\boldsymbol{\epsilon} = \boldsymbol{0}$. On the left, we can see the partition of $\mathcal{X}$ at the beginning of round $t$. Note that $\mathcal{S}_t = \{x_{4,7}, x_{4,8}, x_{2,3}\}$ and $\mathcal{P}_t = \{x_{3,7}, x_{3,8}\}$, while $x_{2,1}$ and $x_{3,3}$ have been discarded in some prior round. On the right, we see the corresponding confidence hyper-rectangles of these nodes. At the beginning of round $t$, prior to the modeling phase, the hyper-rectangle of node $x_{h,i}$ is $\boldsymbol{R}_{t-1}(x_{h,i})$. Note that, in the discarding phase, node $x_{4,8}$ will be discarded since $\max(\boldsymbol{R}_t(x_{4,8})) \preceq_{\boldsymbol{\epsilon}} \min(\boldsymbol{R}_t(x_{4,7}))$. Thus, $x_{4,7}$ will be removed from $\mathcal{S}_t$ by the end of the phase. Furthermore, note that more than one Pareto optimal designs take values in one hyper-rectangle. This is because the node of a region containing Pareto optimal points is not necessarily one of these points. For example, we have $x \in X_{2,3}$ and $\boldsymbol{f}(x) \in \boldsymbol{R}_t(x_{2,3})$.

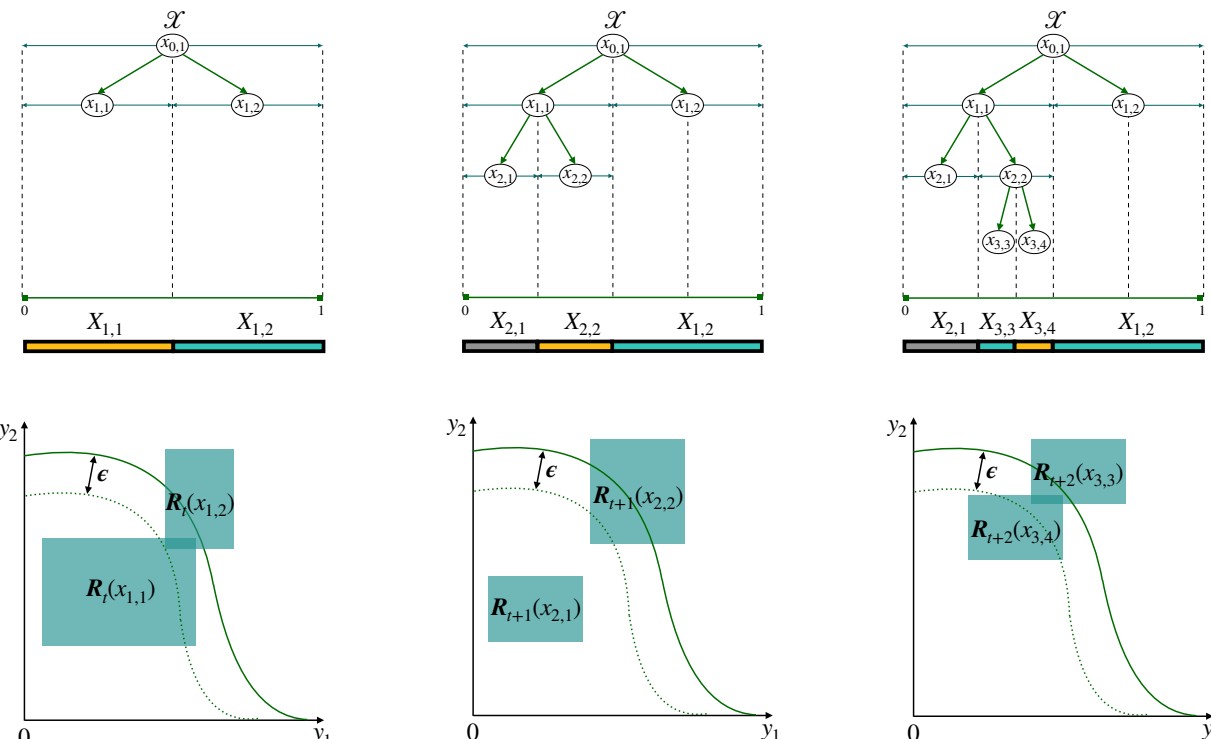

Figure 4: A depiction of three refining iterations of our algorithm together with the associated discarding and $\epsilon$-covering phases. The first row reflects the structural changes in the design space. The second row reflects the changes in the objective space. The green color in the region bars denotes a predicted Pareto region, the orange color denotes an undecided region, while the gray color a discarded region. At time $t$, we have two nodes $x_{1,1}$ and $x_{1,2}$ that partition $\mathcal{X}$. The region $X_{1,2}$ is already a predicted Pareto set, while $X_{1,1}$ is still undecided. This is also reflected in the first figure (from the left) on the second row. Here, we can visually see that $x_{1,1}$ is $\epsilon$-dominated by $x_{1,2}$ under uncertainty. Since there are no more nodes, we proceed to assign $X_{1,2}$ to a predicted Pareto set. In the next iteration, we first refine $x_{1,1}$ further because the uncertainty level is already below the threshold. Then, upon examination, we discard region $X_{2,1}$. Again, the corresponding changes in the objective space are reflected in the second row. In the next iteration, we further refine $x_{2,2}$ and include one of its children regions to the predicted Pareto set. Note that we have not included evaluation steps for simplicity of presentation.

### 4.5 Termination

If $\mathcal{S}_t = \emptyset$ at the beginning of round $t$, then the algorithm terminates. We show in the appendix that at the latest, the algorithm terminates when $\overline{\omega}_t < \min_j \epsilon^j$. Upon termination, it returns a non-empty set $\hat{\mathcal{P}}$ of decided nodes together with the corresponding $\epsilon$-accurate Pareto set $\hat{P} = \bigcup_{x_{h,i} \in \hat{\mathcal{P}}} X_{h,i}$ , which is the union of the cells corresponding to the nodes in $\hat{\mathcal{P}}$. The full procedure is given in Algorithm 1.

## 5 Sample complexity bounds

We state the main result in this section. Its proof is composed of a sequence of lemmas which are given in the appendix. We first state the necessary assumptions (Assumption 1) on the metric and the kernel under which the result holds. Then, we provide a sketch of its proof and subsequently give an example of an $m$-output GP for which the maximum information gain is linear in $T$.

---

**Algorithm 1** Adaptive $\epsilon$-PAL

---

**Input:** $\mathcal{X}$, $(\mathcal{X}_h)_{h\geq 0}$, $(V_h)_{h\geq 0}$, $\epsilon$, $\delta$, $(\beta_\tau)_{\tau\geq 1}$; GP prior $\boldsymbol{\mu}_0 = \boldsymbol{\mu}$, $\boldsymbol{k}_0 = \boldsymbol{k}$

1: **Initialize:** $\mathcal{P}_1 = \emptyset$, $\mathcal{S}_1 = \{x_{0,1}\}$; $\boldsymbol{R}_0(x_{0,1}) = \mathbb{R}^m$, $t = 1$, $\tau = 0$.
2: **while** $\mathcal{S}_t \neq \emptyset$ **do**
3:      $\mathcal{A}_t = \mathcal{P}_t \cup \mathcal{S}_t$; $\mathcal{P}_{pess,t} = p_{pess,t}(\mathcal{A}_t)$.
4:      **for** $x_{h,i} \in \mathcal{A}_t$ **do**                              $\triangleright$ Modeling
5:          Obtain $\boldsymbol{\mu}_\tau(x_{h,i})$ and $\boldsymbol{\sigma}_\tau(x_{h,i})$ by GP inference.
6:          $\boldsymbol{R}_t(x_{h,i}) = \boldsymbol{R}_{t-1}(x_{h,i}) \cap \boldsymbol{Q}_t(x_{h,i})$.
7:      **end for**
8:      **for** $x_{h,i} \in \mathcal{S}_t \setminus \mathcal{P}_{pess,t}$ **do**                     $\triangleright$ Discarding
9:          **if** $\exists x \in \mathcal{P}_{pess,t}$: $\max(\boldsymbol{R}_t(x_{h,i})) \preceq_\epsilon \min(\boldsymbol{R}_t(x))$ **then**
10:             $\mathcal{S}_t = \mathcal{S}_t \setminus \{x_{h,i}\}$.
11:          **end if**
12:      **end for**
13:      $\mathcal{W}_t = \mathcal{S}_t \cup \mathcal{P}_t$.
14:      **for** $x_{h,i} \in \mathcal{S}_t$ **do**                                 $\triangleright$ $\epsilon$-Covering
15:          **if** $\nexists x \in \mathcal{W}_t$: $\min(\boldsymbol{R}_t(x_{h,i})) + \epsilon \preceq \max(\boldsymbol{R}_t(x))$ **then**
16:             $\mathcal{S}_t = \mathcal{S}_t \setminus \{x_{h,i}\}$ ; $\mathcal{P}_t = \mathcal{P}_t \cup \{x_{h,i}\}$.
17:          **end if**
18:      **end for**
19:      **if** $\mathcal{S}_t \neq \emptyset$ **then**                           $\triangleright$ Refining/Evaluating
20:          Select node $x_{h_t,i_t} = \operatorname{argmax}_{x_{h,i} \in \mathcal{W}_t} \omega_t(x_{h,i})$.
21:          **if** $\beta_\tau^{1/2} \|\boldsymbol{\sigma}_\tau(x_{h_t,i_t})\|_2 \leq \|V_{h_t}\|_2$ AND $x_{h_t,i_t} \in \mathcal{S}_t$ **then**
22:             $\mathcal{S}_t = \mathcal{S}_t \setminus \{x_{h_t,i_t}\}$; $\mathcal{S}_t = \mathcal{S}_t \cup \{x_{h_t+1,i}: N(i_t - 1) + 1 \leq i \leq Ni_t\}$.
23:             $\boldsymbol{R}_t(x_{h_t+1,i}) = \boldsymbol{R}_t(x_{h_t,i_t})$ for each $i$ with $N(i_t - 1) + 1 \leq i \leq Ni_t$.
24:          **else if** $\beta_\tau^{1/2} \|\boldsymbol{\sigma}_\tau(x_{h_t,i_t})\|_2 \leq \|V_{h_t}\|_2$ AND $x_{h_t,i_t} \in \mathcal{P}_t$ **then**
25:             $\mathcal{P}_t = \mathcal{P}_t \setminus \{x_{h_t,i_t}\}$; $\mathcal{P}_t = \mathcal{P}_t \cup \{x_{h_t+1,i}: N(i_t - 1) + 1 \leq i \leq Ni_t\}$.
26:             $\boldsymbol{R}_t(x_{h_t+1,i}) = \boldsymbol{R}_t(x_{h_t,i_t})$ for each $i$ with $N(i_t - 1) + 1 \leq i \leq Ni_t$.
27:          **else**
28:             Evaluate design $x_t = x_{h_t,i_t}$ and observe $\boldsymbol{y}_\tau = \boldsymbol{f}(x_{h_t,i_t}) + \boldsymbol{\kappa}_\tau$.
29:             $\tau = \tau + 1$.
30:          **end if**
31:      **end if**
32:      $\mathcal{P}_{t+1} = \mathcal{P}_t$; $\mathcal{S}_{t+1} = \mathcal{S}_t$.
33:      $t = t + 1$.
34: **end while**
35: **return** $\hat{\mathcal{P}} = \mathcal{P}_t$ and $\hat{P} = \bigcup_{x_{h,i} \in \mathcal{P}_t} X_{h,i}$.

---

**Assumption 1.** *The class $\mathcal{K}$ of covariance functions to which we restrict our focus satisfies the following criteria for any $\boldsymbol{k} \in \mathcal{K}$: (1) For any $x, y \in \mathcal{X}$ and $j \in [m]$, we have $l_j(x,y) \leq C_{\boldsymbol{k}} d(x,y)^\alpha$, for suitable $C_{\boldsymbol{k}} > 0$ and $0 < \alpha \leq 1$. Here, $l_j$ is the natural metric induced on $\mathcal{X}$ by the $j$th component of the GP in Definition 6 as given in Remark 1 with covariance function $k^{jj}$. (2) We assume bounded variance, that is, for any $x \in \mathcal{X}$ and $j \in [m]$, we have $k^{jj}(x,x) \leq 1$.*

**Theorem 1.** *Let $\boldsymbol{\epsilon} = [\epsilon^1, \ldots, \epsilon^m]^{\mathsf{T}}$ be given with $\epsilon = \min_{j \in [m]} \epsilon^j > 0$. Let $\delta \in (0,1)$ and $\bar{D} > D_1$. For each $h \geq 0$, let*

$$
V_h = 4C_{\boldsymbol{k}}(v_1 \rho^h)^\alpha \left( \sqrt{C_2 + 2\log\left(\frac{2h^2 \pi^2 m}{6\delta}\right)} + h\log N + \left(\frac{-4D_1}{\alpha} \log\left(C_{\boldsymbol{k}}(v_1\rho^h)^\alpha\right)\right)^+ + C_3 \right) ,
$$

*for some strictly positive constants $C_2$ and $C_3$, where $x^+ := \max\{0, x\}$ for $x \in \mathbb{R}$. Moreover, for each $\tau \in \mathbb{N}$, define $\beta_\tau = 2\log(2m\pi^2 N^{h_{max}+1}(\tau+1)^2/(3\delta))$. When we run Adaptive $\boldsymbol{\epsilon}$-PAL with prior $GP(0, \boldsymbol{k})$ and noise $\mathcal{N}(0, \sigma^2)$, the following holds with probability at least $1 - \delta$:*

*An $\boldsymbol{\epsilon}$-accurate Pareto set can be found with at most $T$ function evaluations, where $T$ is the smallest natural number satisfying*

$$
\min\left\{ K_1 \beta_T T^{\frac{-\alpha}{\bar{D}+2\alpha}} (\log T)^{\frac{-(\bar{D}+\alpha)}{\bar{D}+2\alpha}} + K_2 T^{\frac{-\alpha}{\bar{D}+2\alpha}} (\log T)^{\frac{\alpha}{\bar{D}+2\alpha}}, \sqrt{\frac{C\beta_T \gamma_T}{T}} \right\} < \epsilon ,
$$

*where $C$ and $K_1$ are constants that are defined in the appendix and do not depend on $T$, $K_2$ is logarithmic in $T$, and $\gamma_T$ is the maximum information gain which depends on the choice of $\boldsymbol{k}$.*

*Sketch of the proof of Theorem 1:* We divide the proof of Theorem 1 into three essential parts. First, we prove that the algorithm terminates in finite time, and examine the events that necessitate and the ones that follow termination. Here, it is important to note that if the largest uncertainty diameter in a given round $t$ is less than or equal to $\epsilon$, then the algorithm terminates. This introduces a way on how to proceed on upper bounding sample complexity, namely, we can upper bound the sum of these uncertainty diameters over all rounds. Then, by observing that i) the term $V_h$ decays to 0 as $h$ grows indefinitely, which means that the algorithm refines up until a finite number of tree levels, and ii) a node cannot be refined more than a finite number of times before expansion, we can conclude that the algorithm terminates in finite time. After this point we prove, using Hoeffding bounds, that the true function values live inside the uncertainty hyper-rectangles with high probability. We then proceed on first proving the dimension-type sample complexity bounds and then the information-type bounds. We use the aforementioned termination condition and upper bound the sum of the uncertainty diameters over all rounds. We use Cauchy-Schwarz inequality to manipulate the expression as we desire and then upper bound it in terms of information gain using an already established result in the paper. For the dimension-type bounds, we consider expressing the sum over rounds as a sum over levels $h$ of the tree of partitions and upper bound it using the notion of metric dimension. We take the minimum of the two bounds, thus achieving the bound stated in Theorem 1.

**Remark 2.** *Note that we minimize over two different bounds in Theorem 1. The term that involves $\gamma_T$ corresponds to the information-type bound, while the other term corresponds to the metric dimension-type bound. Equivalently, we can express our information-type bound as $\tilde{O}(g(\epsilon))$ where $g(\epsilon) = \min\{T \geq 1 : \sqrt{\gamma_T/T} < \epsilon\}$ and our metric dimension-type bound as $\tilde{O}(\epsilon^{-(\frac{\bar{D}}{\alpha}+2)})$ for any $\bar{D} > D_1$. For certain kernels, such as squared exponential and Matérn kernels, $\gamma_T$ can be upper bounded by a sublinear function of $T$ (see Srinivas et al. (2012)). Our information type-bound is of the same form as in Zuluaga et al. (2016). When $\mathcal{X}$ is a finite subset of the Euclidean space, we have $D_1 = 0$, and thus, our metric-dimension type bound becomes near-$O(1/\epsilon^2)$, which is along the same lines with the almost optimal, gap-dependent near-$O(1/gap^2)$ bound for Pareto front identification in Auer et al. (2016).*

In order to prove the bounds in Theorem 1, even for infinite $\mathcal{X}$, we propose a novel way of defining the confidence hyper-rectangles and refining them. Since Adaptive $\boldsymbol{\epsilon}$-PAL discards, $\boldsymbol{\epsilon}$-covers and refines/evaluates in ways different than $\boldsymbol{\epsilon}$-PAL in Zuluaga et al. (2016), we use different arguments in the proof to show when the algorithm converges and what it returns when it converges. In particular, for the information-type bound,

we exploit the dependence structure between the objectives. Moreover, having two different bounds allows us to use the best of both, as it is known that for certain kernels, the metric dimension-type bound can be tighter than the information-type bound. That is the implication of our next result. The proof is mainly technical, thus we defer it to the Appendix.

**Proposition 1.** *There exists a multi-output GP $\boldsymbol{f}$, with covariance function satisfying Assumption 1 and a sequence of $T$ noisy observations made on $\boldsymbol{f}$, such that we have $I(\boldsymbol{y}_{[T]}, \boldsymbol{f}_{[T]}) \geq \Omega(T)$.*

# 6 Computational complexity analysis

The most significant subroutines of Adaptive $\boldsymbol{\epsilon}$-PAL in terms of computational complexity are the modeling, discarding, and $\boldsymbol{\epsilon}$-covering phases. Below, we inspect the computational complexity of these three phases separately.

**Modeling.** In the modeling phase, mean and variance values of the GP surrogate are computed for every node point in the leaf set. This results in a complexity of $\mathcal{O}(\tau^3 + n\tau^2)$, where $\tau$ is the number of evaluations and $n$ is the number of node points at a particular round. Note that the bound on $n$ depends on the maximum depth of the search tree.

**Discarding.** This phase can be further separated into two substeps. In the first substep, the pessimistic Pareto front is determined by choosing the leaf nodes whose lower confidence bounds are not dominated by any other point in the leaf set. In the second phase, the leaf points whose upper confidence bounds are $\boldsymbol{\epsilon}$-dominated by pessimistic Pareto set points are discarded. If done naively by comparing each point with all the other points, both of the phases can result in $\mathcal{O}(n^2)$ complexity. However, there are efficient methods that achieve lower computational complexity. In our implementation, we adopt Algorithm 3.1 of Kung et al. (1975) to reduce computational complexity to $\mathcal{O}(n \log n)$ when $m = 2$ or $m = 3$. For $m > 3$, one can use Algorithm 4.1 of Kung et al. (1975) to achieve a $\mathcal{O}(n(\log n)^{m-2})$ complexity.

**$\boldsymbol{\epsilon}$-Covering.** In the $\boldsymbol{\epsilon}$-covering phase, the algorithm moves the nodes that are not dominated by any other node to $\hat{\mathcal{P}}$. Similar to the discarding phase, when implemented naively, the $\boldsymbol{\epsilon}$-covering phase results in $\mathcal{O}(n^2)$ complexity. An adaptation of Kung et al. (1975) algorithm can be implemented as in the case of discarding phase to achieve sub-quadratic complexity which results in $\mathcal{O}(n \log n)$ for $m = 2$ and $m = 3$ and $\mathcal{O}(n(\log n)^{m-2} + n \log n)$ for $m > 3$.

In general, we observed that the number of evaluations was much smaller than the number of nodes throughout a run. Therefore we can say that discarding and $\epsilon$-covering phases are the main bottlenecks in our implementation which scale sub-quadratically with the number of points.

# 7 Experiments

This section empirically evaluates Adaptive $\epsilon$-PAL, comparing its performance and efficiency against other multi-objective Bayesian optimization (MOBO) methods. We focus on validating the effectiveness of the adaptive discretization strategy and assessing the algorithm's ability to find an $\boldsymbol{\epsilon}$-accurate Pareto set sample-efficiently.

## 7.1 Performance metrics

We use a combination of three performance metrics: $\epsilon$-accuracy ratio, $\epsilon$-coverage ratio, and average mean-squared error (MSE). Since the objective functions tested are continuous, the true Pareto front contains infinitely many points. To compute the metrics, we approximate the true Pareto front by sampling $10,000$ points uniformly from the input space, evaluating the objective function at these points, and identifying the Pareto optimal designs among them.

**$\boldsymbol{\epsilon}$-accuracy ratio.** This metric measures the quality of the predicted Pareto set $\hat{P}$. It is computed as the ratio of predicted points $\boldsymbol{f}(x)$ (with $x \in \hat{P}$) that fall within the $\boldsymbol{\epsilon}$-Pareto front $\mathcal{Z}_{\boldsymbol{\epsilon}}(\mathcal{X})$ to the total number of points in $\hat{P}$. Operationally, we check if a predicted point is $\boldsymbol{\epsilon}$-dominated by some point on the approximated true Pareto front, or equivalently, if it is at most $2\boldsymbol{\epsilon}$ away (in the sense of $\preceq_{2\boldsymbol{\epsilon}}$) from the closest true Pareto

point. While high $\epsilon$-accuracy is desirable, it does not guarantee that the entire Pareto front is well-represented. We also need to assess how well the predicted set covers the true front, which is measured by the $\epsilon$-coverage metric.

**$\epsilon$-coverage ratio.** This metric measures how well the predicted set $\hat{P}$ covers the true Pareto front. It is computed as the ratio of true Pareto points (from the discretized approximation) that are $\epsilon$-dominated by at least one point in $\boldsymbol{f}(\hat{P})$ to the total number of true Pareto points. Equivalently, we check if a true Pareto point is within $2\epsilon$ (in the sense of $\preceq_{2\epsilon}$) of the closest predicted point.

**Average mean-squared error (MSE).** This metric complements $\epsilon$-coverage by quantifying the average closeness of the predicted front to the true front. It is computed by averaging the squared Euclidean distance between each true Pareto point (in the objective space) and the closest predicted Pareto point in $\boldsymbol{f}(\hat{P})$. Ideally, we seek high values for both accuracy and coverage, and a low value for MSE.

## 7.2 Multi-objective Bayesian optimization algorithms

We compare Adaptive -PAL with four state-of-the-art MOBO methods: PESMO (Hernández-Lobato et al., 2016), ParEGO (Knowles, 2006), USeMO (Belakaria et al., 2020), and qNEHVI (Daulton et al., 2021). The latter is a hypervolume-based method specifically designed to handle noisy observations by integrating over the posterior uncertainty of the true Pareto front. Since PESMO, ParEGO, USeMO, and qNEHVI require a pre-specified evaluation budget, while Adaptive -PAL terminates based on its confidence criteria, we set the budget for the competitors to be at least as large as the number of evaluations performed by Adaptive -PAL upon termination in our experiments. This ensures competitors have access to at least as much information. We used the implementations for PESMO and ParEGO from the Spearmint Bayesian optimization library[4], for USeMO from the authors' open-source code[5], and for qNEHVI from the BoTorch library.[6]

For completeness, we also attempted to benchmark against the original $\epsilon$-PAL algorithm (Zuluaga et al., 2016). However, we encountered difficulties achieving termination when running both the authors' publicly available code and our own implementation based on the published pseudocode. As we were unable to obtain completed runs, we have excluded $\epsilon$-PAL from the presented results.

## 7.3 Simulation setup & results

We first sampled 10 distinct objective functions from the specified GP prior (detailed below). For each of these functions, we then ran each algorithm 5 times, using different random seeds for each run to vary the observation noise and any internal randomness within the algorithms.

**Setup.** We simulate a problem with a 1-dimensional input space $\mathcal{X} = [0, 1]$ and a 2-dimensional objective space ($m = 2$). The objective function $\boldsymbol{f}$ is sampled from a GP with a zero mean function and independent squared exponential kernels for each objective: $k^1(x, x') = 0.5 \exp(-\frac{(x-x')^2}{2 \times 0.1^2})$ and $k^2(x, x') = 0.1 \exp(-\frac{(x-x')^2}{2 \times 0.06^2})$. Observation noise is Gaussian with $\sigma = 0.01$ (variance $\sigma^2 = 10^{-4}$).

**Adaptive $\epsilon$-PAL.** We run Adaptive $\epsilon$-PAL with a target accuracy $\epsilon = (0.05, 0.05)$ and confidence $\delta = 0.05$. The tree parameters were set to $N = 2$, $\rho = 1/2$, and $v_1 = v_2 = 1$. To investigate the impact of the maximum tree depth, we tested three settings for $h_{\max}$. The first setting used the theoretically derived value $h_{\max} = 24$, calculated using Lemma 4 based on the target $\epsilon$. The other two settings used practical, reduced depths of $h_{\max} = 10$ and $h_{\max} = 9$ to evaluate the trade-off between computational cost and performance. When using these practical depth limits, refinement beyond $h_{\max}$ was prevented by setting $V_h = 0$ for all $h \geq h_{\max}$. In this specific experimental run, Adaptive $\epsilon$-PAL terminated after approximately 50 evaluations for $h_{\max} = 24$, 40 evaluations for $h_{\max} = 10$, and 35 evaluations for $h_{\max} = 9$.

**PESMO and ParEGO.** We run PESMO and ParEGO for 100 iterations (exceeding Adaptive $\epsilon$-PAL's evaluations). We use the same $\delta = 0.05$ where applicable. The acquisition function optimization starts from the best point on a grid of size 1000, followed by L-BFGS optimization.

---

[4]https://github.com/HIPS/Spearmint/tree/PESM
[5]https://github.com/belakaria/USeMO
[6]https://botorch.org/docs/multi_objective

Table 2: Average of the $\epsilon$-accuracy and coverage ratios (as percentages, mean $\pm$ 99%-confidence interval) for different evaluation thresholds $\boldsymbol{\epsilon'} = (\epsilon', \epsilon')$. Highest mean value in each row is bolded. Target accuracy for Adaptive $\boldsymbol{\epsilon}$-PAL was $\boldsymbol{\epsilon} = (0.05, 0.05)$.

| $\epsilon'$ | Adaptive $\epsilon$-PAL ($h_{\max} = 24$) | Adaptive $\epsilon$-PAL ($h_{\max} = 10$) | Adaptive $\epsilon$-PAL ($h_{\max} = 9$) | PESMO | ParEGO | USeMO | qNEHVI |
|---|---|---|---|---|---|---|---|
| 0.050 | $98 \pm 2$ | $\mathbf{99 \pm 1}$ | $\mathbf{99 \pm 1}$ | $92 \pm 2$ | $94 \pm 1$ | $92 \pm 2$ | $\mathbf{99 \pm 1}$ |
| 0.010 | $97 \pm 2$ | $\mathbf{98 \pm 2}$ | $97 \pm 1$ | $90 \pm 3$ | $94 \pm 1$ | $74 \pm 4$ | $96 \pm 1$ |
| 0.005 | $\mathbf{97 \pm 1}$ | $\mathbf{97 \pm 1}$ | $90 \pm 3$ | $90 \pm 3$ | $84 \pm 4$ | $60 \pm 5$ | $95 \pm 2$ |
| 0.001 | $\mathbf{78 \pm 4}$ | $64 \pm 5$ | $42 \pm 5$ | $54 \pm 5$ | $56 \pm 5$ | $26 \pm 4$ | $60 \pm 4$ |

Table 3: Average mean-squared error (MSE) and total running time (hh:mm:ss) of the algorithms. Lowest MSE and runtime are bolded.

| Metric | Adaptive $\epsilon$-PAL ($h_{\max} = 24$) | Adaptive $\epsilon$-PAL ($h_{\max} = 10$) | Adaptive $\epsilon$-PAL ($h_{\max} = 9$) | PESMO | ParEGO | USeMO | qNEHVI |
|---|---|---|---|---|---|---|---|
| MSE ($\times 10^{-6}$) | $\mathbf{5}$ | 8 | 40 | 20 | 30 | 4500 | 10 |
| Runtime | 15:15:24 | 00:01:00 | $\mathbf{00:00:27}$ | 00:09:20 | 00:07:30 | 00:23:40 | 00:00:40 |

**USeMO.** We run USeMO for 100 iterations. Expected Improvement (EI) is used as the acquisition function, aggregated using the Tchebycheff scalarization as recommended by the authors.

**qNEHVI.** We run the Noisy Expected Hypervolume Improvement (qNEHVI) algorithm for 100 iterations. We use the sequential setting (batch size $q = 1$) and the official implementation in BoTorch. The acquisition function is optimized using multi-start L-BFGS-B, consistent with the other baselines.

**Results.** We evaluate the predicted Pareto front points returned by each algorithm using the metrics defined above. Table 2 shows the average of the $\boldsymbol{\epsilon}$-accuracy and $\boldsymbol{\epsilon}$-coverage ratios, calculated for different evaluation thresholds $\boldsymbol{\epsilon'}$. Using smaller $\boldsymbol{\epsilon'}$ values provides a stricter assessment.

Adaptive $\boldsymbol{\epsilon}$-PAL with its theoretical parameters ($h_{\max} = 24$) achieves high performance (99% average accuracy/coverage) when evaluated at its target $\boldsymbol{\epsilon'} = (0.05, 0.05)$. This aligns well with the theoretical expectation of producing a valid $\boldsymbol{\epsilon}$-accurate Pareto set with high probability. The slight deviation from 100% can be attributed to the probabilistic nature of the guarantees ($\delta = 0.05$) and the approximations involved in discretizing the true Pareto front for metric calculation. Notably, reducing $h_{\max}$ to 10 or 9 maintains excellent performance at the target $\epsilon' = 0.05$. However, as expected, using a smaller $h_{\max}$ affects the performance at stricter evaluation thresholds ($\epsilon' < 0.05$), as the algorithm has less resolution to precisely delineate the Pareto front.

Comparing with competitors, Adaptive $\boldsymbol{\epsilon}$-PAL (even with $h_{\max} = 10$) achieves superior combined accuracy and coverage across most evaluation thresholds. qNEHVI matches the best performance at $\epsilon' = 0.05$ and significantly outperforms all other baselines at stricter thresholds. Nonetheless, Adaptive $\boldsymbol{\epsilon}$-PAL with its theoretical parameters ($h_{\max} = 24$) still achieves the highest accuracy at the most stringent thresholds of $\epsilon' \leq 0.005$. USeMO shows good accuracy for $\epsilon' = 0.05$ but degrades quickly, likely due to returning a sparse set of points. PESMO and ParEGO offer reasonable performance but are consistently outperformed by both qNEHVI and Adaptive $\boldsymbol{\epsilon}$-PAL.

Table 3 presents the average MSE and total running time. The MSE results corroborate the accuracy/coverage findings: Adaptive $\boldsymbol{\epsilon}$-PAL with $h_{\max} = 24$ achieves the lowest MSE, indicating its predicted front is closest to the true front on average. The qNEHVI and Adaptive $\boldsymbol{\epsilon}$-PAL ($h_{\max} = 10$) runs yield the next-best MSE values, significantly outperforming other competitors. USeMO has a notably high MSE, consistent with its poor coverage. Observing the running times reveals the practical benefit of tuning $h_{\max}$. Reducing $h_{\max}$ from 24 to 10 or 9 decreased the runtime dramatically while preserving strong performance. qNEHVI is also

highly efficient, with a runtime of just 40 seconds, faster than all baselines except the highly-tuned Adaptive $\epsilon$-PAL ($h_{\max} = 9$). This demonstrates that Adaptive $\epsilon$-PAL can be made exceptionally fast by selecting a practical tree depth limit, while still offering an accuracy-advantage over state-of-the-art methods.

Observing the running times reveals the practical benefit of tuning $h_{\max}$. Reducing $h_{\max}$ from 24 to 10 decreased the runtime dramatically (by over 900 times in this instance) while preserving performance at the target $\epsilon = 0.05$. This demonstrates that Adaptive $\epsilon$-PAL can be made computationally efficient by selecting a practical tree depth limit, while still offering a significant accuracy-advantage over the baseline methods tested.

## 8 Conclusion

In this paper, we proposed a new algorithm for PAL in large design spaces. Our algorithm learns an $\epsilon$-accurate Pareto set of designs in as few evaluations as possible by combining an adaptive discretization strategy with GP inference of the objective values. We proved both information-type and metric-dimension type bounds on the sample complexity of our algorithm. To the best of our knowledge, this is the first sample complexity result for PAL that (i) involves an information gain term, which captures the dependence between objectives and (ii) explains how sample complexity depends on the metric dimension of the design space.

## Acknowledgments

This work was supported by the Scientific and Technological Research Council of Türkiye (TÜBİTAK) under Grant 215E342. The work of C. Tekin was supported by the BAGEP Award of the Science Academy; by the Turkish Academy of Sciences Distinguished Young Scientist Award Program (TÜBA-GEBİP-2023); by TÜBİTAK 2024 Incentive Award.

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

# Appendix

## Table of Contents

## A   Structure and dimensionality of the design space

Before we initiate our theoretical analysis, we first provide some technical definitions to be used throughout. We begin by defining well-behaved metric spaces. Our results will hold under any well-behaved metric space.

**Definition 10.** *(Well-behaved metric space (Bubeck et al., 2011)) The compact metric space $(\mathcal{X}, d)$ is said to be **well-behaved** if there exists a sequence $(\mathcal{X}_h)_{h \geq 0}$ of subsets of $\mathcal{X}$ satisfying the following properties:*

1. *There exists $N \in \mathbb{N}$ such that for each $h \geq 0$, the set $\mathcal{X}_h$ has $N^h$ elements. We write $\mathcal{X}_h = \{x_{h,i} : 1 \leq i \leq N^h\}$ and to each element $x_{h,i}$ is associated a cell $X_{h,i} = \{x \in \mathcal{X} : \forall j \neq i : d(x, x_{h,i}) \leq d(x, x_{h,j})\}$.*
2. *For all $h \geq 0$ and $1 \leq i \leq N^h$, we have $X_{h,i} = \bigcup_{j=N(i-1)+1}^{Ni} X_{h+1,j}$. The nodes $x_{h+1,j}$ for $N(i-1)+1 \leq j \leq Ni$ are called the children of $x_{h,i}$, which in turn is referred to as the parent of these nodes. We write $p(x_{h+1,j}) = x_{h,i}$ for every $N(i-1)+1 \leq j \leq Ni$.*
3. *We assume that the cells have geometrically decaying radii, i.e., there exist $0 < \rho < 1$ and $0 < v_2 \leq 1 \leq v_1$ such that we have $B(x_{h,i}, v_2 \rho^h) \subseteq X_{h,i} \subseteq B(x_{h,i}, v_1 \rho^h)$ for every $h \geq 0$. Note that we have $2v_2 \rho^h \leq \mathrm{diam}(X_{h,i}) \leq 2v_1 \rho^h$, where $\mathrm{diam}(X_{h,i}) = \sup_{x,y \in X_{h,i}} d(x, y)$.*

The first property implies that, for every $h \geq 0$, the cells $X_{h,i}$, $1 \leq i \leq N^h$ partition $\mathcal{X}$. This can be observed trivially by *reductio ad absurdum*. The second property intuitively means that, as $h$ grows, we get a more refined partition. The third property implies that the nodes $x_{h,i}$ are evenly spread out in the space.

Additionally, we make use of a notion of dimensionality intrinsic to the design space, namely, the *metric dimension*.

**Definition 11.** *(Packing, Covering and Metric Dimension (Shekhar et al., 2018)) Let $r \geq 0$.*

- *A subset $\mathcal{X}_1$ of $\mathcal{X}$ is called an $r$-**packing** of $\mathcal{X}$ if for every $x, y \in \mathcal{X}_1$ such that $x \neq y$, we have $d(x, y) > r$. The largest cardinality of such a set is called the $r$-packing number of $\mathcal{X}$ with respect to $d$, and is denoted by $M(\mathcal{X}, r, d)$.*

- *A subset $\mathcal{X}_2$ of $\mathcal{X}$ is called an $r$-**covering**[7] of $\mathcal{X}$ if for every $x \in \mathcal{X}$, there exists $y \in \mathcal{X}_2$ such that $d(x, y) \leq r$. The smallest cardinality of such a set is called the $r$-covering number of $\mathcal{X}$ with respect to $d$, and is denoted by $N(\mathcal{X}, r, d)$.*
- *The **metric dimension** $D_1$ of $(\mathcal{X}, d)$ is defined as $D_1 = \inf\{a \geq 0 \colon \exists C \geq 0, \forall r > 0 \colon \log(N(\mathcal{X}, r, d)) \leq C - a \log(r)\}$.*

This dimension coincides with the usual dimension of the space when $\mathcal{X}$ is a subspace of a finite-dimensional Euclidean space. We will upper bound the sample complexity of the algorithm using the metric dimension of $(\mathcal{X}, d)$.

## B   The proof of Theorem 1

The proof of Theorem 1 is composed of a series of sophisticated steps, and is divided into multiple subsections. First, we describe in Section B.1 a set of preliminary results that will be utilized in obtaining dimension-type and information-type bounds on the sample complexity. Then, in Section B.2, we prove a key result that provides a sufficient condition for the termination of Adaptive $\epsilon$-PAL. We also show in this section that Adaptive $\epsilon$-PAL returns an $\epsilon$-accurate Pareto set when it terminates and bounds the maximum depth node that can be created by the algorithm before it terminates. In the proof, $\tau_s$ represents the number of evaluations performed by the algorithm until termination and $t_s$ represents the round (iteration) at which the algorithm terminates. Throughout the proof, for a given $\boldsymbol{\epsilon}$, we let $\epsilon = \min_j \epsilon^j$, and assume that $\boldsymbol{\epsilon}$ is such that $\epsilon > 0$. Unless noted otherwise, all inequalities that involve random variables hold with probability one.

### B.1   Preliminary results

We start by formulating the relationship between packing number, covering number, and metric dimension, which will help us obtain dimension-type bounds on the sample complexity. Recall that $(\mathcal{X}, d)$ is a compact well-behaved metric space with metric dimension $D_1 < +\infty$.

**Lemma 1.** *For every constant $r > 0$, we have*

$$M(\mathcal{X}, 2r, d) \leq N(\mathcal{X}, r, d).$$

*Moreover, for every $\bar{D} > D_1$, there exists $Q > 0$ such that*

$$M(\mathcal{X}, 2r, d) \leq N(\mathcal{X}, r, d) \leq Q r^{-\bar{D}}.$$

*Proof.* We argue by contradiction. Suppose that we have a $2r$-packing $\{x_1, \ldots, x_M\}$ and an $r$-covering $\{y_1, \ldots, y_M\}$ of $\mathcal{X}$ such that $M \geq N + 1$. Then, by the pigeon-hole principle, we must have that both $x_i$ and $x_j$ lie in the same ball $B(y_k, r)$, for some $i \neq j$ and some $k$, meaning that $d(x_i, x_j) \leq r$, which contradicts the definition of $r$-packing. Thus, the size of any $2r$-packing is less than or equal to the size of any $r$-covering and the first claim of the lemma follows. The second claim is an immediate consequence of Definition 11 and the fact that $D_1 < +\infty$. ∎

Our next result gives a relation between the metric dimension of $\mathcal{X}$ with respect to $d$ and the one of $\mathcal{X}$ with respect to the metrics induced by the GP, which holds under Assumption 1.

**Lemma 2.** *Part 1 of Assumption 1 implies that if $(\mathcal{X}, d)$ has a finite metric dimension $D_1$, then $(\mathcal{X}, l_j)$ has a metric dimension $D_1^j$ such that $D_1^j \leq D_1/\alpha$.*

*Proof.* We will proceed in two steps.

We first claim that $N(\mathcal{X}, r, l_j) \leq N(\mathcal{X}, (\frac{r}{C_{\boldsymbol{k}}})^{\frac{1}{\alpha}}, d)$. In order to show this, let $\tilde{X}$ be an $(\frac{r}{C_{\boldsymbol{k}}})^{\frac{1}{\alpha}}$-covering of $(\mathcal{X}, d)$. Then, it is an $r$-covering of $(\mathcal{X}, l_j)$. Indeed, let $x \in \mathcal{X}$. Then, there exists $y \in \tilde{X}$, such that

$$d(x, y) \leq \left(\frac{r}{C_K}\right)^{\frac{1}{\alpha}} .$$

---

[7]Not to be confused with $\boldsymbol{\epsilon}$-covering in Definition 4, where $\boldsymbol{\epsilon} \in \mathbb{R}_+^m$. The meaning will be clear from the context.

Table 4: *Notation.*

| Symbol | Description |
|---|---|
| $\mathcal{X}$ | The design space |
| $\boldsymbol{f}$ | The latent function drawn from an $m$-output GP |
| $\boldsymbol{\epsilon}$ | Accuracy level given as input to the algorithm |
| $\mathcal{Z}(\mathcal{X})$ | The Pareto front of $\mathcal{X}$ |
| $\mathcal{Z}_{\boldsymbol{\epsilon}}(\mathcal{X})$ | The $\boldsymbol{\epsilon}$-Pareto front of $\mathcal{X}$ |
| $\boldsymbol{y}_\tau = \boldsymbol{f}(\tilde{x}_\tau) + \boldsymbol{\kappa}_\tau$ | $\tau$th noisy observation of $\boldsymbol{f}$ |
| $\boldsymbol{y}_{[\tau]}$ | Vector that represents the first $\tau$ noisy observations |
| $x_{h,i}$ | The node with index $i$ in depth $h$ of the tree |
| $X_{h,i}$ | The cell associated with node $x_{h,i}$ |
| $p(x_{h,i})$ | Parent of node $x_{h,i}$ |
| $\boldsymbol{\mu}_\tau(x_{h,i})$ | The posterior mean after $\tau$ evaluations of $x_{h,i}$ with $j$th component $\mu_\tau^j(x_{h,i})$ |
| $\boldsymbol{\sigma}_\tau(x_{h,i})$ | The posterior variance after $\tau$ evaluations of $x_{h,i}$ with $j$th component $\sigma_\tau^j(x_{h,i})$ |
| $\beta_\tau$ | The confidence term |
| $\boldsymbol{k}$ | The covariance function of the GP |
| $d$ | The metric associated with the design space |
| $l_j$ | The metric on $\mathcal{X}$ induced by the $j$th component of the GP |
| $\mathcal{P}_t$ and $P_t$ | The predicted $\boldsymbol{\epsilon}$-accurate Pareto sets of nodes and regions, respectively, at round $t$ |
| $\mathcal{S}_t$ and $S_t$ | The undecided sets of nodes and regions, respectively, at round $t$ |
| $\hat{\mathcal{P}}$ | The $\boldsymbol{\epsilon}$-accurate Pareto set of nodes returned by Algorithm 1 |
| $\mathcal{A}_t$ | The union of sets $\mathcal{S}_t$ and $\mathcal{P}_t$ at the beginning of round $t$ |
| $\mathcal{W}_t$ | The union of sets $\mathcal{S}_t$ and $\mathcal{P}_t$ at the end of the discarding phase of round $t$ |
| $\boldsymbol{L}_t(x_{h,i})$ and $\boldsymbol{U}_t(x_{h,i})$ | The lower and upper vector-valued indices of node $x_{h,i}$ at time $t$, whose $j$th components are $L_t^j(x_{h,i})$ and $U_t^j(x_{h,i})$ |
| $\overline{B}_t^j(x_{h,i})$ and $\underline{B}_t^j(x_{h,i})$ | The auxiliary indices of the lower and upper index of node $x_{h,i}$ at time $t$ |
| $\boldsymbol{V}_h$ | The $m$-dimensional vector with components are equal to $V_h$, which appears in high probability bounds on the variation of $\boldsymbol{f}$ |
| $\boldsymbol{Q}_t(x_{h,i})$ | The confidence hyper-rectangle associated with node $x_{h,i}$ at round $t$ |
| $\boldsymbol{R}_t(x_{h,i})$ | The cumulative confidence hyper-rectangle associated with node $x_{h,i}$ at round $t$ |
| $\omega_t(x_{h,i})$ | The diameter of the cumulative confidence hyper-rectangle of $x_{h,i}$ at round $t$ |
| $\overline{\omega}_t$ | The maximum $\omega_t(x_{h,i})$ over all active nodes at round $t$ |
| $D_1$ | The metric dimension of $\mathcal{X}$ |
| $\gamma_T$ | The maximum information gain in $T$ evaluations of $\boldsymbol{f}$ |
| $\tau_s$ | The number of evaluations performed by the algorithm until termination |
| $t_s$ | The round in which the algorithm terminates |

Thus, $l_j(x,y) \leq C_{\boldsymbol{k}} d(x,y)^\alpha \leq r$, which implies that $\tilde{X}$ is an $r$-covering of $(\mathcal{X}, l_j)$. By the definition of the covering number, we have

$$N(\mathcal{X}, r, l_j) \leq |\tilde{X}| \,,$$

and since this holds for any $(\frac{r}{C_{\boldsymbol{k}}})^{\frac{1}{\alpha}}$-covering of $(\mathcal{X}, d)$, we conclude that

$$N(\mathcal{X}, r, l_j) \leq N\left(\mathcal{X}, \left(\frac{r}{C_{\boldsymbol{k}}}\right)^{\frac{1}{\alpha}}, d\right).$$

Next, we will show that $D_1^j \leq D_1/\alpha$. For this, it is enough to show that $D_1^j \leq \bar{D}_1/\alpha$ for every $\bar{D}_1 > D_1$. By Lemma 1, there exists a constant $Q_1 \geq 1$ such that $N(\mathcal{X}, r, d) \leq Q_1 r^{-\bar{D}_1}$ for all $r > 0$. In particular, for every $r > 0$, we have

$$N(\mathcal{X}, r, l_j) \leq N\left(\mathcal{X}, \left(\frac{r}{C_{\boldsymbol{k}}}\right)^{\frac{1}{\alpha}}, d\right) \leq Q_1 \left(\frac{r}{C_K}\right)^{\frac{-\bar{D}_1}{\alpha}} \leq \left(\frac{Q_1}{(C_{\boldsymbol{k}})^{-\bar{D}_1/\alpha}}\right) r^{\bar{D}_1/\alpha}.$$

For all $r > 0$, assuming $C_{\boldsymbol{k}} \geq 1$, we have

$$\log N(\mathcal{X}, r, l_j) \leq \log Q_1 + (\bar{D}_1/\alpha) \log(C_{\boldsymbol{k}}) - (\bar{D}_1/\alpha) \log(r).$$

Therefore, there exists $Q_2 > 0$, such that for all $r > 0$, we have $\log N(\mathcal{X}, r, l_j) \leq Q_2 - (\bar{D}_1/\alpha) \log r$, with $Q_2 = \log Q_1 + (\bar{D}_1/\alpha) \log(C_{\boldsymbol{k}})$. Hence, we conclude that $D_1^j \leq D_1/\alpha$. ∎

Next, we state a proposition that relates the information gain with the posterior variance of the GP after each evaluation. This proposition will help us in obtaining information-type bounds on the sample complexity. Since its proof is lengthy, we defer it to Section B.9.

**Proposition 2.** *Let $T \in \mathbb{N}$ and $\tilde{x}_{[T]} = [\tilde{x}_1, \ldots, \tilde{x}_T]^\mathsf{T}$ be the finite sequence of designs that are evaluated. Consider the corresponding vector $\boldsymbol{f}_{[T]} = [\boldsymbol{f}(\tilde{x}_1)^\mathsf{T}, \ldots, \boldsymbol{f}(\tilde{x}_T)^\mathsf{T}]^\mathsf{T}$ of unobserved objective values and the vector $\boldsymbol{y}_{[T]} = [\boldsymbol{y}_1^\mathsf{T}, \ldots, \boldsymbol{y}_T^\mathsf{T}]^\mathsf{T}$ of noisy observations. Then, we have*

$$I(\boldsymbol{y}_{[T]}; \boldsymbol{f}_{[T]}) \geq \frac{1}{2m} \sum_{\tau=1}^{T} \sum_{j=1}^{m} \log(1 + \sigma^{-2}(\sigma_{\tau-1}^j(\tilde{x}_\tau))^2).$$

## B.2 Termination condition

In this section, we derive a sufficient condition under which Adaptive $\boldsymbol{\epsilon}$-PAL terminates. We also give an upper bound on the maximum depth node that can be created by Adaptive $\boldsymbol{\epsilon}$-PAL until it terminates.

In the lemma given below, we show that the algorithm terminates at latest when the diameter of the most uncertain node has fallen below $\min_j \epsilon^j$.

**Lemma 3.** *(Termination condition for Adaptive $\boldsymbol{\epsilon}$-PAL) Let $\epsilon = \min_j \epsilon^j > 0$, where $(\epsilon^1, \ldots, \epsilon^m) = \boldsymbol{\epsilon}$. When running Adaptive $\boldsymbol{\epsilon}$-PAL, if $\overline{\omega}_t < \epsilon$ holds at round $t$, then the algorithm terminates without further sampling.*

*Proof.* Since Adaptive $\boldsymbol{\epsilon}$-PAL updates $\mathcal{S}_t$ and $\mathcal{P}_t$ at the end of discarding and $\boldsymbol{\epsilon}$-covering phases, contents of these sets might change within round $t$. Thus, we let $\mathcal{S}_{t,0}$ ($\mathcal{P}_{t,0}$), $\mathcal{S}_{t,1}$ ($\mathcal{P}_{t,1}$) and $\mathcal{S}_{t,2}$ ($\mathcal{P}_{t,2}$) represent the elements in $\mathcal{S}_t$ ($\mathcal{P}_t$) at the end of modeling, discarding and covering phases, respectively. These sets are related in the following ways: $\mathcal{S}_{t,0} \supseteq \mathcal{S}_{t,1} \supseteq \mathcal{S}_{t,2}$, $\mathcal{P}_{t,0} = \mathcal{P}_{t,1} \subseteq \mathcal{P}_{t,2}$ and $\mathcal{S}_{t,1} \cup \mathcal{P}_{t,1} = \mathcal{S}_{t,2} \cup \mathcal{P}_{t,2}$.

Next, we state a claim from which termination immediately follows.

**Claim 1.** *If $\overline{\omega}_t < \epsilon$ holds at iteration $t$, then for all $x_{h,i} \in \mathcal{S}_{t,0} \setminus \mathcal{P}_{t,2}$, we have $x_{h,i} \notin \mathcal{S}_{t,1}$.*

If Claim 1 holds, then any $x_{h,i} \in \mathcal{S}_{t,0} \setminus \mathcal{P}_{t,2}$ must be discarded by the end of the discarding phase of Adaptive $\boldsymbol{\epsilon}$-PAL. This implies that any $x_{h,i} \in \mathcal{S}_{t,0}$ is either discarded or moved to $\mathcal{P}_{t,2}$ by the end of the $\boldsymbol{\epsilon}$-covering phase of round $t$, thereby completing the proof.

Next, we prove Claim 1. Note that $\overline{\omega}_t$ is an upper bound on $\|\max(\boldsymbol{R}_t(x_{h,i})) - \min(\boldsymbol{R}_t(x_{h,i}))\|_2$, for all $x_{h,i} \in \mathcal{S}_{t,2} \cup \mathcal{P}_{t,2}$. For each $x_{h,i} \in \mathcal{S}_{t,2} \cup \mathcal{P}_{t,2}$ define

$$\boldsymbol{\omega}_{h,i} = \max(\boldsymbol{R}_t(x_{h,i})) - \min(\boldsymbol{R}_t(x_{h,i})).$$

We have $\left\|\boldsymbol{\omega}_{h,i}\right\|_2 \leq \overline{\omega}_t < \epsilon$, which implies that $\boldsymbol{\omega}_{h,i} \prec \boldsymbol{\epsilon}$ since $\omega_{h,i}^j \leq \sqrt{\sum_{j'=1}^m (\omega_{h,i}^{j'})^2} < \epsilon \leq \epsilon^j$ for all $j \in [m]$.

We will show that if $x_{h,i} \in \mathcal{S}_{t,0} \setminus \mathcal{P}_{t,2}$ holds, then $x_{h,i}$ cannot belong to $S_{t,1}$. To prove this, assume that $x_{h,i} \in S_{t,1}$. Since $x_{h,i} \notin \mathcal{P}_{t,2}$, then by the $\boldsymbol{\epsilon}$-covering rule of Adaptive $\boldsymbol{\epsilon}$-PAL specified in line 15 of Algorithm 1, there exists some $y^* \in \mathcal{S}_{t,1} \cup \mathcal{P}_{t,1}$ for which

$$\min(\boldsymbol{R}_t(x_{h,i})) + \boldsymbol{\epsilon} \preceq \max(\boldsymbol{R}_t(y^*)) . \tag{3}$$

Since $\mathcal{S}_{t,1} \cup \mathcal{P}_{t,1} = \mathcal{S}_{t,2} \cup \mathcal{P}_{t,2}$, we have, for all $y \in \mathcal{S}_{t,1} \cup \mathcal{P}_{t,1}$, that

$$\max(\boldsymbol{R}_t(y)) - \min(\boldsymbol{R}_t(y)) \prec \boldsymbol{\epsilon} . \tag{4}$$

Combining equation 3 and equation 4, we obtain

$$\begin{aligned}
\max(\boldsymbol{R}_t(x_{h,i})) = \min(\boldsymbol{R}_t(x_{h,i})) + \boldsymbol{\omega}_{h,i} &\prec \min(\boldsymbol{R}_t(x_{h,i})) + \boldsymbol{\epsilon} \\
&\preceq \max(\boldsymbol{R}_t(y^*)) \\
&\prec \min(\boldsymbol{R}_t(y^*)) + \boldsymbol{\epsilon} .
\end{aligned} \tag{5}$$

An immediate consequence of equation 5 is that

$$\min(\boldsymbol{R}_t(x_{h,i})) \prec \min(\boldsymbol{R}_t(y^*)) . \tag{6}$$

Since $y^* \in \mathcal{S}_{t,0} \cup \mathcal{P}_{t,0}$, by Definition 8 and equation 6, we conclude that $x_{h,i} \notin \mathcal{P}_{pess,t}$. Thus, we must have $x_{h,i} \in S_{t,0} \setminus \mathcal{P}_{pess,t}$. Finally, we claim that $\max(\boldsymbol{R}_t(x_{h,i})) \preceq_{\boldsymbol{\epsilon}} \min(\boldsymbol{R}_t(y^{**}))$ for some $y^{**} \in \mathcal{P}_{pess,t}$. Since equation 5 implies that the condition $\max(\boldsymbol{R}_t(x_{h,i})) \preceq_{\boldsymbol{\epsilon}} \min(\boldsymbol{R}_t(y^*))$ is satisfied, then, if $y^* \in \mathcal{P}_{pess,t}$, we can simply set $y^{**} = y^*$. Else if $y^* \notin \mathcal{P}_{pess,t}$, then this will imply by Definition 8 existence of $y^{**} \in \mathcal{P}_{pess,t}$ such that $\min(\boldsymbol{R}_t(y^*)) \prec \min(\boldsymbol{R}_t(y^{**}))$, which in turn together with equation 5 implies that $\max(\boldsymbol{R}_t(x_{h,i})) \prec_{\boldsymbol{\epsilon}} \min(\boldsymbol{R}_t(y^{**}))$. Then, by the discarding rule of Adaptive $\boldsymbol{\epsilon}$-PAL specified in line 9 of Algorithm 1, we must have $x_{h,i}$ discarded by the end of the discarding phase, which implies that $x_{h,i}$ cannot be in $S_{t,1}$. This proves that all $x_{h,i} \in \mathcal{S}_{t,0} \setminus \mathcal{P}_{t,2}$ must be discarded. ∎

We will prove information-type and dimension-type sample complexity bounds by making use of the termination condition given in Lemma 3.

### B.3 Guarantees on the termination of Adaptive $\epsilon$-PAL

We start by stating a bound on the maximum depth node that can be created by Adaptive $\boldsymbol{\epsilon}$-PAL before it terminates for a given $\boldsymbol{\epsilon}$. This result will be used in defining "good" events that hold with high probability under which $\boldsymbol{f}(x_{h,i})$ lies in the confidence hyper-rectangle of $x_{h,i}$ formed by Adaptive $\boldsymbol{\epsilon}$-PAL, for all possible nodes that can be created by the algorithm (see Section B.4). Our bounds on sample complexity will hold given that these "good" events happen.

**Lemma 4.** *Given $\boldsymbol{\epsilon}$, there exists $h_{max} \in \mathbb{N}$ (dependent on $\boldsymbol{\epsilon}$) such that Adaptive $\boldsymbol{\epsilon}$-PAL stops refining at level $h_{max}$.*

*Proof.* By Lemma 3, at the latest, the algorithm terminates at round $t$ for which $\overline{\omega}_t < \epsilon$, i.e., $\overline{\omega}_t^2 < \epsilon^2$. By definition, at a refining round we have

$$\begin{aligned}
\overline{\omega}_t^2 &\leq \max_{y,y' \in \boldsymbol{Q}_t(x_{h_t,i_t})} \left\|y - y'\right\|_2^2 \\
&= \left\|\boldsymbol{U}_t(x_{h_t,i_t}) - \boldsymbol{L}_t(x_{h_t,i_t})\right\|_2^2 \\
&= \sum_{j=1}^m \left(U_t^j(x_{h_t,i_t}) - L_t^j(x_{h_t,i_t})\right)^2 \\
&= \sum_{j=1}^m \left(\overline{B}_t^j(x_{h_t,i_t}) - \underline{B}_t^j(x_{h_t,i_t}) + 2V_{h_t}\right)^2
\end{aligned}$$

$$\leq \sum_{j=1}^{m} \left( 2\beta_{\tau_t}^{1/2} \sigma_{\tau_t}^j(x_{h_t, i_t}) + 2V_{h_t} \right)^2$$

$$= \left( 4 \sum_{j=1}^{m} \beta_{\tau_t} (\sigma_{\tau_t}^j(x_{h_t, i_t}))^2 + 8 \sum_{j=1}^{m} V_{h_t} \beta_{\tau_t}^{1/2} \sigma_{\tau_t}^j(x_{h_t, i_t}) + 4 \sum_{j=1}^{m} (V_{h_t})^2 \right)$$

$$\leq \left( 4 \sum_{j=1}^{m} (V_{h_t})^2 + 8 \left( \sum_{j=1}^{m} (V_{h_t})^2 \right)^{1/2} \left( \sum_{j=1}^{m} (V_{h_t})^2 \right)^{1/2} + 4 \sum_{j=1}^{m} (V_{h_t})^2 \right) \quad (7)$$

$$= 16 m V_{h_t}^2 \ ,$$

where equation 7 follows from the fact that we refine at round $t$ and from the Cauchy-Schwarz inequality. Thus, if at round $t$, we have

$$16 m V_{h_t}^2 < \epsilon^2 \ ,$$

then we guarantee termination of the algorithm. Recall that we have defined

$$V_h = 4 C_{\boldsymbol{k}}(v_1 \rho^h)^\alpha \left( \sqrt{C_2 + 2\log(2h^2\pi^2 m/6\delta) + h \log N + \max\{0, -4(D_1/\alpha)\log(C_{\boldsymbol{k}}(v_1\rho^h)^\alpha)\}} + C_3 \right) \ ,$$

for positive constants $C_2$ and $C_3$ defined in Corollary 1. Obviously, $V_h$ decays to 0 exponentially in $h$. Thus, by letting $h_{max} = h_{max}(\epsilon)$ to be the smallest $h \geq 0$, for which $16 m V_{h_{max}}^2 < \epsilon^2$ holds, it is observed that the algorithm stops refining at level $h_{max}$. ∎

Our next result gives an upper bound on the maximum number of times a node can be evaluated before it is expanded.

**Lemma 5.** *Let $h \geq 0$ and $i \in [N^h]$. Let $\tau_s$ be the number of evaluations performed by Adaptive $\boldsymbol{\epsilon}$-PAL until termination. Any active node $x_{h,i}$ may be evaluated no more than $q_h$ times before it is expanded, where*

$$q_h = \frac{\sigma^2 \beta_{\tau_s}}{V_h^2} \ .$$

*Furthermore, we have*

$$q_h \leq \frac{\sigma^2 \beta_{\tau_s}}{g(v_1 \rho^h)^2 C_3} \ .$$

*Proof.* The proof is similar to the proof of (Shekhar et al., 2018, Lemma 1). Fix $t \geq 1$. By definition, the vector $\boldsymbol{y}_{[\tau_t]}$ denotes the evaluations made prior to round $t$. Let $\boldsymbol{y}_{x_{h,i}}$ be the vector that represents the subset of evaluations in $\boldsymbol{y}_{[\tau_t]}$ made at node $x_{h,i}$ prior to round $t$, and let $n_t(x_{h,i})$ represent the number of evaluations in $\boldsymbol{y}_{x_{h,i}}$. Similarly, let $\overline{\boldsymbol{y}}_{x_{h,i}}$ be the vector that represents the subset of evaluations in $\boldsymbol{y}_{[\tau_t]}$ made at nodes other than node $x_{h,i}$ prior to round $t$. For any $j \in [m]$, by non-negativity of the information gain, we have

$$I(f^j(x_{h,i}); \boldsymbol{y}_{x_{h,i}} | \overline{\boldsymbol{y}}_{x_{h,i}}) = H(f^j(x_{h,i}) | \boldsymbol{y}_{x_{h,i}}) - H(f^j(x_{h,i}) | \boldsymbol{y}_{x_{h,i}}, \overline{\boldsymbol{y}}_{x_{h,i}}) \geq 0.$$

Furthermore, let $y_{x_{h,i}}^j$ be the vector of evaluations that corresponds to the $j$th objective at node $x_{h,i}$, and $\overline{y}_{x_{h,i}}^j$ be the vector of evaluations that corresponds to the $j$th objective at nodes other than node $x_{h,i}$ prior to round $t$. Since conditioning on more variables reduces the entropy, we have $H(f^j(x_{h,i}) | \boldsymbol{y}_{x_{h,i}}) \leq H(f^j(x_{h,i}) | y_{x_{h,i}}^j)$, and thus,

$$H(f^j(x_{h,i}) | y_{x_{h,i}}^j) - H(f^j(x_{h,i}) | \boldsymbol{y}_{x_{h,i}}, \overline{\boldsymbol{y}}_{x_{h,i}}) \geq 0 \ .$$

Using the definition of conditional entropy for Gaussian random variables, after a short algebraic calculation, we get

$$\frac{1}{2} \log \left( \left| \frac{2\pi e}{n_t(x_{h,i})/\sigma^2 + (k^{jj}(x_{h,i}, x_{h,i}))^{-1}} \right| \right) - \frac{1}{2} \log(|2\pi e(\sigma_{\tau_t}^j(x_{h,i}))^2|) \geq 0 \ .$$

Since $\boldsymbol{k}(x_{h,i}, x_{h,i})$ is positive definite, we have $k^{jj}(x_{h,i}, x_{h,i}) > 0$, and as a result we obtain the following.

$$(\sigma_{\tau_t}^j(x_{h,i}))^{-2} \geq \frac{n_t(x_{h,i})}{\sigma^2} + \frac{1}{k^{jj}(x_{h,i}, x_{h,i})} \geq \frac{n_t(x_{h,i})}{\sigma^2}$$

Thus, we have that $(\sigma_{\tau_t}^j(x_{h,i}))^2 \leq \sigma^2/n_t(x_{h,i})$. If the algorithm has not yet refined, then it means that we have

$$mV_h^2 < \beta_{\tau_t} \sum_{j=1}^m (\sigma_{\tau_t}^j)^2 \leq \beta_{\tau_s} m\sigma^2/n_t(x_{h,i}) \ .$$

Therefore, we obtain

$$n_t(x_{h,i}) \leq q_h = \frac{\sigma^2 \beta_{\tau_s}}{V_h^2} \ .$$

The second statement of the result follows from the trivial observation that

$$\frac{\sigma^2 \beta_{\tau_s}}{V_h^2} \leq \frac{\sigma^2 \beta_{\tau_s}}{g(v_1 \rho^h)^2 C_3} \ .$$

$\blacksquare$

Next, we show that Adaptive $\boldsymbol{\epsilon}$-PAL terminates in finite time.

**Proposition 3.** *Given $\boldsymbol{\epsilon}$ such that $\min_j \epsilon^j > 0$, Adaptive $\boldsymbol{\epsilon}$-PAL terminates in finite time.*

*Proof.* By Lemma 4, the algorithm refines no deeper than $h_{max}(\boldsymbol{\epsilon})$ until termination. Moreover, the algorithm cannot evaluate a node $x_{h,i}$ indefinitely, since there must exist some finite $\tau_t$ for which $\beta_{\tau_t}^{1/2} \|\boldsymbol{\sigma}_{\tau_t}(x_{h,i})\|_2 \leq \|\boldsymbol{V}_h\|_2$ holds. This observation is a consequence of the fact that $(\sigma_{\tau_t}^j(x_{h,i}))^2 \leq \sigma^2/n_t(x_{h,i})$, given in the proof of Lemma 5, where $n_t(x_{h,i})$ represents the number of evaluations made at node $x_{h,i}$ prior to round $t$. Let $t'_s$ be the round that comes just after the round in which $s$ evaluations are made at node $x_{h,i}$. Since $(\sigma_{\tau_{t'_s}}^j(x_{h,i}))^2 \leq \sigma^2/s$, we conclude by observing that there exists $s \in \mathbb{N}$ such that $\beta_{\tau_{t'_s}}^{1/2} \left\|\boldsymbol{\sigma}_{\tau_{t'_s}}(x_{h,i})\right\|_2 \leq \|\boldsymbol{V}_h\|_2$ holds. $\blacksquare$

### B.4 Two "good" events under which the sample complexity will be bounded

First, we show that the indices of all possible nodes that could be created by Adaptive $\boldsymbol{\epsilon}$-PAL do not deviate too much from the true mean objective values in all objectives, and that similar designs yield similar outcomes with high probability. To that end let us denote by $\mathcal{T}_{h_{max}}$ the set of all nodes that can be created until the level $h_{max}$, where $h_{max}$ comes from Lemma 4. Note that we have

$$\mathcal{T}_{h_{max}} = \cup_{h=0}^{h_{max}} \mathcal{X}_h.$$

**Lemma 6.** *(The first "good" event) For any $\delta \in (0, 1)$, the probability of the following event is at least $1 - \delta/2$:*

$$\mathcal{F}_1 = \{\forall j \in [m], \forall \tau \geq 0, \forall x \in \mathcal{T}_{h_{max}} : |f^j(x) - \mu_\tau^j(x)| \leq \beta_\tau^{1/2} \sigma_\tau^j\},$$

*where $\beta_\tau = 2\log(2m\pi^2 N^{h_{max}+1}(\tau + 1)^2/(3\delta))$ with $h_{max}$ being the deepest level of the tree before termination.*

*Proof.* We have:

$$1 - \mathbb{P}(\mathcal{F}_1) = \mathbb{E}\left[\mathbb{I}\left(\exists j \in [m], \exists \tau \geq 0, \exists x \in \mathcal{T}_{h_{max}} : |f^j(x) - \mu_\tau^j(x)| > \beta_\tau^{1/2} \sigma_\tau^j(x)\right)\right]$$

$$\leq \mathbb{E}\left[\sum_{j=1}^m \sum_{\tau \geq 0} \sum_{x \in \mathcal{T}_{h_{max}}} \mathbb{I}\left(|f^j(x) - \mu_\tau^j(x)| > \beta_\tau^{1/2} \sigma_\tau^j(x)\right)\right]$$

$$= \sum_{j=1}^{m} \sum_{\tau \geq 0} \sum_{x \in \mathcal{T}_{h_{max}}} \mathbb{E}\left[\mathbb{E}\left[\mathbb{I}\left(|f^j(x) - \mu_\tau^j(x)| > \beta_\tau^{1/2}\sigma_\tau^j(x)\right) | \boldsymbol{y}_{[\tau]}\right]\right] \tag{8}$$

$$= \sum_{j=1}^{m} \sum_{\tau \geq 0} \sum_{x \in \mathcal{T}_{h_{max}}} \mathbb{E}\left[\mathbb{P}\left\{|f^j(x) - \mu_\tau^j(x)| > \beta_\tau^{1/2}\sigma_\tau^j(x) | \boldsymbol{y}_{[\tau]}\right\}\right]$$

$$\leq \sum_{j=1}^{m} \sum_{\tau \geq 0} \sum_{x \in \mathcal{T}_{h_{max}}} 2e^{-\beta_\tau/2} \tag{9}$$

$$\leq 2mN^{h_{max}+1} \sum_{\tau \geq 0} e^{-\beta_\tau/2} \tag{10}$$

$$= 2mN^{h_{max}+1} \sum_{\tau \geq 0} (2m\pi^2 N^{h_{max}+1}(\tau+1)^2/(3\delta))^{-1}$$

$$= \frac{\delta}{2}\frac{6}{\pi^2} \sum_{\tau \geq 0} (\tau+1)^{-2} = \frac{\delta}{2} \ ,$$

where equation 8 uses the tower rule and linearity of expectation; equation 9 uses Gaussian tail bounds (note that $f^j(x) \sim \mathcal{N}(\mu_{\tau_t}^j(x), \sigma_{\tau_t}^j(x))$ conditioned on $\boldsymbol{y}_{[\tau_t]}$) and equation 10 uses the fact that for any $t \geq 1$, the cardinality of $\mathcal{T}_{h_{max}}$ is $1 + N + N^2 + \ldots + N^{h_{max}} = (N^{h_{max}+1} - 1)/(N-1) \leq N^{h_{max}+1}$, since $N \geq 2$. ∎

Next, we introduce a bound on the maximum variation of the function inside a region. First, we state a result taken from Shekhar et al. (2018) on which this bound is based. Suppose $\{g(x); x \in \mathcal{X}\}$ is a separable zero mean single output Gaussian Process $GP(0, k)$ and let $l$ be the GP-induced metric on $\mathcal{X}$. Let $D_1'$ be the metric dimension of $\mathcal{X}$ with respect to $l$. By Lemma 1, we have that if $D_1' < \infty$, then there exists a positive constant $\tilde{C}_1$ depending on $2D_1'$ such that for any $z \leq \text{diam}(\mathcal{X})$ we have $N(\mathcal{X}, z, l) \leq \tilde{C}_1 z^{-2D_1'}$. Let $\eta_1 = \sum_{n \geq 1} 2^{-(n-1)}\sqrt{\log n}$, $\eta_2 = \sum_{n \geq 1} 2^{-(n-1)}\sqrt{n}$, and define

$$\tilde{C}_2 = 2\log(2\tilde{C}_1^2\pi^2/6) \ \text{ and } \ \tilde{C}_3 = \eta_1 + \eta_2\sqrt{2D_1'\log 2} \ .$$

**Lemma 7.** *(Proposition 1, Section 6, Shekhar et al. (2018)) Let $x_0 \in \mathcal{X}$ and $B(x_0, b, l) \subset \mathcal{X}$ be an l-ball of radius $b > 0$, where $l$ is the GP-induced metric on $\mathcal{X}$. Then, we have for any $u > 0$*

$$\mathbb{P}\left\{\sup_{x \in B(x_0,b,l)} |g(x) - g(x_0)| > \omega(b)\right\} \leq e^{-u} \ ,$$

*where* $\omega(b) = 4b\left(\sqrt{\tilde{C}_2 + 2u + \max\{0, 4D_1'\log(1/b)\}} + \tilde{C}_3\right)$.

**Remark 3.** *Note that by Remark 1, for every $j \in [m]$, the covariance function $k^{jj}(x, x)$ is continuous with respect to the metric $l_j$, and thus, the process $\{f^j(x); x \in \mathcal{X}\}$ is separable.*

**Corollary 1.** *(The second "good" event) For any $\delta \in (0, 1)$, the probability of the following event is at least $1 - \delta/2$:*

$$\mathcal{F}_2 = \left\{\forall h \geq 0, \forall i \in [N^h], \forall j \in [m] : \sup_{x \in B(x_{h,i}, v_1\rho^h, d)} |f^j(x) - f^j(x_{h,i})| \leq V_h\right\} \ ,$$

*where*

$$V_h = 4C_{\boldsymbol{k}}(v_1\rho^h)^\alpha\left(\sqrt{C_2 + 2\log(2h^2\pi^2 m/6\delta) + h\log N + \max\{0, -4(D_1/\alpha)\log(C_{\boldsymbol{k}}(v_1\rho^h)^\alpha)\}} + C_3\right) \ ,$$

*and $C_2$ and $C_3$ are the positive constants defined below which depend on the metric dimension $D_1$ of $\mathcal{X}$ with respect to metric $d$.*

*Proof.* Let $D_1$ be the metric dimension of $\mathcal{X}$ with respect to $d$. By Lemma 2, we have that $D_1^j \leq D_1/\alpha$, where $D_1^j$ is the metric dimension of $\mathcal{X}$ with respect to $l_j$, for all $j \in [m]$. We also have constants $C_1^j$ associated

with $D_1^j$, such that $N(\mathcal{X}, r^\alpha, d) \leq N(\mathcal{X}, r, l_j) \leq C_1^j r^{2D_1^j}$. Let $C_1 = \max_j C_1^j$. Also, let

$$C_2 = 2\log(2C_1^2 \pi^2/6) \text{ and } C_3 = \eta_1 + \eta_2 \sqrt{2D_1 \alpha \log 2} \ .$$

Using Lemma 7 and Remark 2 we let

$$\omega(b) = 4b \left( \sqrt{C_2 + 2u + \max\{0, 4D_1 \log(1/b)\}} + C_3 \right) \ .$$

Now let $u = -\log \delta + \log m + h \log N + \log(2h^2 \pi^2/6)$. We have

$$
\begin{aligned}
1 - \mathbb{P}(\mathcal{F}_2) = \mathbb{P}&\left\{ \exists h \geq 0, \exists i \in [N^h], \exists j \in [m] : \sup_{x \in B(x_{h,i}, v_1 \rho^h, d)} |f^j(x) - f^j(x_{h,i})| > \omega(C_{\boldsymbol{k}}(v_1 \rho^h)^\alpha) \right\} \\
&\leq \sum_{h \geq 0} \sum_{1 \leq i \leq N^h} \sum_{j=1}^m \mathbb{P}\left\{ \sup_{x \in B(x_{h,i}, v_1 \rho^h, d)} |f^j(x) - f^j(x_{h,i})| > \omega(C_{\boldsymbol{k}}(v_1 \rho^h)^\alpha) \right\} \\
&\leq \sum_{h \geq 0} \sum_{1 \leq i \leq N^h} \sum_{j=1}^m \mathbb{P}\left\{ \sup_{x \in B(x_{h,i}, C_{\boldsymbol{k}}(v_1 \rho^h)^\alpha, l)} |f^j(x) - f^j(x_{h,i})| > \omega(C_{\boldsymbol{k}}(v_1 \rho^h)^\alpha) \right\} \quad (11) \\
&\leq \sum_{h \geq 0} \sum_{1 \leq i \leq N^h} \sum_{j=1}^m e^{-u} \leq \delta \frac{\pi^2}{6} \sum_{h \geq 0} \frac{1}{2} m N^h (m N^h)^{-1} h^{-2} \leq \frac{\delta}{2} \ ,
\end{aligned}
$$

where for equation 11 we argue as follows. Note that by Assumption 1, given $x, y \in \mathcal{X}$ and $j \in [m]$, we have $l_j(x, y) \leq C_{\boldsymbol{k}} d(x, y)^\alpha$. In particular, letting $y$ be any design which is $v_1 \rho^h$ away from $x_{h,i}$ under $d$, we have $l_j(x_{h,i}, y) \leq C_{\boldsymbol{k}} d(x_{h,i}, y)^\alpha = C_{\boldsymbol{k}}(v_1 \rho^h)^\alpha$. This implies that $B(x_{h,i}, r, l_j) \subseteq B(x_{h,i}, C_{\boldsymbol{k}}(v_1 \rho^h)^\alpha, l_j)$, where $r := l_j(x_{h,i}, y)$. Note that we have $B(x_{h,i}, r, l_j) = B(x_{h,i}, v_1 \rho^h, d)$. The result follows from observing that the probability that the variation of the function exceeds $\omega(C_{\boldsymbol{k}}(v_1 \rho^h)^\alpha)$ is higher in $B(x_{h,i}, C_{\boldsymbol{k}}(v_1 \rho^h)^\alpha, l_j)$ then in $B(x_{h,i}, v_1 \rho^h, d)$, since $B(x_{h,i}, v_1 \rho^h, d) \subseteq B(x_{h,i}, C_{\boldsymbol{k}}(v_1 \rho^h)^\alpha, l_j)$. ∎

### B.5 Key results that hold under the "good" events

Our next result shows that the objective values of all designs $x \in \mathcal{X}$ belong to the uncertainty hyper-rectangles of the nodes associated to the regions containing them in a given round. To that end, let us denote by $c_t(x)$ the node associated to the cell $C_t(x)$ containing $x$ at the beginning of round $t$. Also, let us denote by $h_t(x)$ the depth of the tree where $c_t(x)$ is located.

**Lemma 8.** *Under events $\mathcal{F}_1$ and $\mathcal{F}_2$, for any round $t \geq 1$ before Adaptive $\boldsymbol{\epsilon}$-PAL terminates and for any $x \in \mathcal{X}$, we have*

$$\boldsymbol{f}(x) \in \boldsymbol{R}_t(c_t(x)) \ .$$

*Proof.* First, let us denote by $1 = s_0 < s_1 < s_2 < \ldots < s_n$ the sequence of stopping times up to round $t$ in which the original node containing design $x$ was refined into children nodes, so that we have $C_{s_n+1}(x) \subseteq C_{s_n}(x) \subseteq \ldots \subseteq C_0(x)$. By the definition of the cumulative confidence hyper-rectangle, for $c_{s_0}(x)$, we have

$$
\begin{aligned}
\boldsymbol{R}_{s_1}(c_{s_0}(x)) &= \boldsymbol{R}_{s_1-1}(c_{s_0}(x)) \cap \boldsymbol{Q}_{t_1}(c_{s_0}(x)) \\
&= \boldsymbol{R}_{s_1-2}(c_{s_0}(x)) \cap \boldsymbol{Q}_{s_1-1}(c_{t_0}(x)) \cap \boldsymbol{Q}_{s_1}(c_{s_0}(x)) \\
&= \boldsymbol{R}_0(c_{s_0}(x)) \cap \boldsymbol{Q}_{s_0}(c_{s_0}(x)) \cap \ldots \cap \boldsymbol{Q}_{s_1}(c_{s_0}(x)) \ ,
\end{aligned}
$$

and since $\boldsymbol{R}_0(c_{s_0}(x)) = \mathbb{R}^m$, we obtain

$$\boldsymbol{R}_{s_1}(c_{s_0}(x)) = \bigcap_{s=1}^{s_1} \boldsymbol{Q}_s(c_{s_0}(x)) \ .$$

Similarly, for $c_{s_1}(x)$ we have

$$\begin{aligned}
\boldsymbol{R}_{s_2}(c_{s_1}(x)) &= \boldsymbol{R}_{s_2-1}(c_{s_1}(x)) \cap \boldsymbol{Q}_{s_2}(c_{s_1}(x)) \\
&= \boldsymbol{R}_{s_1}(c_{s_1}(x)) \cap \boldsymbol{Q}_{s_1+1}(c_{s_1}(x)) \cap \ldots \cap \boldsymbol{Q}_{s_2}(c_{s_1}(x)) \ ,
\end{aligned}$$

where

$$\boldsymbol{R}_{s_1}(c_{s_1}(x)) = \boldsymbol{R}_{s_1}(p(c_{s_1}(x))) = \boldsymbol{R}_{s_1}(c_{s_0}(x)) \ .$$

Thus, we obtain

$$\boldsymbol{R}_{s_2}(c_{s_1}(x)) = \left( \bigcap_{s=1}^{s_1} \boldsymbol{Q}_s(c_{s_0}(x)) \right) \cap \left( \bigcap_{s=s_1+1}^{s_2} \boldsymbol{Q}_s(c_{s_1}(x)) \right) \ .$$

Note that $c_s(x) = c_{s_i}(x)$ for all $s$ such that $s_i < s \le s_{i+1}$ for each $i \in \{0, \ldots, n-1\}$; and $c_s(x) = c_{s_n}(x)$ for all $s$ such that $s_{n+1} < s \le t$. Thus, continuing as above, we can write

$$\begin{aligned}
\boldsymbol{R}_t(c_t(x)) &= \left( \bigcap_{s=1}^{s_1} \boldsymbol{Q}_s(c_{s_0}(x)) \right) \cap \ldots \cap \left( \bigcap_{s=s_{n-1}+1}^{s_n} \boldsymbol{Q}_s(c_{s_{n-1}}(x)) \right) \cap \left( \bigcap_{s=s_n+1}^{t} \boldsymbol{Q}_s(c_{s_n}(x)) \right) \\
&= \bigcap_{s=1}^{t} \boldsymbol{Q}_s(c_s(x)) \ .
\end{aligned}$$

Based on the above display, to prove that $\boldsymbol{f}(x) \in \boldsymbol{R}_t(c_t(x))$, it is enough to show that $\boldsymbol{f}(x) \in \boldsymbol{Q}_s(c_s(x))$, for all $s \le t$. Next, we prove that this is indeed the case, by showing that for any $s \le t$ and $j \in [m]$, it holds that

$$L_s^j(c_s(x)) = \underline{B}_s^j(c_s(x)) - V_{h_s(x)} \le f^j(x) \le \overline{B}_s^j(c_s(x)) + V_{h_s(x)} = U_s^j(c_s(x)) \ . \tag{12}$$

To show this, we first note that by definition

$$\underline{B}_s^j(c_s(x)) = \max\{\mu_{\tau_s}^j(c_s(x)) - \beta_{\tau_s}^{1/2}\sigma_{\tau_s}^j(c_s(x)), \ \mu_{\tau_s}^j(p(c_s(x))) - \beta_{\tau_s}^{1/2}\sigma_{\tau_s}^j(p(c_s(x))) - V_{h_s(x)-1}\} \ ,$$

and

$$\overline{B}_s^j(c_s(x)) = \min\{\mu_{\tau_s}^j(c_s(x)) + \beta_{\tau_s}^{1/2}\sigma_{\tau_s}^j(c_s(x)), \ \mu_{\tau_s}^j(p(c_s(x))) + \beta_{\tau_s}^{1/2}\sigma_{\tau_s}^j(p(c_s(x))) + V_{h_s(x)-1}\} \ .$$

Hence, we need to consider four cases: two cases for $\underline{B}_s^j(c_s(x))$ and two cases for $\overline{B}_s^j(c_s(x))$. Let $j \in [m]$. Note that under events $\mathcal{F}_1$ and $\mathcal{F}_2$, we have

$$\mu_{\tau_s}^j(c_s(x)) - \beta_{\tau_s}^{1/2}\sigma_{\tau_s}^j(c_s(x)) \le f^j(c_t(x)) \le \mu_{\tau_s}^j(c_s(x)) + \beta_{\tau_s}^{1/2}\sigma_{\tau_s}^j(c_s(x)) \ , \tag{13}$$

and

$$f^j(c_s(x)) - V_{h_s(x)} \le f^j(x) \le f^j(c_s(x)) + V_{h_s(x)} \ . \tag{14}$$

The following inequalities hold under $\mathcal{F}_1$ and $\mathcal{F}_2$.

**Case 1:** If we have $L_s^j(c_s(x)) = \mu_{\tau_s}^j(c_s(x)) - \beta_{\tau_s}^{1/2}\sigma_{\tau_s}^j(c_s(x)) - V_{h_s(x)}$, then

$$\begin{aligned}
L_t^j(c_s(x)) &= \mu_{\tau_s}^j(c_s(x)) - \beta_{\tau_s}^{1/2}\sigma_{\tau_s}^j(c_s(x)) - V_{h_s(x)} \\
&\le f^j(c_s(x)) - V_{h_s(x)} \\
&\le f^j(x) \ ,
\end{aligned}$$

where the first inequality follows from equation 13 and the second inequality follows from equation 14.

**Case 2:** If we have $L_s^j(c_s(x)) = \mu_{\tau_s}^j(p(c_s(x))) - \beta_{\tau_s}^{1/2}\sigma_{\tau_s}^j(p(c_s(x))) - V_{h_s(x)-1} - V_{h_s(x)}$, then

$$
\begin{aligned}
L_s^j(c_s(x)) &= \mu_{\tau_s}^j(p(c_s(x))) - \beta_{\tau_s}^{1/2}\sigma_{\tau_s}^j(p(c_s(x))) - V_{h_s(x)-1} - V_{h_s(x)} \\
&\le f^j(p(c_s(x))) - V_{h_s(x)-1} - V_{h_s(x)} \\
&\le f^j(c_s(x)) - V_{h_s(x)} \\
&\le f^j(x) ,
\end{aligned}
$$

where the first inequality follows from equation 13; for the second inequality we use the fact that $c_s(x)$ belongs to the cell associated to $p(c_s(x))$, and thus we have that $f^j(p(c_s(x))) \le f^j(c_s(x)) + V_{h_s(x)-1}$; the third inequality follows from equation 14.

**Case 3:** If we have $U_s^j(c_s(x)) = \mu_{\tau_s}^j(c_s(x)) + \beta_{\tau_s}^{1/2}\sigma_{\tau_s}^j(c_s(x)) + V_{h_s(x)}$, then

$$
\begin{aligned}
f^j(x) &\le f^j(c_s(x)) + V_{h_s(x)} \\
&\le \mu_{\tau_s}^j(c_s(x)) + \beta_{\tau_s}^{1/2}\sigma_{\tau_s}^j(c_s(x)) + V_{h_s(x)} \\
&= U_s^j(c_s(x)) ,
\end{aligned}
$$

where the first inequality follows from equation 14 and the second inequality follows from equation 13.

**Case 4:** If we have $U_s^j(c_s(x)) = \mu_{\tau_s}^j(p(c_s(x))) + \beta_{\tau_s}^{1/2}\sigma_{\tau_s}^j(p(c_s(x))) + V_{h_s(x)-1} + V_{h_s(x)}$, then

$$
\begin{aligned}
f^j(x) &\le f^j(c_s(x)) + V_{h_s(x)} \\
&\le f^j(p(c_s(x))) + V_{h_s(x)-1} + V_{h_s(x)} \\
&\le \mu_{\tau_s}^j(p(c_s(x))) + \beta_{\tau_s}^{1/2}\sigma_{\tau_s}^j(p(c_s(x))) + V_{h_s(x)-1} + V_{h_s(x)} \\
&= U_s^j(c_s(x)) .
\end{aligned}
$$

The analysis of this case follows the same argument as the one of Case 2. This proves that equation 12 holds, and thus, the result follows. ■

Our next result ensures that Adaptive $\epsilon$-PAL does not return "bad" nodes.

**Lemma 9.** *Let $x \in \mathcal{X}$. Under events $\mathcal{F}_1$ and $\mathcal{F}_2$, if $\boldsymbol{f}(x) \notin \mathcal{Z}_\epsilon(\mathcal{X})$, then $x \notin \hat{P}$.*

*Proof.* Suppose that $\boldsymbol{f}(x) \notin \mathcal{Z}_\epsilon(\mathcal{X})$. Then, by Definition 3, we have $\boldsymbol{f}(x) \in \boldsymbol{f}(\mathcal{O}(\mathcal{X})) - 2\epsilon - \mathbb{R}_+^m$, that is, there exists $x^* \in \mathcal{O}(\mathcal{X})$ such that we have

$$\boldsymbol{f}(x) + 2\epsilon \preceq \boldsymbol{f}(x^*) . \tag{15}$$

To get a contradiction, let us assume that $x \in \hat{P}$. This implies that there exists $t \ge 1$ such that $c_t(x)$ is added to $\mathcal{P}_t$ by line 16 of the algorithm pseudocode. Thus, by the $\epsilon$-Pareto front covering rule in line 15 of the algorithm pseudocode, for each $x_{h,i} \in \mathcal{W}_t$, we have

$$\min(\boldsymbol{R}_t(c_t(x))) + \epsilon \npreceq \max(\boldsymbol{R}_t(x_{h,i})) . \tag{16}$$

In particular, we have that $\min(\boldsymbol{R}_t(c_t(x))) + \epsilon \npreceq \max(\boldsymbol{R}_t(c_t(x^*)))$ (note that we may even have $c_t(x) = c_t(x^*)$ and this would yield the same result), which, together with Lemma 8, implies that $\boldsymbol{f}(x) + \epsilon \npreceq \boldsymbol{f}(x^*)$. This contradicts equation 15.

Now since we have $x, x^* \in C_t(x)$, we must have $\min(\boldsymbol{R}_t(c_t(x))) \preceq \boldsymbol{f}(x)$ and $\boldsymbol{f}(x^*) \preceq \max(\boldsymbol{R}_t(c_t(x)))$ by Lemma 8, which implies that $\boldsymbol{f}(x) + \epsilon \npreceq \boldsymbol{f}(x^*)$, thereby contradicting equation 15.

Next, let us assume that $c_t(x^*) \notin \mathcal{W}_t$. This means that there exists a round $s_1 \le t$ at which $c_{s_1}(x^*)$ is discarded. By line 9 of the Algorithm 1, there exists $y_1 \in \mathcal{P}_{pess,s_1}$ such that we have

$$\max(\boldsymbol{R}_{s_1}(c_{s_1}(x^*))) - \epsilon \preceq \min(\boldsymbol{R}_{s_1}(y_1)) . \tag{17}$$

Assume that the child node $c_t(y_1)$ of $y_1$ at round $t$ (due to possible refining in the middle rounds) is still not discarded by round $t$, i.e. $c_t(y_1) \in \mathcal{W}_t$. Then, by equation 16, we have

$$\min(\boldsymbol{R}_t(c_t(x))) + \epsilon \npreceq \max(\boldsymbol{R}_t(c_t(y_1))) ,$$

which implies that

$$\min(\boldsymbol{R}_t(c_t(x))) + \boldsymbol{\epsilon} \not\preceq \min(\boldsymbol{R}_t(c_t(y_1))) \ ,$$

which in return implies that

$$\min(\boldsymbol{R}_t(c_t(x))) + \boldsymbol{\epsilon} \not\preceq \min(\boldsymbol{R}_{s_1}(y_1))$$

due to the fact that $\min(\boldsymbol{R}_{s_1}(y_1)) = \min(\boldsymbol{R}_{s_1}(c_t(y_1))) \preceq \min(\boldsymbol{R}_t(c_t(y_1)))$. Thus, by equation 17, we obtain

$$\min(\boldsymbol{R}_t(c_t(x))) + \boldsymbol{\epsilon} \not\preceq \max(\boldsymbol{R}_{s_1}(c_{s_1}(x^*))) - \boldsymbol{\epsilon} \ ,$$

or equivalently,

$$\min(\boldsymbol{R}_t(c_t(x))) + 2\boldsymbol{\epsilon} \not\preceq \max(\boldsymbol{R}_{s_1}(c_{s_1}(x^*))) \ .$$

By Lemma 8, this implies that $\boldsymbol{f}(x) + 2\boldsymbol{\epsilon} \not\preceq \boldsymbol{f}(x^*)$, contradicting equation 15. Hence, it remains to consider the case $c_t(y_1) \notin \mathcal{W}_t$.

In general, let the finite sequence of nodes $[y_1, y_2, \ldots, y_n]^\mathsf{T}$ be such that all the nodes $y_1$ to $y_{n-1}$ are discarded, meaning that, for each $i \in [n-1]$, the node $y_i$ stopped being part of $\mathcal{P}_{pess,s_{i+1}}$ at some round $s_{i+1}$ by satisfying

$$\min(\boldsymbol{R}_{s_{i+1}}(c_{s_{i+1}}(y_i))) \prec \min(\boldsymbol{R}_{s_{i+1}}(y_{i+1})) \ .$$

Suppose that $c_t(y_n)$ is active in round $t$, meaning that $c_t(y_n) \in \mathcal{W}_t$ (note that we can always find such an $n$ since the algorithm terminates and that we choose the sequence such that it satisfies that condition; furthermore, we have $s_1 < s_2 < \ldots < s_n \leq t$). Since we have $c_t(y_n) \in \mathcal{W}_t$, by equation 16, we obtain

$$\min(\boldsymbol{R}_t(c_t(x))) + \boldsymbol{\epsilon} \not\preceq \max(\boldsymbol{R}_t(c_t(y_n))) \ . \tag{18}$$

Note that equation 17 implies

$$\begin{aligned}
\max(\boldsymbol{R}_{s_1}(c_{s_1}(x^*))) - \boldsymbol{\epsilon} \preceq \min(\boldsymbol{R}_{s_1}(y_1)) &\prec \min(\boldsymbol{R}_{s_2}(c_{s_2}(y_1))) \\
&\prec \min(\boldsymbol{R}_{s_2}(y_2)) \prec \min(\boldsymbol{R}_{s_3}(c_{s_3}(y_2))) \\
&\prec \ldots \\
&\prec \min(\boldsymbol{R}_{s_n}(y_n)) \prec \min(\boldsymbol{R}_t(c_t(y_n))) \ ,
\end{aligned}$$

which in turn implies that

$$\max(\boldsymbol{R}_{s_1}(c_{s_1}(x^*))) - \boldsymbol{\epsilon} \prec \min(\boldsymbol{R}_t(c_t(y_n))) \preceq \max(\boldsymbol{R}_t(c_t(y_n))) \ .$$

This, combined with equation 18 implies

$$\min(\boldsymbol{R}_t(c_t(x))) + 2\boldsymbol{\epsilon} \not\preceq \max(\boldsymbol{R}_{s_1}(c_{s_1}(x^*))) \ .$$

Thus, in all cases we have that $\boldsymbol{f}(x) + 2\boldsymbol{\epsilon} \not\preceq \boldsymbol{f}(x^*)$, contradicting our assumption. We conclude that $x \notin \hat{P}$. ∎

Next, we show that Adaptive $\boldsymbol{\epsilon}$-PAL returns an $\boldsymbol{\epsilon}$-accurate Pareto set when it terminates.

**Lemma 10. *(Adaptive $\boldsymbol{\epsilon}$-PAL returns an $\boldsymbol{\epsilon}$-accurate Pareto set)*** *Under events $\mathcal{F}_1$ and $\mathcal{F}_2$, the set $\hat{P}$ returned by Adaptive $\boldsymbol{\epsilon}$-PAL is an $\boldsymbol{\epsilon}$-accurate Pareto set.*

*Proof.* Let $x \in \mathcal{O}(\mathcal{X})$. We claim that there exists $z \in \hat{P}$ such that $\boldsymbol{f}(x) \preceq_{\boldsymbol{\epsilon}} \boldsymbol{f}(z)$. Note that if $x \in \hat{P}$, then the claim holds trivially. Let us assume that $x \notin \hat{P}$. Then, there exists some round $s_1 \in \mathbb{N}$ at which Adaptive $\boldsymbol{\epsilon}$-PAL discards the node $c_{s_1}(x)$. By line 9 of the algorithm pseudocode, there exists a node $z_1 \in \mathcal{P}_{pess,s_1}$ such that

$$\max(\boldsymbol{R}_{s_1}(c_{s_1}(x))) \preceq \min(\boldsymbol{R}_{s_1}(z_1)) + \boldsymbol{\epsilon} \ .$$

By Lemma 8, we have

$$\boldsymbol{f}(x) \preceq \max(\boldsymbol{R}_{s_1}(c_{s_1}(x))) \preceq \min(\boldsymbol{R}_{s_1}(z_1)) + \boldsymbol{\epsilon} \preceq \boldsymbol{f}(z_1) + \boldsymbol{\epsilon} \ . \tag{19}$$

Hence, if $z_1 \in \hat{P}$, then the claim holds with $z = z_1$.

Now, suppose that $z_1 \notin \hat{P}$. Hence, at some round $s_2 \geq s_1$, the node $c_{s_2}(z_1)$ associated with $z_1$ must be discarded by Adaptive $\boldsymbol{\epsilon}$-PAL. By line 8 of the algorithm pseudocode, we know that a node in $\mathcal{P}_{pess,s_2}$ cannot be discarded at round $s_2$. Hence, $c_{s_2}(z_1) \notin \mathcal{P}_{pess,s_2}$. Then, by Definition 8, there exists a node $z_2 \in \mathcal{A}_{s_2}$ such that

$$\min(\boldsymbol{R}_{s_2}(c_{s_2}(z_1))) \prec \min(\boldsymbol{R}_{s_2}(z_2)) \ ,$$

which, by equation 19, implies that

$$\begin{aligned}
\boldsymbol{f}(x) &\preceq \min(\boldsymbol{R}_{s_1}(z_1)) + \boldsymbol{\epsilon} \\
&= \min(\boldsymbol{R}_{s_1}(c_{s_2}(z_1))) + \boldsymbol{\epsilon} \\
&\preceq \min(\boldsymbol{R}_{s_2}(c_{s_2}(z_1))) + \boldsymbol{\epsilon} \\
&\prec \min(\boldsymbol{R}_{s_2}(z_2)) + \boldsymbol{\epsilon} \\
&\preceq \boldsymbol{f}(z_2) + \boldsymbol{\epsilon} \ .
\end{aligned}$$

Here, the equality the follows since the child node inherits the cumulative confidence hyper-rectangles of its parents at prior rounds (see refining/evaluating phase in Section 4), and the second inequality follows from the fact that the cumulative confidence hyper-rectangles shrink with time. Hence, $\boldsymbol{f}(x) \preceq_{\boldsymbol{\epsilon}} \boldsymbol{f}(z_2)$. If $z_2 \in \hat{P}$, then the claim holds with $z = z_2$.

Suppose that $z_2 \notin \hat{P}$. Then, the above process continues in a similar fashion. Suppose that the process never yields a node in $\hat{P}$. Since the algorithm is guaranteed to terminate by Lemma 3, the process stops and yields a finite sequence $[z_1, \ldots, z_n]^\mathsf{T}$ of designs such that $z_1 \notin \hat{P}, \ldots, z_n \notin \hat{P}$. For each $i \in [n-1]$, let $s_{i+1} \in \mathbb{N}$ be the round at which $c_{s_{i+1}}(z_i)$ is discarded; by the above process, we have

$$\boldsymbol{f}(x) \preceq \min(\boldsymbol{R}_{s_{i+1}}(z_{i+1})) + \boldsymbol{\epsilon} \ .$$

In particular, $\boldsymbol{f}(x) \preceq \min(\boldsymbol{R}_{s_n}(z_n)) + \boldsymbol{\epsilon}$. Since $z_n \notin \hat{P}$, there exists a round $s_{n+1} \geq s_n$ at which the node $c_{s_{n+1}}(z_n)$ is discarded. Consequently, the node $c_{s_{n+1}}(z_n) \notin \mathcal{P}_{pess,s_{n+1}}$. This implies that the condition of Definition 8 is violated at round $s_{n+1}$, that is, there exists $z_{n+1} \in \mathcal{A}_{s_{n+1}}$ such that

$$\min(\boldsymbol{R}_{s_{n+1}}(c_{s_{n+1}}(z_n))) \prec \min(\boldsymbol{R}_{s_{n+1}}(z_{n+1})) \ . \tag{20}$$

At this point, it is important to clarify that $z_{n+1}$ cannot belong to the sequence $[z_1, \ldots, z_n]^\mathsf{T}$ of designs. Note that, by assumption, we have

$$\begin{aligned}
\boldsymbol{f}(x) &\preceq \min(\boldsymbol{R}_{s_1}(z_1)) + \boldsymbol{\epsilon} \preceq \min(\boldsymbol{R}_{s_2}(c_{s_2}(z_1))) + \boldsymbol{\epsilon} \\
&\prec \min(\boldsymbol{R}_{s_2}(z_2)) + \boldsymbol{\epsilon} \preceq \min(\boldsymbol{R}_{s_3}(c_{s_3}(z_2))) + \boldsymbol{\epsilon} \\
&\prec \ldots \\
&\prec \min(\boldsymbol{R}_{s_n}(z_n)) + \boldsymbol{\epsilon} \preceq \min(\boldsymbol{R}_{s_{n+1}}(c_{s_{n+1}}(z_n))) + \boldsymbol{\epsilon} \\
&\prec \min(\boldsymbol{R}_{s_{n+1}}(z_{n+1})) + \boldsymbol{\epsilon} \ .
\end{aligned}$$

Due to the strict domination relation between the nodes in the sequence (which is determined by Definition 8), no two nodes can have identical confidence hyper-rectangles at the time of removal from $\mathcal{P}_{pess,s}$, for $s \in \{s_1, \ldots, s_{n+1}\}$. Moreover, such a condition implies a strict ordering of nodes in the above sense, which implies that $z_{n+1} \notin \{z_1, \ldots, z_n\}$. Hence, equation 20 and the earlier inequalities imply that

$$\begin{aligned}
\boldsymbol{f}(x) &\preceq \min(\boldsymbol{R}_{s_n}(z_n)) + \boldsymbol{\epsilon} \\
&= \min(\boldsymbol{R}_{s_n}(c_{s_{n+1}}(z_n))) + \boldsymbol{\epsilon}
\end{aligned}$$

$$\preceq \min(\boldsymbol{R}_{s_{n+1}}(c_{s_{n+1}}(z_n))) + \boldsymbol{\epsilon}$$
$$\prec \min(\boldsymbol{R}_{s_{n+1}}(z_{n+1})) + \boldsymbol{\epsilon}$$
$$\preceq \boldsymbol{f}(z_{n+1}) + \boldsymbol{\epsilon} \ ,$$

where we have again used the shrinking property of the cumulative confidence hyper-rectangles in the second inequality. This means that $z_{n+1}$ is another node that $\boldsymbol{\epsilon}$-dominates $x$ but it is not in the finite sequence $[z_1, \ldots, z_n]^\mathsf{T}$. This contradicts our assumption that the process stops after finding $n$ designs. Hence, at least one of the designs in $[z_1, \ldots, z_n]^\mathsf{T}$ is in $\hat{P}$; let us call it $z$. This completes the proof of the claim.

Next, let $\boldsymbol{\mu} \in \boldsymbol{f}(\mathcal{O}(\mathcal{X})) - \mathbb{R}_+^m$. Then, there exist $x \in \mathcal{O}(\mathcal{X})$ and $\boldsymbol{\mu}' \in \mathbb{R}_+^m$ such that $\boldsymbol{\mu} = \boldsymbol{f}(x) - \boldsymbol{\mu}'$. By the above claim, there exists $z \in \hat{P}$ such that $\boldsymbol{f}(x) \preceq \boldsymbol{f}(z) + \boldsymbol{\epsilon}$. Hence,

$$\boldsymbol{\mu} \preceq \boldsymbol{\mu} + \boldsymbol{\mu}' = \boldsymbol{f}(x) \preceq \boldsymbol{f}(z) + \boldsymbol{\epsilon} \ .$$

This shows that for every $\boldsymbol{\mu} \in \boldsymbol{f}(\mathcal{O}(\mathcal{X})) - \mathbb{R}_+^m$, there exists $z \in \hat{P}$ such that $\boldsymbol{\mu} \preceq_{\boldsymbol{\epsilon}} \boldsymbol{f}(z)$. By Definition 3, we have $\mathcal{Z}_{\boldsymbol{\epsilon}}(\mathcal{X}) \subseteq \boldsymbol{f}(\mathcal{O}(\mathcal{X})) - \mathbb{R}_+^m$. Hence, for every $\boldsymbol{\mu} \in \mathcal{Z}_{\boldsymbol{\epsilon}}(\mathcal{X})$, there exists $\boldsymbol{\mu}'' \in \boldsymbol{f}(\hat{P})$ such that $\boldsymbol{\mu} \preceq_{\boldsymbol{\epsilon}} \boldsymbol{\mu}''$. On the other hand, since we work under $\mathcal{F}_1 \cap \mathcal{F}_2$, Lemma 9 implies that $\boldsymbol{f}(\hat{P}) \subseteq \mathcal{Z}_{\boldsymbol{\epsilon}}(\mathcal{X})$. Therefore, $\boldsymbol{f}(\hat{P})$ is an $\boldsymbol{\epsilon}$-covering of $\mathcal{Z}_{\boldsymbol{\epsilon}}(\mathcal{X})$ (Definition 4 ), that is, $\hat{P}$ is an $\boldsymbol{\epsilon}$-accurate Pareto set (Definition 5). ∎

### B.6 Derivation of the information-type sample complexity bound

In this section, we provide sample complexity upper bounds that depend on the maximum information gain from observing the evaluated designs.

We begin with an auxiliary lemma whose statement is straightforward for the case $m = 1$.

**Lemma 11.** *Let $T \geq 1$, $x \in \mathcal{X}$. The matrices $\boldsymbol{K}_{[T]} + \boldsymbol{\Sigma}_{[T]}$ and $\boldsymbol{k}_{[T]}(x)(\boldsymbol{K}_{[T]} + \boldsymbol{\Sigma}_{[T]})^{-1}\boldsymbol{k}_{[T]}(x)^\mathsf{T}$ are symmetric and positive definite. In particular, $\left(\boldsymbol{k}_{[T]}(x)(\boldsymbol{K}_{[T]} + \boldsymbol{\Sigma}_{[T]})^{-1}\boldsymbol{k}_{[T]}(x))^\mathsf{T}\right)^{jj} > 0$ for each $j \in [m]$.*

*Proof.* Note that $\boldsymbol{K}_{[T]}$ is the covariance matrix of the random vector $\boldsymbol{f}_{[T]} = [\boldsymbol{f}(\tilde{x}_1)^\mathsf{T}, \ldots, \boldsymbol{f}(\tilde{x}_T)^\mathsf{T}]^\mathsf{T}$; hence, it is symmetric and positive semidefinite. Being a diagonal matrix with positive entries, $\boldsymbol{\Sigma}_{[T]}$ is symmetric and positive definite. Hence, $\boldsymbol{K}_{[T]} + \boldsymbol{\Sigma}_{[T]}$ is symmetric and positive definite; and so is $(\boldsymbol{K}_{[T]} + \boldsymbol{\Sigma}_{[T]})^{-1}$. The latter implies that $\boldsymbol{k}_{[T]}(x)(\boldsymbol{K}_{[T]} + \boldsymbol{\Sigma}_{[T]})^{-1}\boldsymbol{k}_{[T]}(x)^\mathsf{T}$ is symmetric and that

$$\boldsymbol{w}^\mathsf{T}\left(\boldsymbol{k}_{[T]}(x)(\boldsymbol{K}_{[T]} + \boldsymbol{\Sigma}_{[T]})^{-1}\boldsymbol{k}_{[T]}(x)^\mathsf{T}\right)\boldsymbol{w} = (\boldsymbol{k}_{[T]}(x)^\mathsf{T}\boldsymbol{w})^\mathsf{T}(\boldsymbol{K}_{[T]} + \boldsymbol{\Sigma}_{[T]})^{-1}(\boldsymbol{k}_{[T]}(x)^\mathsf{T}\boldsymbol{w}) > 0$$

for every $\boldsymbol{w} \in \mathbb{R}^m$ with $\boldsymbol{w} \neq 0$. Therefore, $\boldsymbol{a} = \boldsymbol{k}_{[T]}(x)(\boldsymbol{K}_{[T]} + \boldsymbol{\Sigma}_{[T]})^{-1}\boldsymbol{k}_{[T]}(x)^\mathsf{T}$ is positive definite. Let $\lambda_{(1)} > 0$ be its minimum eigenvalue. Let $j \in [m]$. By the variational characterization of minimum eigenvalue, we have

$$a^{jj} = \boldsymbol{e}_j^\mathsf{T}\boldsymbol{a}\boldsymbol{e}_j \geq \min_{\boldsymbol{w} \in \mathbb{R}^m \,:\, \|\boldsymbol{w}\|_2 = 1} \boldsymbol{w}^\mathsf{T}\boldsymbol{a}\boldsymbol{w} = \lambda_{(1)} > 0,$$

where $a^{jj} = \left(\boldsymbol{k}_{[T]}(x)(\boldsymbol{K}_{[T]} + \boldsymbol{\Sigma}_{[T]})^{-1}\boldsymbol{k}_{[T]}(x))^\mathsf{T}\right)^{jj}$ and $\boldsymbol{e}_j$ is the $j^{\text{th}}$ unit vector in $\mathbb{R}^m$. This completes the proof. ∎

Recall from equation 2 that, for each $t \geq 1$ such that $S_t \neq \emptyset$, $\overline{\omega}_t = \omega_t(x_{h_t, i_t})$ denotes the diameter of the selected node $x_{h_t, i_t} \in \mathcal{A}_t$ at round $t$.

**Lemma 12.** *Let $\delta \in (0, 1)$. We have*

$$\sum_{\tau=1}^{\tau_s} \overline{\omega}_{t_\tau} \leq \sqrt{\tau_s \left(16\beta_{\tau_s}\sigma^2 Cm\gamma_{\tau_s}\right)} \ ,$$

*where $C = \sigma^{-2}/\log(1 + \sigma^{-2})$ and $\gamma_{\tau_s}$ is the maximum information gain in $\tau_s$ evaluations as defined at the end of Section 3.*

*Proof.* Let $\tau \in [\tau_s]$ and $j \in [m]$. Note that the $\tau$th evaluation is made at round $t_\tau$ and we have $\tau_{t_\tau} = \tau - 1$. Hence, we have

$$
\begin{aligned}
&\overline{B}^j_{t_\tau}(x_{h_{t_\tau}, i_{t_\tau}}) - \underline{B}^j_{t_\tau}(x_{h_{t_\tau}, i_{t_\tau}}) \\
&= \min\{\mu^j_{\tau_{t_\tau}}(x_{h_{t_\tau}, i_{t_\tau}}) + \beta^{1/2}_{\tau_{t_\tau}} \sigma^j_{\tau_{t_\tau}}(x_{h_{t_\tau}, i_{t_\tau}}), \mu^j_{\tau_{t_\tau}}(p(x_{h_{t_\tau}, i_{t_\tau}})) + \beta^{1/2}_{\tau_{t_\tau}} \sigma^j_{\tau_{t_\tau}}(p(x_{h_{t_\tau}, i_{t_\tau}})) + V_{h_{t_\tau}-1}\} \\
&\quad - \max\{\mu^j_{\tau_{t_\tau}}(x_{h_{t_\tau}, i_{t_\tau}}) - \beta^{1/2}_{\tau_{t_\tau}} \sigma^j_{\tau_{t_\tau}}(x_{h_{t_\tau}, i_{t_\tau}}), \mu^j_{\tau_{t_\tau}}(p(x_{h_{t_\tau}, i_{t_\tau}})) - \beta^{1/2}_{\tau_{t_\tau}} \sigma^j_{\tau_{t_\tau}}(p(x_{h_{t_\tau}, i_{t_\tau}})) - V_{h_{t_\tau}-1}\} \\
&= \min\{\mu^j_{\tau-1}(x_{h_{t_\tau}, i_{t_\tau}}) + \beta^{1/2}_{\tau-1} \sigma^j_{\tau-1}(x_{h_{t_\tau}, i_{t_\tau}}), \mu^j_{\tau-1}(p(x_{h_{t_\tau}, i_{t_\tau}})) + \beta^{1/2}_{\tau-1} \sigma^j_{\tau-1}(p(x_{h_{t_\tau}, i_{t_\tau}})) + V_{h_{t_\tau}-1}\} \\
&\quad - \max\{\mu^j_{\tau-1}(x_{h_{t_\tau}, i_{t_\tau}}) - \beta^{1/2}_{\tau-1} \sigma^j_{\tau-1}(x_{h_{t_\tau}, i_{t_\tau}}), \mu^j_{\tau-1}(p(x_{h_{t_\tau}, i_{t_\tau}})) - \beta^{1/2}_{\tau-1} \sigma^j_{\tau-1}(p(x_{h_{t_\tau}, i_{t_\tau}})) - V_{h_{t_\tau}-1}\} \ .
\end{aligned}
$$

We can bound this difference in two ways, so that we can use information-type bounds and dimension-type bounds. In this result, we focus on the information-type bounds and write

$$
\begin{aligned}
&\overline{B}^j_{t_\tau}(x_{h_{t_\tau}, i_{t_\tau}}) - \underline{B}^j_{t_\tau}(x_{h_{t_\tau}, i_{t_\tau}}) \\
&\leq \mu^j_{\tau-1}(x_{h_{t_\tau}, i_{t_\tau}}) + \beta^{1/2}_{\tau-1} \sigma^j_{\tau-1}(x_{h_{t_\tau}, i_{t_\tau}}) - \mu^j_{\tau-1}(x_{h_{t_\tau}, i_{t_\tau}}) + \beta^{1/2}_{\tau-1} \sigma^j_{\tau-1}(x_{h_{t_\tau}, i_{t_\tau}}) \\
&= 2\beta^{1/2}_{\tau-1} \sigma^j_{\tau-1}(x_{h_{t_\tau}, i_{t_\tau}}) \ .
\end{aligned}
\tag{21}
$$

Since the diagonal distance of the hyper-rectangle $\boldsymbol{Q}_{t_\tau}(x_{h_{t_\tau}, i_{t_\tau}})$ is the largest distance between any two points in the hyper-rectangle, we have

$$
\begin{aligned}
\sum_{\tau=1}^{\tau_s} \overline{\omega}^2_{t_\tau} &= \sum_{\tau=1}^{\tau_s} \max_{y,y' \in \boldsymbol{R}_{t_\tau}(x_{h_{t_\tau}, i_{t_\tau}})} \|y - y'\|^2_2 \\
&\leq \sum_{\tau=1}^{\tau_s} \max_{y,y' \in \boldsymbol{Q}_{t_\tau}(x_{h_{t_\tau}, i_{t_\tau}})} \|y - y'\|^2_2 \\
&= \sum_{\tau=1}^{\tau_s} \left\| \boldsymbol{U}_{t_\tau}(x_{h_{t_\tau}, i_{t_\tau}}) - \boldsymbol{L}_{\tau_t}(x_{h_{t_\tau}, i_{t_\tau}}) \right\|^2_2 \\
&= \sum_{\tau=1}^{\tau_s} \sum_{j=1}^m \left( U^j_{t_\tau}(x_{h_{t_\tau}, i_{t_\tau}}) - L^j_{t_\tau}(x_{h_{t_\tau}, i_{t_\tau}}) \right)^2 \\
&= \sum_{\tau=1}^{\tau_s} \sum_{j=1}^m \left( \overline{B}^j_{t_\tau}(x_{h_{t_\tau}, i_{t_\tau}}) - \underline{B}^j_{t_\tau}(x_{h_{t_\tau}, i_{t_\tau}}) + 2V_{h_{t_\tau}} \right)^2 \tag{22} \\
&\leq \sum_{\tau=1}^{\tau_s} \sum_{j=1}^m \left( 2\beta^{1/2}_{\tau-1} \sigma^j_{\tau-1}(x_{h_{t_\tau}, i_{t_\tau}}) + 2V_{h_{t_\tau}} \right)^2 \tag{23} \\
&= 4 \sum_{\tau=1}^{\tau_s} \left( \sum_{j=1}^m \beta_{\tau-1}(\sigma^j_{\tau-1}(x_{h_{t_\tau}, i_{t_\tau}}))^2 + 2\sum_{j=1}^m V_{h_{t_\tau}} \beta^{1/2}_{\tau-1} \sigma^j_{\tau-1}(x_{h_{t_\tau}, i_{t_\tau}}) + \sum_{j=1}^m (V_{h_{t_\tau}})^2 \right) \\
&\leq 4 \sum_{\tau=1}^{\tau_s} \left( \sum_{j=1}^m \beta_{\tau-1}(\sigma^j_{\tau-1}(x_{h_{t_\tau}, i_{t_\tau}}))^2 \right. \\
&\quad \left. + 2\left( \sum_{j=1}^m (V_{h_{t_\tau}})^2 \right)^{1/2} \left( \sum_{j=1}^m \beta_{\tau-1}(\sigma^j_{\tau-1}(x_{h_{t_\tau}, i_{t_\tau}}))^2 \right)^{1/2} + \sum_{j=1}^m (V_{h_{t_\tau}})^2 \right) \tag{24} \\
&\leq 4 \sum_{\tau=1}^{\tau_s} \left( \sum_{j=1}^m \beta_{\tau-1}(\sigma^j_{\tau-1}(x_{h_{t_\tau}, i_{t_\tau}}))^2 + 2\sum_{j=1}^m \beta_{\tau-1}(\sigma^j_{\tau-1}(x_{h_{t_\tau}, i_{t_\tau}}))^2 \right. \\
&\quad \left. + \sum_{j=1}^m \beta_{\tau-1}(\sigma^j_{\tau-1}(x_{h_{t_\tau}, i_{t_\tau}}))^2 \right) \tag{25}
\end{aligned}
$$

$$\leq 16\beta_{\tau_s} \sum_{\tau=1}^{\tau_s} \sum_{j=1}^{m} (\sigma_{\tau-1}^j(x_{h_{t_\tau},i_{t_\tau}}))^2 \tag{26}$$

$$= 16\beta_{\tau_s}\sigma^2 \sum_{\tau=1}^{\tau_s} \sum_{j=1}^{m} \sigma^{-2}(\sigma_{\tau-1}^j(x_{h_{t_\tau},i_{t_\tau}}))^2$$

$$\leq 16\beta_{\tau_s}\sigma^2 C \left( \sum_{\tau=1}^{\tau_s} \sum_{j=1}^{m} \log(1 + \sigma^{-2}(\sigma_{\tau-1}^j(x_{h_{t_\tau},i_{t_\tau}}))^2) \right) \tag{27}$$

$$\leq 16\beta_{\tau_s}\sigma^2 C m I(\boldsymbol{y}_{[\tau_s]}, \boldsymbol{f}_{[\tau_s]}) \tag{28}$$

$$\leq 16\beta_{\tau_s}\sigma^2 C m \gamma_{\tau_s}, \tag{29}$$

where $C = \sigma^{-2}/\log(1+\sigma^{-2})$. In this calculation, equation 22 follows by definitions; equation 23 follows by equation 21; equation 24 follows from Cauchy-Schwarz inequality; equation 25 follows from the fact that we make evaluation at round $t_\tau$ so that we have

$$\beta_{\tau-1}^{1/2} \left\| \boldsymbol{\sigma}_{\tau-1}(x_{h_{t_\tau},i_{t_\tau}}) \right\|_2 = \beta_{\tau_{t_\tau}}^{1/2} \left\| \boldsymbol{\sigma}_{\tau_{t_\tau}}(x_{h_{t_\tau},i_{t_\tau}}) \right\|_2 > \left\| \boldsymbol{V}_{h_{t_\tau}} \right\|_2$$

by the structure of the algorithm; equation 26 holds since $\beta_\tau$ is monotonically non-decreasing in $\tau$ (see the definition of $\beta_\tau$ in Theorem 1); equation 27 follows from the fact that $s \leq C\log(1+s)$ for all $0 \leq s \leq \sigma^{-2}$ and that we have

$$\sigma^{-2}(\sigma_{\tau-1}^j(x_{h_{t_\tau},i_{t_\tau}}))^2$$
$$= \sigma^{-2}\left( k^{jj}(x_{h_{t_\tau},i_{t_\tau}}, x_{h_{t_\tau},i_{t_\tau}}) - \left( \boldsymbol{k}_{[\tau-1]}(x_{h_{t_\tau},i_{t_\tau}})(\boldsymbol{K}_{[\tau-1]} + \boldsymbol{\Sigma}_{[\tau-1]})^{-1}\boldsymbol{k}_{[\tau-1]}(x_{h_{t_\tau},i_{t_\tau}})^{\tau_s})^{jj} \right) \right)$$
$$\leq \sigma^{-2}k^{jj}(x_{h_t,i_t}, x_{h_t,i_t})$$
$$\leq \sigma^{-2}$$

thanks to Lemma 11 and Assumption 1; equation 28 follows from Proposition 2; and equation 29 follows from definition of the maximum information gain.

Finally, by Cauchy-Schwarz inequality, we have

$$\sum_{\tau=1}^{\tau_s} \overline{\omega}_{t_\tau} \leq \sqrt{\tau_s \sum_{\tau=1}^{\tau_s} \overline{\omega}_{t_\tau}^2} \leq \sqrt{\tau_s \left( 16\beta_{\tau_s}\sigma^2 C m \gamma_{\tau_s} \right)} ,$$

which completes the proof. ∎

**Lemma 13.** *Running Adaptive $\boldsymbol{\epsilon}$-PAL with $(\beta_\tau)_{\tau \in \mathbb{N}}$ as defined in Lemma 6, we have*

$$\overline{\omega}_{t_s} \leq \sqrt{\frac{16\beta_{\tau_s}\sigma^2 C m \gamma_{\tau_s}}{\tau_s}} .$$

*Proof.* First we show that the sequence $(\overline{\omega}_t)_{t \in \mathbb{N}}$ is a monotonically non-increasing sequence. To that end, note that by the principle of selection we have that $\omega_{t-1}(x_{h_t,i_t}) \leq \overline{\omega}_{t-1}$. On the other hand, for every $x \in \mathcal{X}$, we have $\omega_t(x) \leq \omega_{t-1}(x)$ since $\boldsymbol{R}_t(x) \subseteq \boldsymbol{R}_{t-1}(x)$. Thus, $\overline{\omega}_t = \omega_t(x_{h_t,i_t}) \leq \omega_{t-1}(x_{h_t,i_t})$. So we obtain $\overline{\omega}_t \leq \overline{\omega}_{t-1}$.

By above and by Lemma 12, we have

$$\overline{\omega}_{t_s} \leq \frac{\sum_{\tau=1}^{\tau_s} \overline{\omega}_{t_\tau}}{\tau_s} \leq \sqrt{\frac{16\beta_{\tau_s}\sigma^2 C m \gamma_{\tau_s}}{\tau_s}} .$$

∎

Finally, we are ready to state the information-type bound on the sample complexity.

**Proposition 4.** *Let $\boldsymbol{\epsilon} = [\epsilon^1, \ldots, \epsilon^m]^\mathsf{T}$ be given with $\epsilon = \min_{j \in [m]} \epsilon^j > 0$. Let $\delta \in (0,1)$ and $\bar{D} > D_1$. For each $h \geq 0$, let $V_h$ be defined as in Corollary 1; for each $\tau \in \mathbb{N}$, let $\beta_\tau$ be defined as in Lemma 6. When we run Adaptive $\boldsymbol{\epsilon}$-PAL with prior $GP(0, \boldsymbol{k})$ and noise $\mathcal{N}(0, \sigma^2)$, the following holds with probability at least $1 - \delta$. An $\boldsymbol{\epsilon}$-accurate Pareto set can be found with at most $T$ function evaluations, where $T$ is the smallest natural number satisfying*

$$\sqrt{\frac{16\beta_T \sigma^2 C m \gamma_T}{T}} < \epsilon \ .$$

*In the above expression, $C$ represents the constant defined in Lemma 12.*

*Proof.* According to Lemma 3, we have $\overline{w}_{t_s} < \epsilon$. In addition, Lemma 13 says that

$$\omega_{\tau_s} \leq \frac{\sum_{\tau=1}^{\tau_s} \overline{w}_{t_\tau}}{\tau_s} \leq \sqrt{\frac{16\beta_{\tau_s} \sigma^2 C m \gamma_{\tau_s}}{\tau_s}} \ .$$

We use these two facts to find an upper bound on $\tau_s$ that holds with probability one. Let

$$T = \min \left\{ \tau \in \mathbb{N} : \sqrt{\frac{16\beta_\tau \sigma^2 C m \gamma_\tau}{\tau}} < \epsilon \right\} \ .$$

Since the event $\{\tau_s = T\}$ implies that $\{\overline{w}_{t_s} < \epsilon\}$, we have $\Pr(\tau_s > T) = 0$. ∎

### B.7 Derivation of the dimension-type sample complexity bound

In order to bound the diameter, we make use of the following observation.

**Remark 4.** *We have*

$$\frac{V_h}{V_{h+1}} \leq N_1 := \rho^{-\alpha} \ .$$

The next lemma bounds diameters of confidence hyper-rectangles of the nodes in $\mathcal{A}_t$ for all rounds $t$.

**Lemma 14.** *In any round $t \geq 1$ before Adaptive $\boldsymbol{\epsilon}$-PAL terminates, we have*

$$\overline{\omega}_t^2 \leq L V_{h_t}^2 \ ,$$

*where $L = m \left( 4N_1^2 + 4N_1^2(2N_1 + 2) + (2N_1 + 2)^2 \right)$.*

*Proof.* We have

$$\overline{\omega}_t^2 = \omega_t^2(x_{h_t, i_t}) = \left( \max_{y, y' \in \boldsymbol{R}_t(x_{h_t, i_t})} \|y - y'\|_2 \right)^2$$

$$\leq \left( \max_{y, y' \in \boldsymbol{Q}_t(x_{h_t, i_t})} \|y - y'\|_2 \right)^2 = \left( \|\boldsymbol{U}_t(x_{h_t, i_t}) - \boldsymbol{L}_t(x_{h_t, i_t})\|_2 \right)^2 = \sum_{j=1}^m \left( \overline{B}_t^j(x_{h_t, i_t}) - \underline{B}_t^j(x_{h_t, i_t}) + 2V_{h_t} \right)^2$$

$$\leq \sum_{j=1}^m \left( 2\beta_{\tau_t}^{1/2} \sigma_{\tau_t}^j(p(x_{h_t, i_t})) + (2N_1 + 2)V_{h_t} \right)^2 \tag{30}$$

$$= 4\beta_{\tau_t} \sum_{j=1}^m \left( \sigma_{\tau_t}^j(p(x_{h_t, i_t})) \right)^2 + 4(2N_1 + 2) \sum_{j=1}^m \beta_{\tau_t}^{1/2} \sigma_{\tau_t}^j(p(x_{h_t, i_t}))V_{h_t} + (2N_1 + 2)^2 \sum_{j=1}^m \left( V_{h_t} \right)^2$$

$$\leq 4 \sum_{j=1}^m \left( V_{h_t - 1} \right)^2 + 4(2N_1 + 2) \sum_{j=1}^m \beta_{\tau_t}^{1/2} \sigma_{\tau_t}^j(p(x_{h_t, i_t}))V_{h_t} + (2N_1 + 2)^2 \sum_{j=1}^m \left( V_{h_t} \right)^2 \tag{31}$$

$$\leq 4 \sum_{j=1}^m \left( V_{h_t - 1} \right)^2 + 4(2N_1 + 2) \left( \sum_{j=1}^m \beta_{\tau_t} \left( \sigma_{\tau_t}^j(p(x_{h_t, i_t})) \right)^2 \right)^{\frac{1}{2}} \left( \sum_{j=1}^m (V_{h_t})^2 \right)^{\frac{1}{2}} + (2N_1 + 2)^2 \sum_{j=1}^m \left( V_{h_t} \right)^2 \tag{32}$$

$$\leq 4\sum_{j=1}^{m}(V_{h_t-1})^2 + 4(2N_1+2)N_1\left(\sum_{j=1}^{m}(V_{h_t-1})^2\right)^{\frac{1}{2}}\left(\sum_{j=1}^{m}(V_{h_t})^2\right)^{\frac{1}{2}} + (2N_1+2)^2\sum_{j=1}^{m}(V_{h_t})^2 \tag{33}$$

$$\leq m\left(4N_1^2 + 4N_1^2(2N_1+2) + (2N_1+2)^2\right)(V_{h_t})^2 = LV_{h_t}^2 \ . \tag{34}$$

In the above expression, equation 30 follows from the fact that for $j \in [m]$ and $t \geq 1$

$$\overline{B}_t^j(x_{h_t,i_t}) - \underline{B}_t^j(x_{h_t,i_t})$$
$$= \min\{\mu_{\tau_t}^j(x_{h_t,i_t}) + \beta_{\tau_t}^{1/2}\sigma_{\tau_t}^j(x_{h_t,i_t}),\ \mu_{\tau_t}^j(p(x_{h_t,i_t})) + \beta_{\tau_t}^{1/2}\sigma_{\tau_t}^j(p(x_{h_t,i_t})) + V_{h_t-1}\}$$
$$\quad - \max\{\mu_{\tau_t}^j(x_{h_t,i_t}) - \beta_{\tau_t}^{1/2}\sigma_{\tau_t}^j(x_{h_t,i_t}),\ \mu_{\tau_t}^j(p(x_{h_t,i_t})) - \beta_{\tau_t}^{1/2}\sigma_{\tau_t}^j(p(x_{h_t,i_t})) - V_{h_t-1}\}$$
$$\leq \mu_{\tau_t}^j(p(x_{h_t,i_t})) + \beta_{\tau_t}^{1/2}\sigma_{\tau_t}^j(p(x_{h_t,i_t})) + V_{h_t-1} - \mu_{\tau_t}^j(p(x_{h_t,i_t})) + \beta_{\tau_t}^{1/2}\sigma_{\tau_t}^j(p(x_{h_t,i_t})) + V_{h_t-1}$$
$$\leq 2\beta_{\tau_t}^{1/2}\sigma_{\tau_t}^j(p(x_{h_t,i_t})) + 2V_{h_t-1}$$
$$\leq 2\beta_{\tau_t}^{1/2}\sigma_{\tau_t}^j(p(x_{h_t,i_t})) + 2N_1V_{h_t} \ , \tag{35}$$

where equation 35 follows from Remark 4. equation 31 is obtained by observing that the node $p(x_{h_t,i_t})$ has been refined, and hence, it holds that $\beta_{\tau_t}^{1/2}\|\sigma_{\tau_t}(p(x_{h_t,i_t}))\|_2 \leq \|V_{h_t-1}\|_2$. equation 32 follows from application of the Cauchy-Schwarz inequality. equation 33 follows again from the fact that $\beta_{\tau_t}^{1/2}\|\sigma_{\tau_t}(p(x_{h_t,i_t}))\|_2 \leq \|V_{h_t-1}\|_2$, and equation 34 follows from Remark 4. ∎

Let $\mathcal{E}$ denote the set of rounds in which Adaptive $\epsilon$-PAL performs design evaluations. Note that $|\mathcal{E}| = \tau_s$. Let $t_s$ denote the round in which the algorithm terminates. Next, we use Lemma 1 together with Lemma 14 to upper bound $\overline{\omega}_{t_s}$ as a function of the number evaluations until termination.

**Lemma 15.** *Let $\delta \in (0,1)$ and $\overline{D} > D_1$. Running Adaptive $\epsilon$-PAL with $\beta_\tau$ defined in Lemma 6, there exists a constant $Q > 0$ such that the following event holds almost surely.*

$$\overline{\omega}_{t_s} \leq K_1 \tau_s^{\frac{-\alpha}{\overline{D}+2\alpha}}(\log\tau_s)^{\frac{-(\overline{D}+\alpha)}{\overline{D}+2\alpha}} + K_2\tau_s^{\frac{-\alpha}{\overline{D}+2\alpha}}(\log\tau_s)^{\frac{\alpha}{\overline{D}+2\alpha}} \ ,$$

*where*

$$K_1 = \frac{\sqrt{L}Q\sigma^2\beta_{\tau_s}}{C_{\boldsymbol{k}}v_1^\alpha v_2^{\overline{D}}(\rho^{-(\overline{D}+\alpha)}-1)} \ ,$$

$$K_2 = 4\sqrt{L}C_{\boldsymbol{k}}v_1^\alpha\left(\sqrt{C_2 + 2\log(2H^2\pi^2m/6\delta)} + H\log N + \max\{0, -4(D_1/\alpha)\log(C_{\boldsymbol{k}}(v_1\rho^H)^\alpha)\} + C_3\right) \ ,$$

$$H = \left\lfloor\frac{\log\tau_s - \log(\log\tau_s)}{\log(1/\rho)(\overline{D}+2\alpha)}\right\rfloor \ .$$

*Proof.* We have

$$\overline{\omega}_{t_s} \leq \frac{\sum_{t\in\mathcal{E}}\overline{\omega}_t}{\tau_s} \leq \frac{\sqrt{L}\sum_{t\in\mathcal{E}}V_{h_t}}{\tau_s} \ , \tag{36}$$

where the first inequality follows from the fact that $\overline{\omega}_t \leq \overline{\omega}_{t-1}$ for all $t \geq 1$ and the second inequality is the result of Lemma 14. We will bound $\sum_{t\in\mathcal{E}}V_{h_t}$ and use it to obtain a bound for equation 36. First, we define

$$S_1 = \sum_{\substack{t\in\mathcal{E}:\\h_t<H}}V_{h_t} \text{ and } S_2 = \sum_{\substack{t\in\mathcal{E}:\\h_t\geq H}}V_{h_t} \ ,$$

and write $\sum_{t\in\mathcal{E}}V_{h_t} = S_1 + S_2$. We have

$$S_1 = \sum_{\substack{t\geq 1:\\h_t<H}}V_{h_t}\mathbb{I}(t\in\mathcal{E}) = \sum_{t\geq 1}\sum_{h<H}V_{h_t}\mathbb{I}(h_t=h)\mathbb{I}(t\in\mathcal{E})$$

$$= \sum_{h<H} \sum_{t\geq 1} V_{h_t} \mathbb{I}(h_t = h)\mathbb{I}(t \in \mathcal{E})$$

$$\leq \sum_{h<H} V_h Q(v_2 \rho^h)^{-\bar{D}} q_h \tag{37}$$

$$\leq \sum_{h<H} V_h Q(v_2 \rho^h)^{-\bar{D}} \frac{\sigma^2 \beta_T}{V_h^2} \tag{38}$$

$$\leq \sum_{h<H} Q(v_2 \rho^h)^{-\bar{D}} \frac{\sigma^2 \beta_T}{C_{\boldsymbol{k}}(v_1 \rho^h)^\alpha} \tag{39}$$

$$\leq \frac{Q\sigma^2 \beta_T}{C_{\boldsymbol{k}} v_1^\alpha v_2^{\bar{D}}} \sum_{h<H} \rho^{-(\bar{D}+\alpha)h} \leq \frac{Q\sigma^2 \beta_T}{C_{\boldsymbol{k}} v_1^\alpha v_2^{\bar{D}}} \frac{\rho^{-(\bar{D}+\alpha)H}}{\rho^{-(\bar{D}+\alpha)} - 1} \ . \tag{40}$$

In the above display, to obtain equation 37, we note that for a fixed $h$ the cells $X_{h,i}$ are disjoint, a ball of radius $v_2 \rho^h$ should be able to fit in each cell, and thus, the number of depth $h$ cells is upper bounded by the number of radius $v_2 \rho^h$ balls we can pack in $(\mathcal{X}, d)$, which is in turn upper bounded by $M(\mathcal{X}, 2v_2 \rho^h, d)$. The rest follows from Lemma 1, which states that there exists a positive constant $Q$, such that $M(\mathcal{X}, 2v_2 \rho^h, d) \leq N(\mathcal{X}, v_2 \rho^h, d) \leq Q(v_2 \rho^h)^{-\bar{D}}$. For equation 38, we upper bound $q_h$ using Lemma 5, and equation 39 follows by observing that $V_h \geq C_{\boldsymbol{k}}(v_1 \rho^h)^\alpha$.

Since $V_h$ is decreasing in $h$, the remainder of the sum can be bounded as

$$S_2 = \sum_{\substack{t\geq 1: \\ h_t \geq H}} V_{h_t} \mathbb{I}(t \in \mathcal{E}) \leq \sum_{t \in \mathcal{E}} V_H = \tau_s V_H = (K_2/\sqrt{L})\tau_s \rho^{H\alpha} \ . \tag{41}$$

Combining equation 40 and equation 41, we obtain

$$\sum_{t \in \mathcal{E}} V_{h_t} \leq \frac{Q\sigma^2 \beta_{\tau_s}}{C_{\boldsymbol{k}} v_1^\alpha v_2^{\bar{D}}} \frac{\rho^{-(\bar{D}+\alpha)H}}{\rho^{-(\bar{D}+\alpha)} - 1} + (K_2/\sqrt{L})\tau_s \rho^{H\alpha} \ . \tag{42}$$

Since

$$H = \left\lfloor \frac{\log \tau_s - \log(\log \tau_s)}{\log(1/\rho)(\bar{D} + 2\alpha)} \right\rfloor = \left\lfloor -\log_\rho \left( \frac{\tau_s}{\log \tau_s} \right)^{\frac{1}{\bar{D}+2\alpha}} \right\rfloor,$$

we have

$$\rho^{-H(\bar{D}+2\alpha)} \leq \tau_s^{\frac{\bar{D}+\alpha}{\bar{D}+2\alpha}} (\log \tau_s)^{\frac{-(\bar{D}+\alpha)}{\bar{D}+2\alpha}} \text{ and } \tau_s \rho^{H\alpha} \leq \rho^\alpha \tau_s^{1-\frac{\alpha}{\bar{D}+2\alpha}} (\log \tau_s)^{\frac{\alpha}{\bar{D}+2\alpha}} \ .$$

Finally, we use the values found above to upper bound equation 42, and then use this upper bound in equation 36, which gives us

$$\overline{\omega}_{t_s} \leq \sqrt{L}\left( \frac{Q\sigma^2 \beta_{\tau_s}}{C_{\boldsymbol{k}} v_1^\alpha v_2^{\bar{D}}(\rho^{-(\bar{D}+\alpha)} - 1)} \tau_s^{\frac{-\alpha}{\bar{D}+2\alpha}} (\log \tau_s)^{\frac{-(\bar{D}+\alpha)}{\bar{D}+2\alpha}} + (K_2/\sqrt{L})\tau_s^{\frac{-\alpha}{\bar{D}+2\alpha}} (\log \tau_s)^{\frac{\alpha}{\bar{D}+2\alpha}} \right) \ .$$

$\blacksquare$

Finally, we are ready to state the metric dimension-type bound on the sample complexity.

**Proposition 5.** *Let $\boldsymbol{\epsilon} = [\epsilon^1, \ldots, \epsilon^m]^\mathsf{T}$ be given with $\epsilon = \min_{j \in [m]} \epsilon^j > 0$. Let $\delta \in (0, 1)$ and $\bar{D} > D_1$. For each $h \geq 0$, let $V_h$ be defined as in Corollary 1; for each $\tau \in \mathbb{N}$, let $\beta_\tau$ be defined as in Lemma 6. When we run Adaptive $\boldsymbol{\epsilon}$-PAL with prior $GP(0, \boldsymbol{k})$ and noise $\mathcal{N}(0, \sigma^2)$, the following holds with probability at least $1 - \delta$.*

*An $\boldsymbol{\epsilon}$-accurate Pareto set can be found with at most $T$ function evaluations, where $T$ is the smallest natural number satisfying*

$$K_1 T^{\frac{-\alpha}{\bar{D}+2\alpha}} (\log T)^{\frac{-(\bar{D}+\alpha)}{\bar{D}+2\alpha}} + K_2 T^{\frac{-\alpha}{\bar{D}+2\alpha}} (\log T)^{\frac{\alpha}{\bar{D}+2\alpha}} < \epsilon \ ,$$

*where $K_1$ and $K_2$ are the constants defined in Lemma 15.*

*Proof.* According to Lemma 3, we have $\overline{w}_{t_s} < \epsilon$. In addition, Lemma 15 says that

$$\overline{\omega}_{t_s} \leq K_1 \tau_s^{\frac{-\alpha}{\bar{D}+2\alpha}} \left(\log \tau_s\right)^{\frac{-(\bar{D}+\alpha)}{\bar{D}+2\alpha}} + K_2 \tau_s^{\frac{-\alpha}{\bar{D}+2\alpha}} \left(\log \tau_s\right)^{\frac{\alpha}{\bar{D}+2\alpha}} .$$

We use these two facts to find an upper bound on $\tau_s$ that holds with probability one. Let

$$T = \min\left\{\tau \in \mathbb{N} : K_1 \tau^{\frac{-\alpha}{\bar{D}+2\alpha}} \left(\log \tau\right)^{\frac{-(\bar{D}+\alpha)}{\bar{D}+2\alpha}} + K_2 \tau^{\frac{-\alpha}{\bar{D}+2\alpha}} \left(\log \tau\right)^{\frac{\alpha}{\bar{D}+2\alpha}} < \epsilon\right\} .$$

Since the event $\{\tau_s = T\}$ implies that $\{\overline{w}_{t_s} < \epsilon\}$, we have $\Pr(\tau_s > T) = 0$. ∎

### B.8 The final step of the proof of Theorem 1

We take the minimum over the two bounds presented in Proposition 4 and Proposition 5 to obtain the result in Theorem 1. ∎

### B.9 Proof of Proposition 2

First, let us review some well-known facts about entropies. For an $m$-dimensional Gaussian random vector $\boldsymbol{g}$ with distribution $\mathcal{N}(\boldsymbol{a}, \boldsymbol{b})$ with $\boldsymbol{a} \in \mathbb{R}^m, \boldsymbol{b} \in \mathbb{R}^{m \times m}$, the entropy of $\boldsymbol{g}$ is calculated by

$$H(\boldsymbol{g}) = H(\mathcal{N}(\boldsymbol{a}, \boldsymbol{b})) = \frac{1}{2}\log|2\pi e \boldsymbol{b}|.$$

More generally, if $h$ is another random variable (with arbitrary measurable state space $H$) and the regular conditional distribution of $\boldsymbol{g}$ given $h$ is $\mathcal{N}(\boldsymbol{a}(h), \boldsymbol{b}(h))$ for some measurable functions $a \colon H \to \mathbb{R}^m$ and $b \colon H \to \mathbb{R}^{m \times m}$, then the conditional entropy of $\boldsymbol{g}$ given $h$ is calculated by

$$H(\boldsymbol{g}|h) = \frac{1}{2}\mathbb{E}\left[\log|2\pi e \boldsymbol{b}(h)|\right].$$

**Lemma 16.** *We have*

$$I(\boldsymbol{y}_{[T]}; \boldsymbol{f}_{[T]}) = \sum_{\tau=1}^{T} \frac{1}{2}\log|\boldsymbol{I}_m + \sigma^{-2}\boldsymbol{k}_{\tau-1}(\tilde{x}_\tau, \tilde{x}_\tau)|,$$

*where $\boldsymbol{k}_0(\tilde{x}_1, \tilde{x}_1) = \boldsymbol{k}(\tilde{x}_1, \tilde{x}_1)$.*

*Proof.* We have

$$\begin{aligned}
I(\boldsymbol{y}_{[T]}; \boldsymbol{f}_{[T]}) &= H(\boldsymbol{y}_{[T]}) - H(\boldsymbol{y}_{[T]}|\boldsymbol{f}_{[T]}) \\
&= H(\boldsymbol{y}_T, \boldsymbol{y}_{[T-1]}) - H(\boldsymbol{y}_{[T]}|\boldsymbol{f}_{[T]}) \\
&= H(\boldsymbol{y}_T|\boldsymbol{y}_{[T-1]}) + H(\boldsymbol{y}_{[T-1]}) - H(\boldsymbol{y}_{[T]}|\boldsymbol{f}_{[T]}).
\end{aligned}$$

Re-iterating this calculation inductively, we obtain

$$I(\boldsymbol{y}_{[T]}; \boldsymbol{f}_{[T]}) = \sum_{\tau=2}^{T} H(\boldsymbol{y}_\tau|\boldsymbol{y}_{[\tau-1]}) + H(\boldsymbol{y}_1) - H(\boldsymbol{y}_{[T]}|\boldsymbol{f}_{[T]})$$

since $\boldsymbol{y}_{[1]} = \boldsymbol{y}_1$. Note that

$$H(\boldsymbol{y}_{[T]}|\boldsymbol{f}_{[T]}) = H(\boldsymbol{f}(\tilde{x}_1) + \boldsymbol{\epsilon}_1, \ldots, \boldsymbol{f}(\tilde{x}_T) + \boldsymbol{\epsilon}_T|\boldsymbol{f}_{[T]}) = \sum_{\tau=1}^{T} H(\boldsymbol{\epsilon}_\tau) = \frac{T}{2}\log|2\pi e \sigma^2 \boldsymbol{I}_m|.$$

On the other hand, the conditional distribution of $\boldsymbol{y}_\tau$ given $\boldsymbol{y}_{[\tau-1]}$ is $\mathcal{N}(\boldsymbol{\mu}_{\tau-1}(\tilde{x}_\tau), \boldsymbol{k}_{\tau-1}(\tilde{x}_\tau, \tilde{x}_\tau) + \sigma^2 \boldsymbol{I}_m)$ and the distribution of $\boldsymbol{y}_1$ is $\mathcal{N}(0, \boldsymbol{k}(\tilde{x}_1, \tilde{x}_1) + \sigma^2 \boldsymbol{I}_m)$. Hence,

$$I(\boldsymbol{y}_{[T]}; \boldsymbol{f}_{[T]}) = \sum_{\tau=2}^{T} H(\boldsymbol{y}_\tau|\boldsymbol{y}_{[\tau-1]}) + H(\boldsymbol{y}_1) - H(\boldsymbol{y}_{[T]}|\boldsymbol{f}_{[T]})$$

$$= \sum_{\tau=2}^{T} \frac{1}{2} \left[ \log |2\pi e (\boldsymbol{k}_{\tau-1}(\tilde{x}_\tau, \tilde{x}_\tau) + \sigma^2 \boldsymbol{I}_m)| \right] + \frac{1}{2} \log |2\pi e (\boldsymbol{k}(\tilde{x}_\tau, \tilde{x}_\tau) + \sigma^2 \boldsymbol{I}_m)| - \frac{T}{2} \log |2\pi e \sigma^2 \boldsymbol{I}_m|$$

$$= \sum_{\tau=1}^{T} \frac{1}{2} \log |2\pi e (\boldsymbol{k}_{\tau-1}(\tilde{x}_\tau, \tilde{x}_\tau) + \sigma^2 \boldsymbol{I}_m)| - \frac{T}{2} \log |2\pi e \sigma^2 \boldsymbol{I}_m|$$

$$= \sum_{\tau=1}^{T} \frac{1}{2} \log |2\pi e \sigma^2 \boldsymbol{I}_m (\sigma^{-2} \boldsymbol{k}_{\tau-1}(\tilde{x}_\tau, \tilde{x}_\tau) + \boldsymbol{I}_m)| - \frac{T}{2} \log |2\pi e \sigma^2 \boldsymbol{I}_m|$$

$$= \sum_{\tau=1}^{T} \frac{1}{2} \log |2\pi e \sigma^2 \boldsymbol{I}_m| + \sum_{\tau=1}^{T} \frac{1}{2} \log |\sigma^{-2} \boldsymbol{k}_{\tau-1}(\tilde{x}_\tau, \tilde{x}_\tau) + \boldsymbol{I}_m| - \frac{T}{2} \log |2\pi e \sigma^2 \boldsymbol{I}_m|$$

$$= \sum_{\tau=1}^{T} \frac{1}{2} \log |\sigma^{-2} \boldsymbol{k}_{\tau-1}(\tilde{x}_\tau, \tilde{x}_\tau) + \boldsymbol{I}_m|.$$

∎

**Lemma 17.** *Let $\boldsymbol{a} = (a_{ij})_{i,j \in [m]}$ be a symmetric positive definite $m \times m$-matrix. Then,*

$$|\boldsymbol{a} + \boldsymbol{I}_m| \geq \max_{j \in [m]} (1 + a_{jj}).$$

*Proof.* Let $0 < \lambda_{(1)} \leq \ldots \leq \lambda_{(m)}$ be the ordered eigenvalues of $\boldsymbol{a}$. It is easy to check that $\lambda \in \mathbb{R}$ is an eigenvalue of $\boldsymbol{a}$ if and only if $1 + \lambda$ is an eigenvalue of $\boldsymbol{a} + \boldsymbol{I}_m$. In particular, $1 + \lambda_{(m)}$ is the largest eigenvalue of $\boldsymbol{a} + \boldsymbol{I}_m$ so that

$$|\boldsymbol{a} + \boldsymbol{I}_m| = \prod_{j \in [m]} (1 + \lambda_{(j)}) = \left( \prod_{j \in [m-1]} (1 + \lambda_{(j)}) \right) (1 + \lambda_{(m)}) \geq (1 + \lambda_{(m)}). \tag{43}$$

On the other hand, thanks to the variational characterization of the maximum eigenvalue $\lambda_{(m)}$, we have

$$\lambda_{(m)} = \max_{\boldsymbol{x} \in \mathbb{R}^m : \|x\|=1} \boldsymbol{x}^\mathsf{T} \boldsymbol{a} \boldsymbol{x} \geq \boldsymbol{e}_j^\mathsf{T} \boldsymbol{a} \boldsymbol{e}_j = a_{jj} \tag{44}$$

for each $j \in [m]$, where $\boldsymbol{e}_j$ is the $j^\text{th}$ unit vector in $\mathbb{R}^m$. The claim of the lemma follows by combining equation 43 and equation 44. ∎

Next, we will make use of the lemmas derived above to complete the proof. Note that

$$I(\boldsymbol{y}_{[T]}; \boldsymbol{f}_{[T]}) = \sum_{\tau=1}^{T} \frac{1}{2} \log |\boldsymbol{I}_m + \sigma^{-2} \boldsymbol{k}_{\tau-1}(\tilde{x}_\tau, \tilde{x}_\tau)|$$

$$\geq \sum_{\tau=1}^{T} \frac{1}{2} \max_{j \in [m]} \log(1 + \sigma^{-2} k_{\tau-1}^{jj}(\tilde{x}_\tau, \tilde{x}_\tau))$$

$$= \sum_{\tau=1}^{T} \frac{1}{2} \max_{j \in [m]} \log(1 + \sigma^{-2} (\sigma_{\tau-1}^j(\tilde{x}_\tau))^2)$$

$$\geq \sum_{\tau=1}^{T} \sum_{j \in [m]} \frac{1}{2m} \log(1 + \sigma^{-2} (\sigma_{\tau-1}^j(\tilde{x}_\tau))^2),$$

where the first equality is by Lemma 16 and the first inequality is by Lemma 17. Hence, the claim of the proposition follows.

### B.10 Determination of $\alpha$ and $C_K$ for Squared Exponential type kernels

For Squared Exponential type kernels, the covariance function of objective $j$ can be written as:

$$k^{jj}(x,y) = k^j(r) = \nu e^{-\frac{r^2}{L^2}} \ ,$$

where $r = \|x - y\|$ and $L$ and $\nu$ are the length-scale and variance hyper-parameters of the $j^{\text{th}}$ objective. Hence, by Remark 1, the metric $l_j$ induced by the $j^{\text{th}}$ objective of the GP on $\mathcal{X}$ is given by:

$$l_j(x,y) = \sqrt{2k(0) - 2k(0)e^{-\frac{r^2}{L^2}}} = \sqrt{2\nu(1 - e^{-\frac{r^2}{L^2}})} \leq \frac{\sqrt{2\nu}}{L} r \ .$$

Thus, we can set $C_{\boldsymbol{k}} = \frac{\sqrt{2\nu}}{L}$ and $\alpha = 1$ in Assumption 1.

## C The proof of Proposition 1

In this section, we provide a constructive proof of Proposition 1, which is an extension of Example 1 in Shekhar et al. (2018) to the multi-output GP setting. We restate the proposition here for convenience.

**Statement.** *There exists a multi-output GP $\boldsymbol{f}$, with covariance function satisfying Assumption 1 and a sequence of $T$ noisy observations made on $\boldsymbol{f}$, such that we have*

$$I(\boldsymbol{y}_{[T]}, \boldsymbol{f}_{[T]}) \geq \Omega(T) \ .$$

*Proof.* We provide a constructive proof, which is an extension of Example 1 in Shekhar et al. (2018) to the multi-output GP setting. First, let $\mathcal{X} = [0,1]$. Let $j \in [m]$, $x \in \mathcal{X}$, and let us define

$$\tilde{f}^j(x) = \sum_{i=1}^{\infty} 4\bar{a}_{ij} X_{ij} \left((3^i x - 1)(2 - 3^i x)\mathbb{I}(x \in L_i) - (3^i x - 2)(3 - 3^i x)\mathbb{I}(x \in R_i)\right) \ ,$$

where, for each $j \in [m]$, $(\bar{a}_{ij})_{i=1}^{\infty}$ is a non-increasing sequence of positive real numbers with $\bar{a}_{1j} \leq 1$; $(X_{ij})_{i=1,j=1}^{\infty,m}$ is a sequence of independent and identically distributed standard Gaussian random variables; and we let $L_i := [3^{-i}, 2 \cdot 3^{-i})$ for all $i \geq 1$, $R_i := [2 \cdot 3^{-i}, 3^{-i+1})$ for all $i \geq 2$, and $R_i = [2 \cdot 3^{-i}, 3^{-i+1}]$ for $i = 1$. Note that we have

$$\mathcal{X} = \bigcup_{i=1}^{\infty}(L_i \cup R_i) \ ,$$

and moreover, $L_1, R_1, L_2, R_2, \dots$ do not overlap, thus they partition $\mathcal{X}$. Therefore, for any $x \in \mathcal{X}$, there exists $i(x) \geq 1$ such that $x \in L_{i(x)} \cup R_{i(x)}$ and $x \notin L_l \cup R_l$, for any $l \neq i$. So we have

$$\tilde{f}^j(x) = 4\bar{a}_{ij} X_{ij} \left((3^{i(x)} x - 1)(2 - 3^{i(x)} x)\mathbb{I}(x \in L_{i(x)}) - (3^{i(x)} x - 2)(3 - 3^{i(x)} x)\mathbb{I}(x \in R_{i(x)})\right) \ .$$

Let $x, x' \in \mathcal{X}$ and $j, l \in [m]$. We define $\tilde{\boldsymbol{f}}(x) = [\tilde{f}^1(x), \dots, \tilde{f}^m(x)]^{\mathsf{T}}$. Note that we have $\mathbb{E}[\tilde{\boldsymbol{f}}(x)] = [0, \dots, 0]^{\mathsf{T}}$. Moreover, we let $\tilde{\boldsymbol{k}}(x, x')$ denote the covariance matrix of the $m$-output GP $(\tilde{\boldsymbol{f}}(x))_{x \in \mathcal{X}}$. Note that we have

$$\mathbb{E}[\tilde{f}^j(x)\tilde{f}^l(x)] = 0, \ \text{ for } j \neq l \ ,$$

due to independence of $X_{ij}$ across objectives which implies that

$$\tilde{\boldsymbol{k}}(x,x) = \begin{bmatrix} \tilde{k}^{11}(x,x) & \dots & 0 \\ \vdots & & \vdots \\ 0 & \dots & \tilde{k}^{mm}(x,x) \end{bmatrix}$$

Now let $A = (a_{pq})_{p,q \in [m]} \in \mathbb{R}^{m \times m}$ be a square matrix such that $\|A_j\|_2 = 1$, for all $j \in [m]$, where $A_j$ denotes the $j$th row of $A$. Furthermore, let us define $\boldsymbol{f}(x) = A\tilde{\boldsymbol{f}}$. Let $\boldsymbol{k}$ be the covariance function associated with the $m$-output GP $(\boldsymbol{f}(x))_{x \in \mathcal{X}}$. Assume that $T$ designs of the form $x_i = 1/3^i + 1/(2 \cdot 3^i)$ are selected for evaluation and subsequently the noisy observations $\boldsymbol{y}_i$ are obtained, for $i \leq T$. Note that in this case we have $i(x_i) = i$. Now we explicitly calculate the variances of these evaluated points at all objectives. Let $j \leq m$ and $i \leq T$. We have

$$k^{jj}(x_i, x_i) = \mathbb{E}[(A\tilde{\boldsymbol{f}}(x_i))(A\tilde{\boldsymbol{f}}(x))^{\mathsf{T}}] = A_j \tilde{\boldsymbol{k}}(x_i, x_i) A_j^{\mathsf{T}} = \sum_{l=1}^{m} \tilde{k}^{ll}(x_i, x_i) a_{jl}^2 \ ,$$

where the third equality follows from the fact that $\tilde{\boldsymbol{k}}(x, x)$ is diagonal. Now we have

$$\tilde{f}^j(x_i) = 4\bar{a}_{ij} X_{ij} \left( \left(3^i x - 1\right)\left(2 - 3^i x\right) \mathbb{I}(x \in L_i) - \left(3^i x - 2\right)\left(3 - 3^i x\right) \mathbb{I}(x \in R_i) \right)$$
$$= 4\bar{a}_{ij} X_{ij} \left( 3^i \left( \frac{1}{3^i} + \frac{1}{2 \cdot 3^i} \right) - 1 \right) \left( 2 - 3^i \left( \frac{1}{3^i} + \frac{1}{2 \cdot 3^i} \right) \right) = 4\bar{a}_{ij} X_{ij} \frac{1}{4} = \bar{a}_{ij} X_{ij} \ .$$

Therefore, we have that $\tilde{k}^{ll}(x_i, x_i) = \mathbb{E}[(\tilde{f}^l(x_i))^2] = \bar{a}_{il}^2$, from which we obtain

$$k^{jj}(x_i, x_i) = \sum_{l=1}^{m} \bar{a}_{il}^2 a_{jl}^2 \leq \sum_{l=1}^{m} \bar{a}_{1l}^2 a_{jl}^2 \leq \|A_j\|_2^2 = 1 \tag{45}$$

using the assumptions on the sequence $(\bar{a}_{il})$ and the matrix $A$. The observations at different designs are uncorrelated, hence independent, by the choice of the designs, and this implies that the posterior distribution is the same as the prior distribution. This, together with Proposition 2 implies

$$\gamma_T \geq I(\boldsymbol{y}_{[T]}; \boldsymbol{f}_{[T]}) \geq \frac{1}{m} \sum_{i=1}^{T} \sum_{j=1}^{m} \frac{1}{2} \log \left( 1 + \sigma^{-2}(\sigma_{i-1}^j(x_i))^2 \right) = \frac{1}{m} \sum_{i=1}^{T} \sum_{j=1}^{m} \frac{1}{2} \log \left( 1 + \sigma^{-2}(\bar{a}_{ij}^2) \right)$$
$$\geq \frac{T}{2m} \sum_{j=1}^{m} \log \left( 1 + \frac{\bar{a}_{Tj}^2}{\sigma^2} \right) \geq T \frac{1}{2m} \sum_{j=1}^{m} \frac{\bar{a}_{Tj}^2/\sigma^2}{1 + \bar{a}_{Tj}^2/\sigma^2} = T \frac{1}{2m} \sum_{j=1}^{m} \frac{\bar{a}_{Tj}^2}{\bar{a}_{Tj}^2 + \sigma^2} \ ,$$

where in the third inequality we have used the fact that the sequence $(a_{ij})_{i=1}^{\infty}$ is decreasing and in the fourth inequality we use the fact that $\log(1 + x) \geq x/(1 + x)$. Thus, we obtain $\gamma_T = \Omega(T)$. By Theorem 1, we see that the information-type quantity $\sqrt{C\beta_T \gamma_T/T}$ increases logarithmically in $T$ and the metric dimension-type quantity $K_1 \beta_T T^{\frac{-\alpha}{\bar{D}+2\alpha}} (\log T)^{\frac{-(\bar{D}+\alpha)}{\bar{D}+2\alpha}} + K_2 T^{\frac{-\alpha}{\bar{D}+2\alpha}} (\log T)^{\frac{\alpha}{\bar{D}+2\alpha}}$ decreases exponentially in $T$, for any choice of $\bar{D} > D_1$ and $0 < \alpha \leq 1$.

On the other hand, for a suitably chosen metric $d$, the first part of Assumption 1 holds. Similar to equation 45, it can also be checked that $k^{jj}(x, x) \leq 1$ for all $x \in \mathcal{X}$. Hence, the second part of Assumption 1 holds as well. ∎

