# OpenReview forum: "Beyond Grids: Multi-objective Bayesian Optimization With Adaptive Discretization"
_TMLR — Accepted by TMLR_

### Review · Reviewer_vHvi · 2025-05-25

**Summary Of Contributions:**

This paper proposes a multi-objective Bayesian optimization method that finds the \\(\epsilon\\)-accurate Pareto set.
The proposed method is based on a tree-based adaptive discretization and an extension of the \\(\epsilon\\)-PAL algorithm proposed by Zuluaga et al. (2016).
The authors prove a regret bound of the proposed method and show that it outperforms existing methods empirically on GP sample paths.

**Audience:**

Yes

**Claims And Evidence:**

No

**Requested Changes:**

Restructure and rewrite Section 4.

**Strengths And Weaknesses:**

1. The writing is quite disappointing, especially for a journal submission.
At its current stage, the proposed method is vague and hard to understand.
In particular, Section 4 is somewhat unreadable due to the poor writing.
I am still quite lost after spending quite some time reading the paper (and I am familiar with relevant tools used in the paper).
The following are some suggestions for improving the writing.
    - Deferred definitions / undefined symbols.
    Many symbols are not defined at their first appearance, e.g., \\(\beta_\tau\\) and \\(V_h\\) at the top of page 9.
    Some notations are not defined at all throughout the main paper, e.g., \\(\bar D\\) in Theorem 1.
    - Unnecessary definitions.
    The notions of covering and packing numbers in Definition 7 are never used in the main paper.
    There is no point defining a notation but never use it.
    They should be deferred to the appendix.
    Otherwise, they are merely a distraction to the readers.
    - Please include a pseudo code in the main paper to describe the **entire** algorithm, along with a **detailed caption**.
    The main components of the proposed algorithm are described in three subsections 4.1, 4.2, and 4.3.
    However, this is not enough to clearly describe the proposed algorithm.
    - Please include a text description every time a symbol appears (or at least do so as often as possible).
    Taking section 5.1 as an example, instead of writing "if \\(\mathcal{S}_t = \emptyset\\) at the begining of round \\(t\\), ...", write it as "if the *undecided nodes* \\(\mathcal{S}_t = \emptyset\\) at the begining of round \\(t\\), ...".
    Without a text description of the symbol, the readers often need to go back looking for its definition every time a symbol appears.

1. Weak experiment.
    - The only experiment is based on GP sample paths.
    Please consider include more objective functions.
    - Please consider include a few more recent baselines, e.g., Daulton et al. (2021).

1. Not necessarily a weakness, but a question.
The acquisition function is based on querying the point with the largest radius of its confidence hyper-rectangle, as shown in Eq (2).
However, it appears that this is a pure exploration strategy.
How does it balance exploration and exploitation, as is often crucial in Bayesian optimization?

Daulton, M. Balandat, and E. Bakshy. Parallel Bayesian Optimization of Multiple Noisy Objectives with Expected Hypervolume Improvement. Advances in Neural Information Processing Systems 34, 2021.

---

> ### Author Response · Authors · 2025-06-18
> **Response**
>
> We thank the reviewer for their insightful and beneficial comments. We have tried to address all of them in the following. Changes made in the manuscript in response to the reviewers’ comments are marked in blue.
>
> **Comment:** "The writing is quite disappointing, especially for a journal submission. At its current stage..."
>
> **Response:** We have restructured Section 4 as per the suggestions of the reviewer. All changes can be found in the edited manuscript. In particular, we have first added an introductory paragraph which intuitively explains the rationale of the algorithm. We reproduce it here for convenience.
>
> “On a high level, our algorithm's rationale can be explained as follows. First, our algorithm maintains an adaptively changing resolution over the design space, structured along so-called nodes of the tree. That is, starting from the center of the space (the center node), we only 'zoom in' (thus expanding the tree) on points that are of potential interest.  In every iteration of the algorithm, we are given the current set of active nodes in our tree-based partition of the space. These nodes serve as proxies for regions of points in the design space. Now, the goal of the algorithm is to return a set of nodes (and, as a consequence, their associated regions) that form an approximate Pareto front with high probability. In order to do that, the algorithm maintains a confidence region for every active node. Applying worst-case arguments over these confidence regions, the algorithm decides which points to discard in every iteration, and which points to maintain as potentially optimal (in the Pareto sense). Next, we move points about which we are confident enough to a predicted Pareto set. Basically, we keep shrinking the set of points about which are undecided, and we keep growing the set of points which we believe are approximately optimal. Finally, if enough information is obtained on the most uncertain active node, then we decide to expand it into children nodes, since there is enough reason to believe that more relevant information will be obtained from those new nodes.”
>
> Next, since we omit the discussion on well-behaved metric spaces to the Appendix, we also modify the paragraph where we introduce such structure as follows:
>
> “Since iteration over individual designs is not possible in large spaces, we will instead iterate over 'regions' of interest. Here, we focus on subspaces $\mathcal{X}$ of the Euclidean space $\mathbb{R}^{m'}$, for some $m'\in\mathbb{N}$. However, we do this purely for simplicity of presentation. In Appendix, we show that our analysis holds for any general `well-behaved' metric spaces. For such a metric space,  one can easily partition the design space along a tree structure, each level $h$ of which is associated with a partition of $\mathcal{X}$ into $N^h$  equal-sized regions $X_{h,i}$, centered at a node $x_{h,i}$, for all $0\leq i\leq N^h$, where $N\in \mathbb{N}$.”
>
> We also introduce the $\beta$ and $V_h$ terms as follows:
>
> “Here, $\beta_\tau\in O(\log(\tau^2/\delta))$ and $V_h\in\tilde{O}(\rho^{\alpha h})$ is a high probability upper bound on the maximum variation of the objective $j$ inside region $X_{h,i}$.”
>
> We moreover provide explanations of the discarding and covering phases as follows:
>
> “Basically, what the condition of Definition 7 is saying is that, if the best possible value that $x$ can have is still approximately dominated by the worst possible value that $y$ can have, then we can say that $y$ dominates $x$ is with overwhelming probability.”
> “Intuitively, this step determines which points can be safely predicted to be in the approximately accurate Pareto set if there is no other active node which approximately dominates it in the worst case possible.”
>
> Finally, the discussion on well-behaved metric spaces is relegated to Appendix A.
>
> **Comment:** "Deferred definitions / undefined symbols. Many symbols are not..."
>
> **Response:** We thank the reviewer for raising this point. While the main reason we do not provide detailed definitions of $V_h$ or $\beta_\tau$ was for simplicity of presentation, we agree that it might be confusing to the reader. Thus, we have now included their definitions in the main paper. We first specify their dependence on important parameters in Section 4, and then formally define them in the statement of Theorem 1.
>
> **Comment:** "Unnecessary definitions. The notions of covering and packing numbers..."
>
> **Response:** We agree with the reviewer. We have deferred such discussion to the Appendix and kept only the necessary discussion in the main paper. In particular, we provide arguments related to Euclidean spaces in the main paper and mention that our analysis holds for general ‘well-behaved’ spaces, deferred to Appendix.

---

> ### Author Response · Authors · 2025-06-18
> **Response**
>
> **Comment:** "Please include a pseudo code in the main paper to describe the..."
>
> **Response:** Yes, we have included the main algorithm pseudo-code in the main paper to facilitate comprehension. The changes can be seen in the updated manuscript. A detailed caption is provided for all figures which illustrate the algorithm, and an additional Figure 4 which is meant to illustrate several steps of the procedure. We hope this will enhance the understanding of our method.
>
> **Comment:** "Please include a text description every time a symbol appears..."
>
> **Response:** We have taken into account all of these suggestions in the rewrite of Section 4.
>
> **Comment:** "Please consider include a few more recent baselines,..."
>
> **Response:** We have added the qNEHVI algorithm of Daulton et al., 2021.
>
> **Comment:** "Not necessarily a weakness, but a question. The acquisition function is based on..."
>
> **Response:** Indeed, the reviewer is correct in pointing out the exploratory nature of our evaluation strategy. Note that we evaluate the node which offers the most information (i.e. has the highest uncertainty). This takes care of exploration. On the other hand, through our $\epsilon$-covering phase, we consistently populate the approximate Pareto set, which is our end goal. It is worth mentioning that, strictly speaking, our setting is not the most suitable example of showcasing the exploration-exploitation paradigm, since we are interested not in regret minimization, but in Pareto set identification in as few samples as possible.

---

> > ### Comment · Reviewer_vHvi · 2025-07-02
> >
> > I acknowledge the revision the authors have made which substantially improves the draft. I have no further question.

---

### Review · Reviewer_hZSi · 2025-05-26

**Summary Of Contributions:**

The problem considers the problem of black-box optimization of multiple objectives. The key idea is to use a tree adaptive discretization with Pareto active learning (PAL). The paper provides two sample-complexity bounds: information-type and metric-dimension type. Experiments are shown on 1D synthetic objective functions.

**Audience:**

Yes

**Claims And Evidence:**

Yes

**Requested Changes:**

- I encourage explicitly writing the key innovations that were involved in extending the existing ideas to multiobjective settings and what are the challenges of the extension.

**Strengths And Weaknesses:**

- The paper is well written. I found the related work section describing the multiobjective blackobx optimization literature well.

- The paper is mostly a straightforward but sound synthesis of two existing ideas from black-box optimization: extending multiobjective pareto-active learning (PAL) with HOO/BamSOO ([1,2,3]) style tree adaptive discretization.

- The key novel step in PAL algorithm to this multiobjective setting is in the discard phase (section 4.2). The paper uses the idea to discard a cell when the best corner of the rectangle formed by its points is \epsilon-dominated by the worst corner of some pessimistic rectangle (definition 9). This "rectangular dominance" idea is known to be very conservative ([4,5]) and ignores any correlation between objectives. The second new step in the paper is to use the norm of the objective vector in refininig/evaluating phase.

- The paper also mentions noisy observations in the beginning but it is not clear how is that captured in the theoretical analysis. Please mention if I missed something.

[1] Wang, Ziyu, et al. "Bayesian multi-scale optimistic optimization." Artificial Intelligence and Statistics. PMLR, 2014.

[2] Shekhar, Shubhanshu, and Tara Javidi. "Gaussian process bandits with adaptive discretization." (2018): 3829-3874.

[3] Zuluaga, Marcela, et al. "Active learning for multi-objective optimization." International conference on machine learning. PMLR, 2013.

[4] Suzuki, Shinya, et al. "Multi-objective Bayesian optimization using Pareto-frontier entropy." International conference on machine learning. PMLR, 2020.

[5] Zhang, Richard, and Daniel Golovin. "Random hypervolume scalarizations for provable multi-objective black box optimization." International conference on machine learning. PMLR, 2020.

---

> ### Author Response · Authors · 2025-06-18
> **Response**
>
> We thank the reviewer for their insightful and beneficial comments. We have tried to address all of them in the following. Changes made in the manuscript in response to the reviewers’ comments are marked in blue.
>
> **Comment:** "The paper also mentions noisy observations in the beginning but it is not clear how..."
>
> **Response:**  If the design model is captured by our Gaussian process, then the confidence hyper-rectangles provide high-probability bounds on the mean of the observations, which also allows for noisy observations.
>
> **Comment:** "I encourage explicitly writing the key innovations that were involved in extending..."
>
> **Response:** We should clarify that the extension is not to the multi-objective setting. Zuluaga et al. (2016) already analyses a multi-objective setting but for a finite design space. Our extension is to general compact design spaces, which may involve infinitely many points. Applying fixed discretization procedures would be sub-optimal at best, first due to a fixed level of granularity, and then due to unnecessary partitioning in irrelevant regions. Thus, we utilize the most well-known efficient approach for such settings, namely, adaptive discretization.
>
> Now, the analysis from the finite setting to such a setting does not directly generalize. There are several key challenges which require novel algorithmic design tailored for such a setting. We mention them here and will include them in the final version of the paper:
> 1. First, due to the nature of the problem, we need to properly define novel confidence hyper-rectangle objects which are designed to capture uncertainty, not only over individual designs, but over whole regions. This requires integrating information and metric-type components into our hyper-rectangle definitions.
> 2. Second, the refining, evaluating and discarding steps of our procedure are substantially different from their analogues in e-PAL. Again, this is due to the large space of designs to consider. Here, we need to be confident for regions of space, rather than individual points, and choose whether to discard them or maintain them.
> 3. Due to uncertainty and limited budget, we need to make good use of our iterations. Specifically, in addition to the Pareto front identification problem in e-PAL, here we also have the problem of choosing whether to expand our tree of nodes, and in which direction to expand it to (note that the tree is not uniformly explored). Thus, we need additional conditions for that which involve information-theoretic and geometric components integrated into a threshold condition.
> 4. All of the above necessitate new additional results which involve algebraic and analytic arguments to carry through the analysis. Specifically, results such as Lemma 8 (which requires a careful step-by-step analysis applicable only to our setting), Lemma 10, Lemma 12 and Lemma 16, are all results which require non-trivial reasoning steps necessary for our setting.
>
> We also briefly discuss the novelty of our approach in the paper in the following paragraph (reproduced here for convenience):
>
> In order to prove the bounds in Theorem 1, even for infinite X , we propose a novel way of defining the confidence hyper-rectangles and refining them. Since Adaptive e-PAL discards, e-covers and refines/evaluates in ways different from e-PAL in Zuluaga et al. (2016), we use different arguments in the proof to show when the algorithm converges and what it returns when it converges. In particular, for the information-type bound, we exploit the dependence structure between the objectives. Moreover, having two different bounds allows us to use the best of both, as it is known that for certain kernels, the metric dimension-type bound can be tighter than the information-type bound.

---

> > ### Comment · Reviewer_hZSi · 2025-06-22
> >
> > Thanks to the authors for their response. I am happy with the response.

---

### Review · Reviewer_E2JZ · 2025-05-26

**Summary Of Contributions:**

The authors consider the problem of identifying (approximate) Pareto sets for a multi-objective optimization problem where the objectives to be optimized are uncertain and need be learned from data. To deal with function uncertainty, the authors use Gaussian Process surrogates to model each objective. To deal with the large (and possibly infinite) cardinality of the input space X, the authors use a tree-based adaptive discretization approach to progressively partition input space and refine the putative Pareto set. The authors provide rigorous bounds on the sample complexity of their Pareto set identification algorithm (Adaptive $\epsilon$-PAL) and provide an empirical comparison to a few baseline methods. The main contribution of the submission would appear to be its construction of an appropriate adaptive discretization strategy that is appropriate to large/infinite input spaces and that admits a rigorous termination and sample complexity analysis.

**Audience:**

Yes

**Claims And Evidence:**

Yes

**Requested Changes:**

Points of clarification that I think need to be addressed:
- You assume a shrinking "cumulative confidence hyper-rectangle" (eqn 1). This may make sense in a world with a fixed kernel, but probably doesn't make much sense if the kernel hyperparameters are chosen adaptively. Do you see this assumption as crucial to the construction of the algorithm per se, or is this more of a convenience for the theoretical analysis? Similarly, this seems to be related to the fact that (as I understand it) that once points are added to $\mathcal{P}$ they are never removed. Can you please comment on this? More broadly, given the many choices made in designing the algorithm, I would love to see a discussion of which choices in the algorithm are largely driven by a desire to simplify the theoretical analysis, as opposed to choices that are thought to be important for good performance.
- I'm a bit confused on the nature of $V_h$, which we are told is a "high probability upper bound on the maximum variation of the objective". But it is also described as an input to Algorithm 1. What is this quantity exactly and what role does it play? On a related noted, what is the nature of the assumption on $V_h$ in Thm 1? Why is this a reasonable assumption or, more broadly, what does this assumption assume/imply?
- I'm confused how quantities like $\mu(x_{h,i})$ are defined and computed, since to my understanding $x_{h,i}$ is a cell and not a particular point in the input space. Can you please specify more clearly how and when you move back and forth between cells and points in the input space?

Points of clarification that I think should/could be addressed:
- How would you modify the algorithm if the user doesn't want to explore the full Pareto front and is only interested, e.g., in the Pareto front within some hyperrectangle?
- You briefly mention that you "adopt an efficient algorithm in Kung". I'm assuming this is a novel algorithm. As such it would be worth describing in the appendix. Otherwise it would appear to be difficult for a reader to reproduce your results.
- While you have some helpful figures that illustrate what the algorithm is doing, given the complexity of the algorithm I would love to see a more complete walkthrough of ~3 iterations of the algorithm in a toy setting. Such a walkthrough could do wonders to help ground the reader in the core ideas of the algorithm. Perhaps it would be sufficient to expand Figure 3 and its caption.
- I would love to get more intuition about some of your particular algorithmic choices, e.g. the notion of "pessimistic" you adopt and why this might be expected to perform better than some other choices. Similarly, it would be great to see some of these choices ablated in the empirical evaluation.

**Strengths And Weaknesses:**

Strengths:
- While long and technically dense, the submission is generally carefully written and some care has been taken to make the rather complex algorithm comprehensible.
- The approach adopted appears technically sound and well-adapted to the problem at hand. To the best of my knowledge, it is novel.
- The authors provide a rigorous sample complexity analysis that includes both metric-dimension-based and information theoretic bounds. I cannot comment on the novelty or correctness of these bounds, as I have not gone through them in any detail.
- The algorithm performs well in a (rather limited) empirical evaluation.

Weaknesses:
- While the submission is generally carefully written, I believe readability could still be improved. I would suggest that the authors focus their main exposition on the core components of their algorithm and relegate more of the details to the appendix. For example, while it is a strength that the algorithm is applicable to well-behaved compact metric spaces, the various complications that go with the generalization beyond, say, a euclidean input space (introducing metric dimension etc) significantly complicate the exposition and do little to illuminate the core ideas of the algorithm. I believe a streamlined exposition could make it much easier for readers to digest the core ideas. For example: "While we have assumed in the above a euclidean input space, in fact our algorithm immediately generalizes to well-behaved compact metric spaces; for details see Appendix XXX." This is but one possible example. I believe the broader theme of focusing on core ideas could be profitably applied to the manuscript as a whole.
- The empirical evaluation is quite limited. In particular it only considers: i) objectives drawn from a known GP kernel and doesn't consider the realistic case of kernel mis-specification; ii) only considers the nearly deterministic low function noise scenario; iii) only considers one-dimensional inputs X and two-component objectives.
- Due to the limited empirical evaluation, it is largely unclear how well the algorithm would perform in practice. Unless the function noise is known to be small or the input space is of low dimension (such that isotropic length scales can be a reasonably good assumption) good performance from GP surrogates is generally expected to require fitting kernel hyperparameters one way or another (e.g. by maximizing the marginal likelihood). Granted that a version of the algorithm that chooses kernel hyperparameters adaptively would be much harder to analyze theoretically, so limiting the theoretical analysis to that setting is understandable. However, I myself would be much more likely to consider implementing and testing out the proposed algorithm if I had some confidence that it might work in more realistic settings. Absent that, I view the contribution of this submission as largely theoretical. To gain some confidence in the broader applicability of the proposed algorithm, at a minimum I would like to see empirical evaluation that considers noisy functions $f$, higher dimensional input spaces, and $m > 2$.

---

> ### Author Response · Authors · 2025-06-18
> **Response**
>
> We thank the reviewer for their insightful and beneficial comments. We have tried to address all of them in the following. Changes made in the manuscript in response to the reviewers’ comments are marked in blue.
>
> **Comment:** "While the submission is generally carefully written, I believe readability could still be improved. I would suggest that the authors focus their main exposition..."
>
> **Response:** As suggested, we have relegated the whole discussion on the structure of our design space to the Appendix and maintained a simple exposition relevant to Euclidean spaces in the main paper. Moreover, we have also restructured Section 4 and added a paragraph which intuitively explains our algorithm.
>
> **Comment:** "The empirical evaluation is quite limited..."
>
> **Response:** We clarify that our simulations do not assume known kernel hyperparameters. Instead, adaptive e-PAL is implemented with unknown kernel hyperparameters, which are learned as evaluations are made.
>
> **Comment:** "You assume a shrinking "cumulative confidence hyper-rectangle" (eqn 1). This may make sense in a world with a fixed kernel, but probably doesn't make much sense..."
>
> **Response:** We assume that, once kernel hyperparameters are chosen, the algorithm is run with that choice. Adaptive selection of hyperparameters can be performed along several runs of the algorithm. Then, if the hyperparameters remain fixed during a single run, the monotonous shrinking of the confidence hyper-rectangles is a straightforward implication of its definition. The essential necessity for guaranteeing such property is that, as time progresses, we need to be more confident about the nodes which we can discard, and that only happens if evaluation of certain points (which means more information about them) implies shrinkage of confidence hyper-rectangles.
>
> However, in practice, one can always reset the undecided and Pareto sets to their initial conditions whenever the kernel hyperparameters are being updated. This way, theory still holds and the algorithm can still utilize previous history of evaluations and the tree structure of the design space, thus providing a warm restart whenever the kernel is updated.
>
> Regarding the second point of the reviewer, once we discard a region, we are almost certain (with overwhelming probability) that there is at least one region that dominates it. Therefore, it cannot contain points that are in the Pareto front. In fact, this is a crucial point in the design of the algorithm, as it allows us to safely discard points that are not in the Pareto front. This, among the other algorithmic choices, is done not so much to simplify theoretical analysis, as it is to recover best-of-both-worlds types of bounds, which hold under a general setting, assuming only a regularized class of kernels and bounded variance. As we show in Appendix, this type of bounds is essential in settings where the information-type bounds can blow up, while dimension-type bounds decay gracefully. This guarantees that our algorithm will terminate and return a meaningful output.
>
> **Comment:** " I'm a bit confused on the nature of $V_h$, which we are told is a..."
>
> **Response:** The main reason why we did not include the detailed definition of V_h in the main paper is to avoid over-complicating terms. However, as per the suggestion of the reviewer, we now include both definitions in the main text.. The definition of $V_h$ depends on quantities that are known to the algorithm designer. Thus, $V_h$ is a fixed parameter for any depth h. It is precisely the deliberate choice of $V_h$ that ensures a high-probability upper bound in our proofs. Its intuitive role is to provide us with a threshold of variance beyond which it is safe to expand a given node into children nodes. In other words, if the variance of a node is lower than a function of $V_h$ (i.e. if we have gained enough information on the node), then, we should expand into children nodes (i.e. there are probably children nodes worth exploring at this point).  Basically, $V_h$ is a quantity that connects the structural properties of our space with the probabilistic properties of the points in that space. Finally, note that there is no assumption on $V_h$ in the statement of Theorem 1. We merely provide an O-notation exposition of it for comprehension.
>
> **Comment:** " I'm confused how quantities like $\mu(x_{h,i})$ are defined and computed..."
>
> **Response:** Let us recall that $x_{h,i}$ are, in fact, nodes which represent the centers of the cells denoted by $X_{h,i}$. Thus, we can exactly compute which points in our space these centers correspond to. For instance, in bounded Euclidean spaces, the first node would be the center of the space, which would then dictate the further partition of the space into children cells (and corresponding cell centers, aka nodes). Once we have access to such nodes, we can then compute the relevant quantities.

---

> ### Author Response · Authors · 2025-06-18
> **Response**
>
> **Comment:** "How would you modify the algorithm if the user doesn't want to explore the full Pareto front..."
>
> **Response:** A direct way to address the proposed setting is the following. If we are given the coordinates of the hyper-rectangle of interest, we can then restrict our design space only to such a hyper-rectangle. Thus, we can run our full algorithm on this hyper-rectangle instead of the whole space. This would then guarantee that the output of the algorithm would be an approximate Pareto front with high probability.
>
> **Comment:** "You briefly mention that you "adopt an efficient algorithm in Kung"..."
>
> **Response:** In the revised manuscript, we now explicitly specify that we use Algorithm 3.1 and Algorithm 4.1 from Kung et al. (1975). As these algorithms are already described in detail with pseudocode in the original reference, we refer readers directly to that source for full implementation details.
>
>
> **Comment:** "While you have some helpful figures that illustrate what the algorithm is doing, given the complexity..."
>
> **Response:** We completely agree with the reviewer that the complexity of the algorithm makes it hard for the reader to fully grasp its intuitive background. That was also the main reason for providing additional material such as Figure 3 and the other figures that explain the algorithmic steps, all of which are meant to facilitate understanding. Regarding the suggestion of the reviewers, the caption of Figure 3 already contains two steps of the algorithmic procedure. However, we now also include an additional figure which captures three iterations of the algorithm (evaluation is not shown for simplicity), with changes reflected both on the design space and the objective space.
>
> **Comment:** "I would love to get more intuition about some of your particular algorithmic choices, e.g. the notion of..."
>
> **Response:** The main reason we employ a pessimistic approach in our algorithm is due to the nature of our problem. Specifically, in order for us to safely discard regions which do not contain design points that are in the Pareto front, we need to first find those points which “may” be in the front. That is, even if the value of a node is in the worst-case corner of its confidence hyper-rectangle, we still check whether the point is dominated by other worst-case values or not. If not, then we cannot safely discard it. So we exclude such points from the set of points which we then consider for the discarding phase. This step is crucial since it prevents us from throwing away potentially optimal points just because we don’t have enough information about them.

---

> > ### Comment · Reviewer_E2JZ · 2025-06-21
> >
> > I thank the authors for answering my questions and taking my feedback into account when revising the manuscript. By adding Figure 4, relegating some details to the appendix, adding new baselines, etc., I believe the authors have substantially improved the manuscript. As such I am satisfied.
> >
> > To gain more confidence in the broader applicability of the proposed algorithm, I would love to see empirical evaluation that considers noisy functions, higher dimensional input spaces, and $m > 2$. However I leave it to the wisdom of the authors, other reviewers, and editor whether any such changes should be required.

---

### Decision · Action_Editor_xWbR · 2025-07-09

**Recommendation:** Accept as is

**Additional Comments:**

This paper proposes an adaptive algorithm for learning an $\varepsilon$-close Pareto front. The key idea is to adaptively discretize the input space and have a GP-estimator of the objective values over the discretization. The algorithm is analyzed and empirically evaluated.

This is a technically-sound paper with algorithmic novelty and sample-complexity bounds. The reviewers had several concerns:

1. The paper is technically heavy.

2. The algorithm is not clearly stated.

3. Weak experiments.

The authors addressed the concerns as follows:

1. A major rewrite. The technical content that was not necessary to expose the algorithm and bounds was moved to Appendix.

2. The pseudocode was added and additionally explained.

3. The authors added a hypervolume maximizing baseline.

My only remaining concern is that the empirical evaluation is weak. In particular, all problems in experiments are synthetic and small (one-dimensional input space and two objectives). Moreover, the hypervolume maximizing baseline is competitive with the proposed algorithm and based on my experience much more practical in higher dimensions (both input space and objective). That being said, the paper has other contributions than experiments and all reviewers support its acceptance.

**Audience:**

Yes

**Audience Explanation:**

This paper is on the intersection of bandits and multi-objective optimization, both of which are large communities.

**Claims And Evidence:**

Yes

**Claims Explanation:**

The paper proposes a novel algorithm, analyzes it, and empirically evaluates it. My only concern is that the empirical evaluation is weak. In particular, all problems in experiments are synthetic and small (one-dimensional input space and two objectives).